# Scale-invariant Optimal Sampling for Rare-events Data with Sparse Models

**Jing Wang**
Department of Statistics
University of Connecticut
Storrs, CT 06269
jing.7.wang@uconn.edu

**HaiYing Wang**
Department of Statistics
University of Connecticut
Storrs, CT 06269
haiying.wang@uconn.edu

**Hao Helen Zhang**
Department of Mathematics
University of Arizona
hzhang@math.arizona.edu

## Abstract

Subsampling is effective in tackling computational challenges for massive data with rare events. Overly aggressive subsampling may adversely affect estimation efficiency, and optimal subsampling is essential to mitigate the information loss. However, existing optimal subsampling probabilities depends on data scales, and some scaling transformations may result in inefficient subsamples. This problem is more significant when there are inactive features, because their influence on the subsampling probabilities can be arbitrarily magnified by inappropriate scaling transformations. We tackle this challenge and introduce a scale-invariant optimal subsampling function in the context of sparse models, where inactive features are commonly assumed. Instead of focusing on estimating model parameters, we define an optimal subsampling function to minimize the prediction error, using adaptive lasso as an example to outline the estimation procedure and study its theoretical guarantee. We first introduce the adaptive lasso estimator for rare-events data and establish its oracle properties, thereby validating the use of subsampling. Then we derive a scale-invariant optimal subsampling function that minimizes the prediction error of the inverse probability weighted (IPW) adaptive lasso. Finally, we present an estimator based on the maximum sampled conditional likelihood (MSCL) to further improve the estimation efficiency. We conduct numerical experiments using both simulated and real-world data sets to demonstrate the performance of the proposed methods.

## 1 Introduction

Rare-events data refer to binary-response data that are highly imbalanced, i.e., the number of zeros (a.k.a "controls" or "negative instances") are possibly hundreds or thousands of times as large as the number of ones (a.k.a. "cases" or "positive instances"). This type of data is common in various fields, such as medicine, natural science, political science, and social science, where examples of rare events can be rare diseases, natural disasters, wars, and financial crises, respectively. Modern technologies also prompt us to pay more attention to rare-events data. For example, in modern online recommendation systems, clicks are usually rare events compared with nonclicks. Statistical analyses, including parameter estimation and inferences, pose unique challenges for rare-events data because of high imbalance. In addition, rare-events data often involve sparse models. For instance,

38th Conference on Neural Information Processing Systems (NeurIPS 2024).

rare diseases might be linked to a limited number of key genes. Therefore, researchers frequently adopt sparse models in genome-wide association studies for analyzing rare diseases. A different yet related example is the use of deep neural networks to predict click-through rates in modern online recommendation systems. These networks are typically overparameterized, necessitating methods that balance rare-events data with the sparsity of the underlying models. Data balancing is a popular approach to overcome challenges caused by imbalanced data and is usually accomplished through subsampling the zeros [5, 15] or oversampling the ones [3, 12, 16, 4]. In addition, rare-events data are often massive in order to obtain an adequate number of ones, and computation is demanding. Therefore, we focus on the subsampling approach since it addresses the imbalance issue and reduce the computational burden simultaneously.

It is shown in [20] that the efficiency of parameter estimation is essentially determined by the number of ones for rare-events logistic regression, and subsampling does not reduce the estimation efficiency as long as sufficient zeros are kept. In case of excessive removal of zeros, [22] developed an optimal sampling approach to minimize information loss. However, the optimal sampling probabilities in [22] are scale-dependent, which may lead to inefficient results. Figure 1 illustrates the issue using a simulated example, with details in Section D.1 of the appendix. We generate the data from the same logistic regression model and transform one of the covariates with different scales $s = 0.01, 0.1, 1, 10$, and $100$. Then we apply two optimal subsampling methods in [22], labeled with "A-OS" and "L-OS" in Figure 1. It is observed that the prediction errors of A-OS and L-OS are significantly impacted by the data scaling. The A-OS may perform similarly to the Uni (simple random sampling or uniform sampling) in Figure 1a when $s = 0.01$; so is the L-OS in Figure 1b when $s = 100$. This scale-dependent issue is not specific to logistic regression and rare-events data in [22]; it is a wide concern in literature for various data types and models, including but not limited to [1, 29, 21, 14, 26, 25, 24]. In this paper, we propose a scale-invariant optimal subsampling method to overcome the issue. It is labeled "P-OS" in Figure 1.

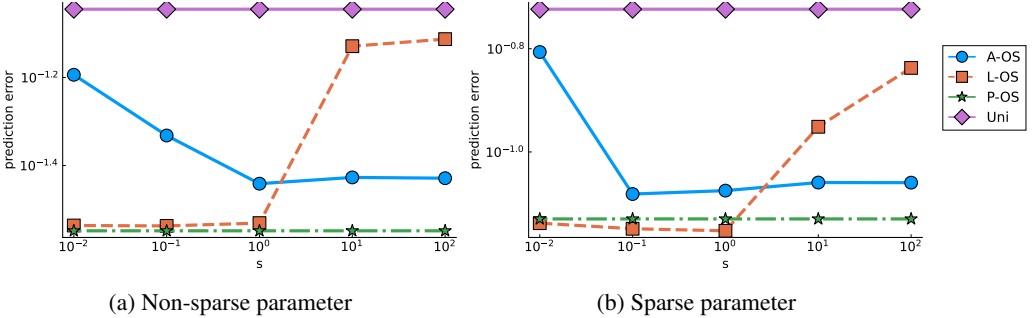

(a) Non-sparse parameter        (b) Sparse parameter

Figure 1: Prediction errors with different scale transformation of the same model. (a): with non-sparse parameter $(-1, -1, -0.01, -0.01, -0.01, -0.01)^{\mathrm{T}}$. (b): with sparse parameter $(-1, 0, 0, 0, 0, 0)^{\mathrm{T}}$.

The scale-dependence issue can seriously impact variable selection results for sparse models, where true parameters are zero for inactive covariates. In this case, inactive variables may be arbitrarily transformed without changing the underlying model, but the A-OS or L-OS would be highly influenced and may lead to misleading results. To resolve this issue, we investigate scale-invariant optimal subsampling in the context of variable selection, for which one main goal is to distinguish active and inactive features.

Penalty-based feature selection methods are widely used. Specifically, the adaptive lasso is a popular choice due to its oracle properties, convexity, and practical ease of implementation [see 30, 28]. While penalization methods have been used for bias reduction in rare-events analysis [7], variable selection for rare-events data has not been investigated. Conducting effective variable selection is difficult in the context of rare-events data analysis, mainly due to the scarcity of information available for ones. An inaccurate variable selection result can subsequently impact both the effectiveness of optimal subsampling and the efficiency of parameter estimation. In this paper, we address the challenge of variable selection in the context of rare-events data. First, we propose the full data adaptive lasso and study its theoretical properties. Next, we introduce a novel subsampling estimator that seamlessly combines penalty-based variable selection and optimal sampling into one unified framework for rare-events data. The implementation of the adaptive lasso requires a pilot estimator to construct

data-dependent weights for covariates. Given that optimal sampling also relies on pilot estimates [see 23, 1], the adaptive lasso emerges as a natural choice for conducting variable selection method in the context of subsampled rare-events data. We validate the new estimators by proving their oracle properties and also develop an efficient algorithm to facilitate their practical implementation when handling massive real-world data sets. In summary, our main contributions are listed as follows:

- We propose scale-invariant optimal subsampling to enhance parameter estimation and variable selection. Existing optimal subsampling methods are scale-dependent, which may lead to unreliable or misleading results.

- We define adaptive lasso and establish its oracle properties for rare-events data, which show that the asymptotic variances are determined by the number of ones in the data and the active features in the model.

- We present a practical subsampling algorithm based on optimal probabilities that significantly reduces the computational burden and accelerates the optimization for penalty-based feature selection methods.

The rest of the paper is organized as follows. Section 2 introduces the model setup. Section 3 investigates nonuniform sampling and variable selection tailored for rare-events data. We propose new methods to construct scale-invariant optimal probabilities. Section 4 discusses theoretical properties of the MSCL estimator and presents a two-step algorithm to implement the proposed methods. Section 5 conducts numerical experiments on simulated and real data sets. Section 6 concludes the paper. Proofs and mathematical details are presented in the appendix.

## 2 Background and model setup

We use the subscript $_\mathrm{t}$ to indicate the true parameters. For a $p$-dimensional vector $\boldsymbol{x}$, we use $x_{(i)}$ to represent its $i$-th element. For an index subset $\mathcal{A} \subset \{i : 1, 2, ..., p\}$, we use $\boldsymbol{x}_{(\mathcal{A})}$ to denote the subvector of $\boldsymbol{x}$, whose elements correspond to the indexes in $\mathcal{A}$. Furthermore, we use $\boldsymbol{x}^{\otimes 2}$ to denote $\boldsymbol{x}\boldsymbol{x}^\mathrm{T}$, use "$\rightsquigarrow$" to denote convergence in distribution, use "$\xrightarrow{P}$" to denote convergence in probability, and use "$\xrightarrow{a.s.}$" to denote convergence almost surely. We use $\boldsymbol{I}$ to denote an identity matrix of a suitable dimension and use $\boldsymbol{0}$ to denote a vector of zeros of a suitable dimension.

Let $(\boldsymbol{x}_1, y_1), (\boldsymbol{x}_2, y_2), ..., (\boldsymbol{x}_N, y_N)$ denote $N$ sample points from the joint distribution of $(\boldsymbol{x}, y)$, where $\{\boldsymbol{x}_i\}_{i=1}^N$ denote the $p$-dimensional predictors and $\{y_i\}_{i=1}^N$ the binary responses. Assume that the probability of $y$ being a one ($y = 1$) given $\boldsymbol{x}$ is

$$p(\boldsymbol{x}; \boldsymbol{\theta}_\mathrm{t}) := \mathbb{P}(y = 1|\boldsymbol{x}) = \frac{e^{\alpha_\mathrm{t} + f(\boldsymbol{x}; \boldsymbol{\beta}_\mathrm{t})}}{1 + e^{\alpha_\mathrm{t} + f(\boldsymbol{x}; \boldsymbol{\beta}_\mathrm{t})}} = \frac{e^{g(\boldsymbol{x}; \boldsymbol{\theta}_\mathrm{t})}}{1 + e^{g(\boldsymbol{x}; \boldsymbol{\theta}_\mathrm{t})}},$$

where $\boldsymbol{\theta}_\mathrm{t} = (\alpha_\mathrm{t}, \boldsymbol{\beta}_\mathrm{t}^\mathrm{T})^\mathrm{T}$ is the vector of true parameters and $f(\boldsymbol{x}; \boldsymbol{\beta}_t)$ is a smooth function of $\boldsymbol{\beta}_t$. For rare-events data, $N_1 \ll N_0$, where $N_1 = \sum_{i=1}^N y_i$ is the number of ones (i.e. $y_i = 1$) and $N_0 = N - N_1$ is the number of zeros (i.e. $y_i = 0$). Following the model setup used in [22], we assume that $\alpha_\mathrm{t} \to -\infty$ as $N \to \infty$, which implies that, under appropriate moment conditions,

$$\frac{N_1}{N_0} = \frac{\mathbb{E}\{p(\boldsymbol{x}; \boldsymbol{\theta}_\mathrm{t})\}}{1 - \mathbb{E}\{p(\boldsymbol{x}; \boldsymbol{\theta}_\mathrm{t})\}} + o(1) = \mathbb{E}\{e^{\alpha_\mathrm{t} + f(\boldsymbol{x}; \boldsymbol{\beta}_\mathrm{t})}\} + o(1) \to 0, \text{ almost surely.} \quad (1)$$

Under this assumption, the asymptotic variance of the full data maximum likelihood estimator (MLE) is of order $1/N_1$ instead of $1/N$, indicating that the estimation efficiency is determined by the number of rare ones. Therefore, we can keep all the ones and sample the zeros to save computational costs. There could be a variance inflation due to aggressive subsampling, and [22] developed optimal subsampling functions to reduce the variance inflation. Specifically, the authors proposed non-uniform optimal sampling functions under the A- and L-optimality criteria, respectively, as follows: $\varphi_{\mathrm{A-OS}}^{\mathrm{scale}}(\boldsymbol{x}) \propto p(\boldsymbol{x}; \boldsymbol{\theta}_\mathrm{t})\|\boldsymbol{M}^{-1}\dot{g}(\boldsymbol{x}; \boldsymbol{\theta}_\mathrm{t})\|$ and $\varphi_{\mathrm{L-OS}}^{\mathrm{scale}}(\boldsymbol{x}) \propto p(\boldsymbol{x}; \boldsymbol{\theta}_\mathrm{t})\|\dot{g}(\boldsymbol{x}; \boldsymbol{\theta}_\mathrm{t})\|$, where $\boldsymbol{M} = \mathbb{E}\{e^{f(\boldsymbol{x}; \boldsymbol{\beta}_\mathrm{t})}\dot{g}^{\otimes 2}(\boldsymbol{x}; \boldsymbol{\theta}_\mathrm{t})\}$ and $\dot{g}(\boldsymbol{x}; \boldsymbol{\theta})$ denotes the derivative of $g(\boldsymbol{x}; \boldsymbol{\theta})$ with respect to $\boldsymbol{\theta}$. However, the sampling functions $\varphi_{\mathrm{A-OS}}^{\mathrm{scale}}(\boldsymbol{x})$ and $\varphi_{\mathrm{L-OS}}^{\mathrm{scale}}(\boldsymbol{x})$ proposed in [22] depend on the scale of $\boldsymbol{x}$, and may not perform well for certain measurement scale of $\boldsymbol{x}$. For example, if $g(\boldsymbol{x}; \boldsymbol{\theta}_\mathrm{t}) = \alpha_\mathrm{t} + \boldsymbol{x}^\mathrm{T}\boldsymbol{\theta}_\mathrm{t}$, then $\varphi_{\mathrm{L-OS}}^{\mathrm{scale}}(\boldsymbol{x})$ is proportional to $1 + \|\boldsymbol{x}\|$, which will be influenced by the scale of $\boldsymbol{x}$. Similarly, scale

changes in $\boldsymbol{x}$ may also change $\varphi_{\mathrm{A-OS}}^{\mathrm{scale}}(\boldsymbol{x})$, although the impact may not be in the same direction, as demonstrated in Figure 1. Besides parameter estimation, variable selection is another important topic, which has not been studied in the literature on rare-events data. This work aims to fill this gap.

## 3 Nonuniform sampling with variable selection for rare-events data

The adaptive lasso [30, 28] is a popular variable selection method because it has oracle properties and is easy to implement. We define the full data adaptive lasso for rare-events data as

$$\hat{\boldsymbol{\theta}}_{\mathrm{mle}}^{\mathrm{adp}} := \arg\max_{\boldsymbol{\theta}} \left\{ \sum_{i=1}^{N} [y_i g(\boldsymbol{x}_i; \boldsymbol{\theta}) - \log\{1 + e^{g(\boldsymbol{x}_i; \boldsymbol{\theta})}\}] - \lambda_N \sum_{j=1}^{p} \frac{|\beta_{(j)}|}{|\hat{\beta}_{\mathrm{pl}(j)}|^{\gamma}} \right\}, \qquad (2)$$

where $\lambda_N$ and $\gamma$ are tuning parameters, and $\hat{\boldsymbol{\beta}}_{\mathrm{pl}}$ is a consistent pilot estimator of $\boldsymbol{\beta}_t$. In practice, it is common to set $\gamma = 1$. In the literature, iterative algorithms such as coordinate descent are commonly used to solve the adaptive lasso [8]. However, their computational demand can become prohibitive when dealing with massive data. It is feasible to alleviate the computational burden by subsampling zeros and create a smaller subset of data for adaptive lasso. To be specific, consider Algorithm 1.

---

**Algorithm 1** Poisson Subsampling algorithm

---

1: For $i = 1, ..., N$:
2: **if** $y_i = 1$ **then**
3:     include $(\boldsymbol{x}_i, y_i)$ in the subsample;
4: **else**
5:     Compute $\varphi(\boldsymbol{x}_i)$ and generate $u_i \sim U[0, 1]$;
6:     **if** $u_i \leq \pi(\boldsymbol{x}_i, y_i)$ **then**
7:         include $(\boldsymbol{x}_i, y_i)$ and record $\rho\varphi(\boldsymbol{x}_i)$ in the subsample;
8:     **end if**
9: **end if**

---

The inclusion probability in Algorithm 1 for the $i$th observation is $\pi(\boldsymbol{x}_i, y_i) = y_i + (1 - y_i)\rho\varphi(\boldsymbol{x}_i)$, where $\rho$ is the baseline sampling rate for the zeros and $\varphi(\boldsymbol{x}) > 0$ satisfies $\mathbb{E}\{\varphi(\boldsymbol{x})\} = 1$. Let the subsample from Algorithm 1 be $\{\boldsymbol{x}_i^{\mathrm{sub}}, y_i^{\mathrm{sub}}\}_{i=1}^{N_{\mathrm{sub}}^*}$, which is biased since $\pi(\boldsymbol{x}_i, y_i)$'s depend on the responses. We introduce an Inverse Probability Weighting (IPW) adaptive lasso estimator to correct for the bias, defined as

$$\hat{\boldsymbol{\theta}}_{\mathrm{w}}^{\mathrm{adp}} := \arg\max_{\boldsymbol{\theta}} \left\{ \sum_{i=1}^{N_{\mathrm{sub}}^*} \frac{[y_i^{\mathrm{sub}} g(\boldsymbol{x}_i^{\mathrm{sub}}; \boldsymbol{\theta}) - \log\{1 + e^{g(\boldsymbol{x}_i^{\mathrm{sub}}; \boldsymbol{\theta})}\}]}{\pi(\boldsymbol{x}_i^{\mathrm{sub}}, y_i^{\mathrm{sub}})} - \lambda_N \sum_{j=1}^{p} \frac{|\beta_{(j)}|}{|\hat{\beta}_{\mathrm{pl}(j)}|^{\gamma}} \right\}. \qquad (3)$$

To save space, we put the general assumptions used throughout this paper in Section B.1 of the appendix. We use $\mathcal{A}$ to denote the set of indexes of active variables, i.e., $\mathcal{A} = \{j : \beta_{\mathrm{t}(j)} \neq 0\}$ and $\mathcal{A}^c$ to denote the set of indexes of inactive variables, i.e., $\mathcal{A}^c = \{j : \beta_{\mathrm{t}(j)} = 0\}$. We first study the asymptotic properties of $\hat{\boldsymbol{\theta}}_{\mathrm{w}}^{\mathrm{adp}}$ in the following theorem.

**Theorem 1.** *Let $\hat{\boldsymbol{\beta}}_{\mathrm{pl}}$ be a consistent pilot estimate such that $\lambda_N/(\sqrt{N_1}|\hat{\beta}_{\mathrm{pl}(j)}|^{\gamma}) \xrightarrow{P} \infty$ for $j \in \mathcal{A}^c$. Under Assumptions 1-4, if $\lambda_N/\sqrt{N_1} \to 0$, then the IPW adaptive lasso estimator defined in (3) has the following properties:*

1. *Consistency in variable selection: The estimated active set $\hat{\mathcal{A}}_{\mathrm{w}} := \{j : \hat{\beta}_{\mathrm{w}(j)}^{\mathrm{adp}} \neq 0\}$ satisfies that $\lim_{N \to \infty} \mathbb{P}(\hat{\mathcal{A}}_{\mathrm{w}} = \mathcal{A}) = 1$.*

2. *Asymptotic normality: The estimator of the active parameter vector satisfies that*

$$\sqrt{N_1} \boldsymbol{V}_{\mathrm{w}(\mathcal{A})}^{-1/2} (\hat{\boldsymbol{\theta}}_{\mathrm{w}(\mathcal{A})}^{\mathrm{adp}} - \boldsymbol{\theta}_{\mathrm{t}(\mathcal{A})}) \rightsquigarrow \mathbb{N}(\boldsymbol{0}, \boldsymbol{I}),$$

*where $\boldsymbol{V}_{\mathrm{w}(\mathcal{A})} = \mathbb{E}\{e^{f(\boldsymbol{x}; \boldsymbol{\beta}_{\mathrm{t}})}\} \boldsymbol{M}_{(\mathcal{A})}^{-1} \boldsymbol{M}_{\mathrm{w}(\mathcal{A})} \boldsymbol{M}_{(\mathcal{A})}^{-1} = \mathbb{E}\{e^{f(\boldsymbol{x}; \boldsymbol{\beta}_{\mathrm{t}})}\} \{\boldsymbol{M}_{(\mathcal{A})}^{-1} + c\boldsymbol{V}_{\mathrm{sub}(\mathcal{A})}\}$, $\boldsymbol{M}_{(\mathcal{A})} = \mathbb{E}\{e^{f(\boldsymbol{x}; \boldsymbol{\beta}_{\mathrm{t}})} \dot{g}_{(\mathcal{A})}^{\otimes 2}(\boldsymbol{x}; \boldsymbol{\theta}_{\mathrm{t}})\}$, $\boldsymbol{V}_{\mathrm{sub}(\mathcal{A})} = \boldsymbol{M}_{(\mathcal{A})}^{-1} \mathbb{E}\{\frac{e^{2f(\boldsymbol{x}; \boldsymbol{\beta}_{\mathrm{t}})}}{\varphi(\boldsymbol{x})} \dot{g}_{(\mathcal{A})}^{\otimes 2}(\boldsymbol{x}; \boldsymbol{\theta}_{\mathrm{t}})\} \boldsymbol{M}_{(\mathcal{A})}^{-1}$, $c =$*

$\lim_{N\to\infty} e^{\alpha_t}/\rho$, and $\dot{g}_{(\mathcal{A})}(\boldsymbol{x}; \boldsymbol{\theta}_t)$ consists of the elements of gradient vector $\dot{g}(\boldsymbol{x}; \boldsymbol{\theta}_t)$ with indexes in the active set $\mathcal{A}$.

**Remark 1.** *Theorem 1 shows that the estimation efficiency of $\hat{\boldsymbol{\theta}}^{adp}_{w(\mathcal{A})}$ is predominantly determined by the number of ones instead of the full data size. The term $c\boldsymbol{V}_{sub(\mathcal{A})}$ is the variation inflation due to subsampling. The full data adaptive lasso in (2) correspond to the scenario with $\rho = 1$ and $\varphi(\boldsymbol{x}) = 1$, for which $c = \lim_{N\to\infty} e^{\alpha_t}/\rho = 0$. Intuitively, $c$ can be interpreted as the imbalance rate in the subsample. If we include sufficient zeros ($c = 0$), the subsampling does not reduce the estimation efficiency of $\hat{\boldsymbol{\theta}}^{adp}_{w(\mathcal{A})}$.*

From Theorem 1, we see that there maybe information loss reflected as an inflated variance if $c \neq 0$. To minimize the information loss due to sampling, we derive optimal functions as follows, where $\varphi^{adp}_{A-OS}(\boldsymbol{x})$ corresponds to the A-optimality criterion [17] and $\varphi^{adp}_{L-OS}(\boldsymbol{x})$ corresponds to the L-optimality criterion [17] in design of experiments. Here, the A-optimality minimizes the trace of the asymptotic variance of $\hat{\boldsymbol{\theta}}^{adp}_{w(\mathcal{A})}$; the L-optimality focuses on the asymptotic variance of a linearly transformed estimator $\boldsymbol{M}_{(\mathcal{A})}\hat{\boldsymbol{\theta}}^{adp}_{w(\mathcal{A})}$, which is proportional to $\boldsymbol{M}_{w(\mathcal{A})}$. The A-optimality criterion has a more direct interpretation, while an advantage of the L-optimality criterion is that the resulting optimal function is often faster to calculate.

**Proposition 1.** *The A-optimal function that minimizes $tr(\boldsymbol{V}_{w(\mathcal{A})})$ is*

$$\varphi^{adp}_{A-OS}(\boldsymbol{x}) = \frac{p(\boldsymbol{x}; \boldsymbol{\theta}_t)\|\boldsymbol{M}^{-1}_{(\mathcal{A})}\dot{g}_{(\mathcal{A})}(\boldsymbol{x}; \boldsymbol{\theta}_t)\|}{\mathbb{E}\left\{p(\boldsymbol{x}; \boldsymbol{\theta}_t)\|\boldsymbol{M}^{-1}_{(\mathcal{A})}\dot{g}_{(\mathcal{A})}(\boldsymbol{x}; \boldsymbol{\theta}_t)\|\right\}}. \tag{4}$$

*The L-optimal function that minimizes $tr(\boldsymbol{M}_{w(\mathcal{A})})$ is*

$$\varphi^{adp}_{L-OS}(\boldsymbol{x}) = \frac{p(\boldsymbol{x}; \boldsymbol{\theta}_t)\|\dot{g}_{(\mathcal{A})}(\boldsymbol{x}; \boldsymbol{\theta}_t)\|}{\mathbb{E}\left\{p(\boldsymbol{x}; \boldsymbol{\theta}_t)\|\dot{g}_{(\mathcal{A})}(\boldsymbol{x}; \boldsymbol{\theta}_t)\|\right\}}. \tag{5}$$

Unlike the optimal sampling function in [22], $\varphi^{adp}_{A-OS}(\boldsymbol{x})$ (or $\varphi^{adp}_{L-OS}(\boldsymbol{x})$) relies only on the active variables. This implies that a first-step pilot estimator given by the adaptive lasso algorithm can benefit from sparse estimation methods when calculating optimal probabilities. For example, employing the standard lasso can effectively eliminate a large number of inactive variables to facilitate the computation of optimal $\varphi^{adp}_{A-OS}(\boldsymbol{x})$ and $\varphi^{adp}_{L-OS}(\boldsymbol{x})$. However, in practice, pilot estimators are often obtained from a small subsample size, introducing additional uncertainty. Therefore, it becomes crucial to exercise caution and be conservative by over-selecting variables during the first step to prevent the exclusion of important variables. As a consequence, although theoretically $\varphi^{adp}_{A-OS}(\boldsymbol{x})$ and $\varphi^{adp}_{L-OS}(\boldsymbol{x})$ do not depend on inactive variables, they are affected by inactive variables in practical implementations.

### 3.1 Scale invariant optimal function

As discussed in Section 1, scaling dependent optimal probabilities may impact the performance of variable selection in practice. To address the issue, we propose to construct a scale invariant optimal function by focusing on the prediction error of an estimator $\hat{\boldsymbol{\theta}}$, defined below.

$$\text{MSPE}(\hat{\boldsymbol{\theta}}) = \mathbb{E}_{\boldsymbol{x}}\left[\left\{p(\boldsymbol{x}; \hat{\boldsymbol{\theta}}) - p(\boldsymbol{x}; \boldsymbol{\theta}_t)\right\}^2\right] = \int \left\{p(\boldsymbol{x}; \hat{\boldsymbol{\theta}}) - p(\boldsymbol{x}; \boldsymbol{\theta}_t)\right\}^2 d\mathbb{P}_{\boldsymbol{x}},$$

where $\mathbb{P}_{\boldsymbol{x}}$ is the probability measure of $\boldsymbol{x}$. The probability term $p(\boldsymbol{x}; \boldsymbol{\theta}_t)$ involves both the covariates $\boldsymbol{x}$ and the parameter vector $\boldsymbol{\theta}_t$, and it often does not depend on the scale of $\boldsymbol{x}$. For example, in the logistic regression model, the value $p(\boldsymbol{x}; \boldsymbol{\theta}_t)$ is only related to $\boldsymbol{x}^T\boldsymbol{\beta}_t$. If we change the scale of $x_{(j)}$, the value of $\boldsymbol{\theta}_t$ would change accordingly under the same data-generating model and so $p(\boldsymbol{x}; \boldsymbol{\theta}_t)$ remains the same. Thus, re-scaling covariates would not affect this criterion. In the following, we give an optimal function that minimizes the prediction error.

**Theorem 2.** *Under the assumptions of Theorem 1, for the IPW adaptive lasso estimator defined in (3), its prediction error satisfies*

$$N_1 e^{-2\alpha_t}\text{MSPE}(\hat{\boldsymbol{\theta}}^{adp}_{w(\mathcal{A})}) \rightsquigarrow \mathbb{E}^{-1}\left\{e^{f(\boldsymbol{x};\boldsymbol{\beta}_t)}\right\} \boldsymbol{Z}^T_{(\mathcal{A})}\boldsymbol{M}^{1/2}_{w(\mathcal{A})}\boldsymbol{M}^{-1}_{(\mathcal{A})}\boldsymbol{\Omega}_{(\mathcal{A})}\boldsymbol{M}^{-1}_{(\mathcal{A})}\boldsymbol{M}^{1/2}_{w(\mathcal{A})}\boldsymbol{Z}_{(\mathcal{A})}. \tag{6}$$

where $\boldsymbol{Z}_{(\mathcal{A})} \sim \mathbb{N}(\boldsymbol{0}, \boldsymbol{I})$, and $\boldsymbol{\Omega}_{(\mathcal{A})} = \mathbb{E}\left[e^{2f(\boldsymbol{x};\boldsymbol{\beta}_{\mathrm{t}})}\dot{g}_{(\mathcal{A})}^{\otimes 2}(\boldsymbol{x}, \boldsymbol{\theta}_{\mathrm{t}})\right]$. *The optimal function that minimizes the asymptotic mean of the prediction error in (6) is given as*

$$\varphi_{\mathrm{P-OS}}^{\mathrm{adp}}(\boldsymbol{x}) = \frac{p(\boldsymbol{x};\boldsymbol{\theta}_{\mathrm{t}})\|\boldsymbol{\Omega}_{(\mathcal{A})}^{\frac{1}{2}}\boldsymbol{M}_{(\mathcal{A})}^{-1}\dot{g}_{(\mathcal{A})}(\boldsymbol{x};\boldsymbol{\theta}_{\mathrm{t}})\|}{\mathbb{E}\left[p(\boldsymbol{x};\boldsymbol{\theta}_{\mathrm{t}})\|\boldsymbol{\Omega}_{(\mathcal{A})}^{\frac{1}{2}}\boldsymbol{M}_{(\mathcal{A})}^{-1}\dot{g}_{(\mathcal{A})}(\boldsymbol{x};\boldsymbol{\theta}_{\mathrm{t}})\|\right]}. \tag{7}$$

We refer this prediction oriented criterion as P-optimality criterion. As we expect, the optimal function in (7) is unaffected by the scale of $\boldsymbol{x}$ for a class of functions $g$. The following proposition proves that $\varphi_{\mathrm{P-OS}}^{\mathrm{adp}}(\boldsymbol{x})$ is invariant to rescaling of $\boldsymbol{x}$.

**Proposition 2.** *If $g(\boldsymbol{x};\boldsymbol{\theta})$ satisfies that for every non-singular matrix $\boldsymbol{A}$ there exists a non-singular matrix $\boldsymbol{B}$, such that*

$$g(\boldsymbol{A}\boldsymbol{x};\boldsymbol{B}^{\mathrm{T}}\boldsymbol{\theta}) = g(\boldsymbol{x};\boldsymbol{\theta}), \tag{8}$$

*then, $\varphi_{\mathrm{P-OS}}^{\mathrm{adp}}(\boldsymbol{x})$ is invariant to scale changes of $\boldsymbol{x}$.*

**Remark 2.** *The condition in (8) is not restrictive and it is quite easy to satisfy. One simple example of $g(\boldsymbol{x};\boldsymbol{\theta})$ that satisfies the condition is a linear function $g(\boldsymbol{x};\boldsymbol{\theta}) = \alpha + \boldsymbol{x}^{\mathrm{T}}\boldsymbol{\beta}$, which corresponds to the logistic regression. The condition is also satisfied by more complex models. For example, consider an L-layer neural network*

$$g(\boldsymbol{x};\boldsymbol{W}^1, \boldsymbol{W}^2, ..., \boldsymbol{W}^L, \boldsymbol{b}^1, ..., \boldsymbol{b}^L) = f^L(f^{L-1}(...f^1(\boldsymbol{x}^{\mathrm{T}}\boldsymbol{W}^1 + \boldsymbol{b}^1))^{\mathrm{T}}\boldsymbol{W} + \boldsymbol{b}^L),$$

*where $\boldsymbol{W}^l$ are the weights and $\boldsymbol{b}^l$ are the biases in each layer, $l = 1, 2, ..., L$. If $\boldsymbol{x}$ is rescaled to $\boldsymbol{A}\boldsymbol{x}$, we can change $\boldsymbol{W}^1$ to $(\boldsymbol{A}^T)^{-1}\boldsymbol{W}^1$ so that the value of $g$ does not change. That is*

$$g(\boldsymbol{A}\boldsymbol{x};(\boldsymbol{A}^T)^{-1}\boldsymbol{W}^1, \boldsymbol{W}^2, ..., \boldsymbol{W}^L, \boldsymbol{b}^1, ..., \boldsymbol{b}^L) = f^L(f^{L-1}(...f^1(\boldsymbol{x}^{\mathrm{T}}\boldsymbol{W}^1 + \boldsymbol{b}^1))^{\mathrm{T}}\boldsymbol{W} + \boldsymbol{b}^L)$$
$$= g(\boldsymbol{x};\boldsymbol{W}^1, \boldsymbol{W}^2, ..., \boldsymbol{W}^L, \boldsymbol{b}^1, ..., \boldsymbol{b}^L).$$

## 4   Penalized MSCL estimator

The IPW estimator in (3) is not the most efficient estimator, because it assigns smaller weights for more informative data points with larger sampling probabilities. To improve the estimation efficiency, we propose the penalized MSCL estimator for variable selection given as

$$\hat{\boldsymbol{\theta}}_{\mathrm{mscl}}^{\mathrm{adp}} := \arg\max_{\boldsymbol{\theta}} \left\{ \sum_{i=1}^{N_{\mathrm{sub}}^*}[y_i^{\mathrm{sub}}g(\boldsymbol{x}_i^{\mathrm{sub}};\boldsymbol{\theta}) - \log\{1 + e^{g(\boldsymbol{x}_i^{\mathrm{sub}};\boldsymbol{\theta})+l_i^{\mathrm{sub}}}\}] - \lambda_N \sum_{j=1}^{p} \frac{|\beta_{(j)}|}{|\hat{\beta}_{\mathrm{pl}(j)}|^{\gamma}} \right\}, \tag{9}$$

where $l_i^{\mathrm{sub}} = -\log\left\{\rho\varphi(\boldsymbol{x}_i^{\mathrm{sub}})\right\}$. The MSCL estimator introduced in [22] is defined as the minimizer of the objective function in (9), excluding the penalization term. In this paper, we extend this approach by proposing a penalized MSCL estimator to ensure model sparsity. We present the oracle properties of the penalized MSCL estimator in the following theorem.

**Theorem 3.** *Let $\hat{\beta}_{\mathrm{pl}}$ be a consistent pilot estimate such that $\lambda_N/(\sqrt{N_1}|\hat{\beta}_{\mathrm{pl}(j)}|^{\gamma}) \xrightarrow{P} \infty$ for $j \in \mathcal{A}^c$. Under Assumptions 1-3 and 5, if $\lambda_N/\sqrt{N_1} \to 0$, the estimator based on MSCL function with adaptive lasso penalty defined in (9) have the following properties:*

1. *Consistency in variable selection: The estimated active set $\hat{\mathcal{A}}_{\mathrm{mscl}} := \{j : \hat{\beta}_{\mathrm{mscl}(j)}^{\mathrm{adp}} \neq 0\}$ satisfies that $\lim_{N\to\infty} \mathbb{P}(\hat{\mathcal{A}}_{\mathrm{mscl}} = \mathcal{A}) = 1$*

2. *Asymptotic normality: The estimator of the active parameter vector satisfies that*

$$\sqrt{N_1}\boldsymbol{V}_{\mathrm{mscl}(\mathcal{A})}^{-1/2}(\hat{\boldsymbol{\theta}}_{\mathrm{mscl}(\mathcal{A})}^{\mathrm{adp}} - \boldsymbol{\theta}_{\mathrm{t}(\mathcal{A})}) \rightsquigarrow \mathbb{N}(\boldsymbol{0}, \boldsymbol{I}), \tag{10}$$

*where $\boldsymbol{V}_{\mathrm{mscl}(\mathcal{A})} = \mathbb{E}\left\{e^{f(\boldsymbol{x};\boldsymbol{\beta}_{\mathrm{t}})}\right\}\boldsymbol{\Lambda}_{\mathrm{mscl}(\mathcal{A})}^{-1}$ and $\boldsymbol{\Lambda}_{\mathrm{mscl}(\mathcal{A})} = \mathbb{E}\left[\frac{e^{f(\boldsymbol{x};\boldsymbol{\beta}_{\mathrm{t}})}\dot{g}_{(\mathcal{A})}^{\otimes 2}(\boldsymbol{x};\boldsymbol{\beta}_{\mathrm{t}})}{1+c\varphi^{-1}(\boldsymbol{x})e^{f(\boldsymbol{x};\boldsymbol{\beta}_{\mathrm{t}})}}\right]$.*

The penalized MSCL estimator has the same asymptotic variance as the MSCL estimator under the true model, indicating that it is more efficient than the penalized IPW estimator [22]. We prove this by comparing the asymptotic variances and present the result in the following theorem.

**Theorem 4.** *If the asymptotic variances $V_{w(\mathcal{A})}$ for $\hat{\boldsymbol{\theta}}_{w(\mathcal{A})}^{\mathrm{adp}}$ in (2) and $V_{\mathrm{mscl}(\mathcal{A})}$ for $\hat{\boldsymbol{\theta}}_{\mathrm{mscl}(\mathcal{A})}^{\mathrm{adp}}$ in (9), are finite, i.e., $0 < V_{w(\mathcal{A})}, V_{\mathrm{mscl}(\mathcal{A})} < \infty$, then $V_{\mathrm{mscl}(\mathcal{A})} \leq V_{w(\mathcal{A})}$, where the inequalities hold in the sense of Loewner ordering.*

Thus, we give a practical two-step algorithm based on the penalized MSCL estimator. Since the optimal sampling functions contain unknown values and the adaptive lasso penalty also requires a consistent pilot estimator to build weights, it is natural to combine optimal sampling and the adaptive lasso into one unified framework. We recommend to use the lasso for pilot estimation. One reason is that it does estimation and variable selection simultaneously, and excluding some inactive variables improves the estimation accuracy of optimal probabilities. This also reduces the computational burden for subsequent steps. Another reason is that the lasso estimator tends to include more variables in practice and therefore has a low risk of excluding important variables in the pilot step. We present an outline of the practical implementation in Algorithm 2. More details are given in Section C.

---

**Algorithm 2** Two-step subsampling adaptive lasso algorithm

---

1: First stage screening:
- Take a pilot sample of expected sample size $N_{\mathrm{pl}}$ using $\{\pi(y_i) = \rho_0 + y_i(\rho_1 - \rho_0)\}_{i=1}^{N}$ and obtain a lasso penalized MSCL pilot estimator and an estimated active set $\hat{\mathcal{A}}_{\mathrm{pl}}$.
- Calculate approximate optimal sampling probabilities $\{\hat{\pi}(\boldsymbol{x}_i, y_i) = y_i + (1 - y_i)\rho\hat{\varphi}(x_i)\}_{i=1}^{N}$ based on (4), (5), or (7).

2: Second stage screening: Use Algorithm 1 with the estimated optimal sampling probabilities to obtain a subsample of expected sample size $N_{\mathrm{sub}}$ and compute the adaptive lasso penalized MSCL estimator based on $\hat{\mathcal{A}}_{\mathrm{pl}}$.

---

## 5 Numerical experiments

In this section, we use numerical experiments on both simulated and real data to investigate the performances of our proposed optimal subsampling and variable selection procedures.

### 5.1 Simulation design

We consider a logistic regression with $g(\boldsymbol{x}; \boldsymbol{\theta}) = \alpha + \boldsymbol{x}^{\mathrm{T}}\boldsymbol{\beta}$ and the following three true parameters $\boldsymbol{\beta}_{\mathrm{t}}$ of dimension 50. We set different $\alpha_{\mathrm{t}}$ so that the proportion of ones is $0.005$:

(1) **Case A:** $\boldsymbol{\beta}_{\mathrm{t}} = (0.75, 0.75, \boldsymbol{0}_7^{\mathrm{T}}, 0.75, 0, 0.75, 0.75, \boldsymbol{0}_{37}^{\mathrm{T}})^{\mathrm{T}}$ and $\alpha_{\mathrm{t}} = -5.8$.

(2) **Case B:** $\boldsymbol{\beta}_{\mathrm{t}} = (3, -2, \boldsymbol{0}_7^{\mathrm{T}}, 0.85, 0, -0.75, \boldsymbol{0}_{38}^{\mathrm{T}})^{\mathrm{T}}$ and $\alpha_{\mathrm{t}} = -6.2$.

(3) **Case C:** $\boldsymbol{\beta}_{\mathrm{t}} = (3, 2, \boldsymbol{0}_7^{\mathrm{T}}, 0.85, \boldsymbol{0}_{40}^{\mathrm{T}})^{\mathrm{T}}$ and $\alpha_{\mathrm{t}} = -7.5$.

Here, $\boldsymbol{0}_d$ denotes the zero vector of dimension $d$. We use $p_{\mathcal{A}}$ and $p_{\mathcal{A}^c}$ to denote the number of active and inactive variables, respectively, and assume that $\boldsymbol{x}$ is a normal random vector. The active components $x_{(\mathcal{A},j)}$, $1 \leq j \leq p_{\mathcal{A}}$ of $\boldsymbol{x}$ have variances $0.25$ and the inactive components $x_{(\mathcal{A}^c,j)}$, $1 \leq j \leq p_{\mathcal{A}^c}$ of $\boldsymbol{x}$ have variances $100/p_{\mathcal{A}^c}^3, 100/(p_{\mathcal{A}^c} - 1)^3, ..., 100/3^3, 100/2^3, 100/1^3$. The correlation between the $i$-th and $j$-th elements of $\boldsymbol{x}$ is $0.5^{|i-j|}, 1 \leq i, j \leq p$. We repeat our experiments $S = 500$ times generating $N = 500000$ data points in each run and use a pilot sample of size $N_{\mathrm{pl}} = 500$ for obtaining pilot estimates based on the lasso. We consider uniform sampling, the full data lasso, and the full data adaptive lasso for comparison. We use the 5-fold cross-validation and Bayesian information criterion (BIC) to determine the tuning parameter $\lambda$ for the lasso and the adaptive lasso, and choose $\gamma = 1$ for the adaptive lasso.

#### 5.1.1 Estimation and prediction efficiency

We present the empirical median squared error (eMSE) for parameter estimation in Figure 2. All optimal sampling estimators outperform the uniform sampling. As the sampling rate increases, sampling estimators outperform the full data lasso estimator eventually in all of the three cases. Among the three optimal subsampling methods, $\hat{\boldsymbol{\beta}}_{\mathrm{P-OS}}^{\mathrm{adp}}$ performs better than the other two subsampling methods.

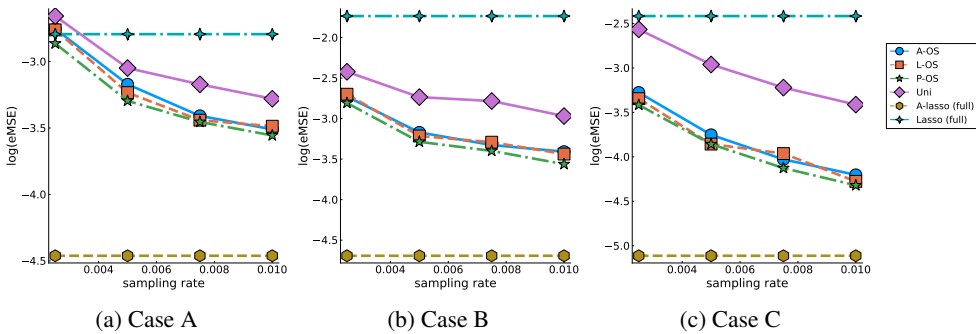

(a) Case A        (b) Case B        (c) Case C

Figure 2: eMSE for different true parameters with different sampling rates.

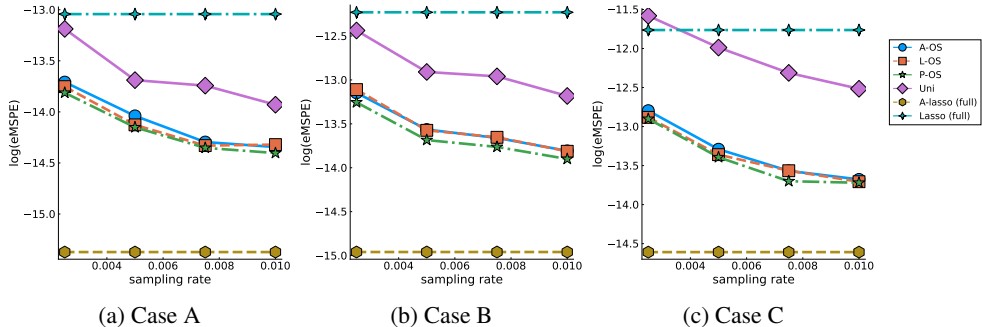

(a) Case A        (b) Case B        (c) Case C

Figure 3: eMPSE of estimated probability with different sampling rates.

Figure 3 shows the results of the empirical median squared prediction error (eMSPE). Similarly to the results of eMSE, optimal sampling estimators perform better than the uniform sampling, meaning optimal sampling results in less information loss. It is possible that sampling estimators outperform the full data lasso estimator as the sampling rate increases, despite that the latter uses all of the data. In general, $\hat{\beta}_{\mathrm{P-OS}}^{\mathrm{adp}}$ performs the best among the three optimal subsampling algorithms.

### 5.1.2 Variable selection and computational complexity

In this section, we discuss the results of variable selection in terms of the first stage screening and the second stage screening. Table 1 presents the mean numbers of selected variables in Case C, where the numbers in the parentheses are the corresponding standard errors. Results for Cases A and B are similar so are put in Table 4 of the appendix.

Table 1: Mean number of selected variables in Case C

| $\rho$ | first-stage | Uni | A-OS | L-OS | P-OS |
|---|---|---|---|---|---|
| 0.0025 | 13.27(0.34) | 2.84(0.02) | 2.97(0.02) | 2.96(0.02) | 2.96(0.02) |
| 0.005 | 12.46(0.32) | 2.94(0.02) | 3.04(0.03) | 3.05(0.03) | 3.06(0.03) |
| 0.0075 | 12.76(0.33) | 2.97(0.01) | 3.04(0.02) | 3.03(0.02) | 3.03(0.02) |
| 0.01 | 12.81(0.34) | 2.96(0.01) | 3.03(0.02) | 3.02(0.01) | 3.02(0.01) |

While the first stage screening significantly reduces the dimension in Table 1, it indeed includes inactive variables as expected. In the second stage screening, the mean numbers of selected variables are close to the true numbers of active variables for all subsampling methods. However, the mean number of selected variables from uniform sampling is smaller than the true number of active variables especially when the sampling rate is low. This indicates that the second-stage screening of uniform sampling may exclude active variable. We present the rates of missing active variables in Table 2 for Case C. It shows that uniform sampling has higher rates of excluding active variables than optimal subsampling procedures, so optimal sampling may be preferable in practice. Results for Cases A and B are similar and are put in Section E.1. We also investgate the rates of selecting the true model in that section.

Table 2: Rates of excluding active variables (false negative rate) in Case C

| $\rho$ | Uni | A-OS | L-OS | P-OS |
|---|---|---|---|---|
| 0.0025 | 0.168(0.017) | 0.086(0.013) | 0.088(0.013) | 0.084(0.013) |
| 0.005 | 0.100(0.013) | 0.068(0.011) | 0.066(0.011) | 0.066(0.011) |
| 0.0075 | 0.066(0.011) | 0.046(0.009) | 0.048(0.010) | 0.046(0.009) |
| 0.01 | 0.068(0.011) | 0.052(0.010) | 0.054(0.010) | 0.054(0.010) |

### 5.1.3 Computational time

We present the mean computational times of different algorithms in Table 3. Our codes are written in the *julia* programming language [2] and implemented on a Linux workstation. The lasso pathes are solved with *Lasso.jl* [13]. As shown in Table 3, subsampling algorithms significantly reduce the computational times compared with full data estimators. Although optimal sampling requires to calculate sampling probabilities, they use only about 0.77% of the computational time that the full data adaptive lasso requires. As we discussed in Section C.2, optimal sampling algorithms reduce both sample size and the data dimension. Therefore, the computational cost of the coordinate decent algorithm, which often requires a large number of iterations, is significantly reduced.

Table 3: Mean computational time (seconds)

| Case | Uni | A-OS | L-OS | P-OS | A-lasso (full) | Lasso (full) |
|---|---|---|---|---|---|---|
| A | 0.29 | 1.09 | 0.91 | 1.06 | 129.62 | 112.97 |
| B | 0.31 | 1.23 | 1.20 | 1.27 | 129.89 | 122.40 |
| C | 0.31 | 1.02 | 0.93 | 1.00 | 130.33 | 121.29 |

### 5.2 Real data

We evaluate the performances of proposed estimators on two real data sets.

(i) **Covtype data set:** It is available at `https://archive.ics.uci.edu/ml/datasets/covertype`, with $N = 581012$ observations and 54 covariates – 10 being quantitative and 44 being qualitative with dummy coding. We drop the 14th and 54th columns to avoid exact colinearity of the dummy variables. Our goal is to classify whether the forest cover type is Cottonwood/Willow (labeled as 1) or not (labeled as 0). The proportion of Cottonwood/Willow is 0.473%, which is highly imbalanced.

(ii) **Font data set:** It is available at `https://archive.ics.uci.edu/ml/datasets/Character+Font+Images`, with 0.50% of the $N = 832670$ responses being the GADUGI font. The first 10 covariates are about the value, size, and style of the characters and there are additional 400 pixel values of the $20 \times 20$ images. We remove the 4th, 9th, and 10th covariates because they are constants.

For both data sets, we apply Algorithm 2 on the logarithmic-transformed data. We use pilot samples of size $N_{\mathrm{pl}} = 1000$ for the covtype data and $N_{\mathrm{pl}} = 1500$ for the font data due to its higher dimension. Since we do not know the true parameter for real data, we use area under the curve (AUC) to measure the performances of subsampling algorithms. We repeat the experiment for $S = 500$ and compute the empirical median AUC using the full data. The results are summarized in Figure 4. As shown in Figure 4, nonuniform sampling outperforms uniform sampling in general. There is one case for font data set that $\hat{\boldsymbol{\beta}}_{\mathrm{A-OS}}^{\mathrm{adp}}$ is worse than the uniform sampling when the sampling rate is high. For the covtype data set, among the three estimators based on optimal sampling, $\hat{\boldsymbol{\beta}}_{\mathrm{P-OS}}^{\mathrm{adp}}$ performs the best and $\hat{\boldsymbol{\beta}}_{\mathrm{L-OS}}^{\mathrm{adp}}$ is worst. For the font data set, $\hat{\boldsymbol{\beta}}_{\mathrm{A-OS}}^{\mathrm{adp}}$ and $\hat{\boldsymbol{\beta}}_{\mathrm{L-OS}}^{\mathrm{adp}}$ are similar, and $\hat{\boldsymbol{\beta}}_{\mathrm{P-OS}}^{\mathrm{adp}}$ based on the scale invariant optimal sampling function is significantly better.

## 6 Conclusion and limitations

In this paper, we investigated the problem of scale-invariant optimal subsampling in the context of variable selection for rare-events data. We derived optimal probabilities based on the A- and L-

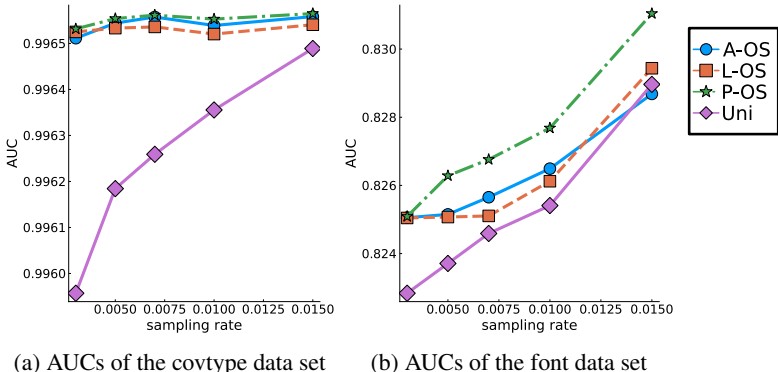

| (a) AUCs of the covtype data set | (b) AUCs of the font data set |

Figure 4: Empirical median AUCs for two real data sets

optimality criteria, and discussed their limitations. Furthermore, we proposed scale-invariant optimal probabilities based on prediction errors to overcome the limitations. Both analytical and numerical results show the desirable properties of the proposed methods.

Our investigation has the following limitations.

- Our proposed criterion optimizes the probabilities by minimizing the asymptotic mean squared error in estimating rare-event probabilities. While this prioritizes the accuracy of estimation, it puts less emphasis on the quality of variable selection. Further research is needed to devise optimal probabilities that focus on variable selection performance metrics.

- Our theoretical analysis is based on asymptotic properties, with optimal probabilities defined through the asymptotic normality. Although our results may hold for sufficiently sparse models, they may not generalize to cases where the model is dense or over-parameterized, because asymptotic normality may no longer be applicable. Therefore, an important direction for future research is to study the non-asymptotic properties of our estimators, such as prediction error bounds. Non-asymptotic behaviors are particularly of interest in high-dimensional regimes.

- We employ Lasso as the pilot estimator. However, other variable selection methodologies, such as sure independence screening, can also be considered. Exploring the impact of different pilot estimators on our method's performance represents another avenue for future investigations.

- We assume that the underlying full model is correctly specified and possesses a sparse structure. Our analysis does not account for model misspecification. Further research is required to address scenarios where the model is possibly misspecified or where the number of features vastly exceeds the number of observations.

## Acknowledgments and Disclosure of Funding

The authors are grateful to Professor Kun Chen for the insightful comments and suggestions on the development of the manuscript. Funding in direct support of this work: NEI grant R21EY035710, NSF grant 2105571, UConn CLAS Research Funding in Academic Themes, GPUs donated by NVIDIA.

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

# A    Appendix / supplemental material

In this appendix, we present the details of the proof, the practical algorithm and simuation settings in the paper. Details of mathematical proofs are provided in Section B. Details of the practical algorithm are provided in Section C. We present the details of simulation settings in Section D and in Section E, we give some additional simulation results.

# B    Details of mathematical proofs

In this section, we provide details of mathematical proofs.

## B.1    General assumptions in the main paper

We begin with some general assumptions used throughout this paper.

**Assumption 1.** *The first, second and third derivatives of $f(\boldsymbol{x}; \boldsymbol{\theta})$ and $e^{f(\boldsymbol{x};\boldsymbol{\beta})}f(\boldsymbol{x}; \boldsymbol{\beta})$ with respect to $\boldsymbol{\beta}$ are bouned by a square intergrable random variable $B(\boldsymbol{x})$.*

**Assumption 2.** *The matrix $\mathbb{E}\left\{\dot{g}^{\otimes 2}(\boldsymbol{x}; \boldsymbol{\theta})\right\}$ is finite and positive definite.*

**Assumption 3.** *The subsampling rate $\rho$ satisfies that $c_N = e^{\alpha_{\rm t}}/\rho \to c$, where $0 \le c < \infty$ is a constant.*

**Assumption 4.** *The integral $\mathbb{E}\left[\left\{\varphi(\boldsymbol{x}) + \varphi^{-1}(\boldsymbol{x})\right\}B^2(\boldsymbol{x})\right]$ is finite, where $B(\boldsymbol{x})$ is a square-integrable function that dominates the first, second, and third derivatives of $f(\boldsymbol{x}; \boldsymbol{\theta})$ and $e^{f(\boldsymbol{x};\boldsymbol{\beta})}f(\boldsymbol{x}; \boldsymbol{\beta})$ with respect to $\boldsymbol{\beta}$.*

**Assumption 5.** *The integral $\mathbb{E}\left\{e^{f(\boldsymbol{x};\boldsymbol{\beta})}\varphi^{-1}(\boldsymbol{x})B(\boldsymbol{x})\right\}$ is finite.*

These assumptions are the same assumptions used in [22]. Here, we remind some notations used in the main paper:

$$p(\boldsymbol{x}; \boldsymbol{\theta}) = \frac{e^{\alpha + f(\boldsymbol{x};\boldsymbol{\beta})}}{1 + e^{\alpha + f(\boldsymbol{x};\boldsymbol{\beta})}},$$

$$\phi(\boldsymbol{x}; \boldsymbol{\theta}) = p(\boldsymbol{x}; \boldsymbol{\theta})\left\{1 - p(\boldsymbol{x}; \boldsymbol{\theta})\right\},$$

$$\boldsymbol{M} = \mathbb{E}\left\{e^{f(\boldsymbol{x}_i;\boldsymbol{\beta}_{\rm t})}\dot{g}^{\otimes 2}(\boldsymbol{x}_i, \boldsymbol{\theta}_{\rm t})\right\},$$

and

$$\boldsymbol{\Lambda}_{\mathrm{mscl}} = \mathbb{E}\left[\frac{e^{f(\boldsymbol{x};\boldsymbol{\beta}_{\mathrm{t}})}\dot{g}^{\otimes 2}(\boldsymbol{x};\boldsymbol{\beta}_{\mathrm{t}})}{1 + c\varphi^{-1}(\boldsymbol{x})e^{f(\boldsymbol{x};\boldsymbol{\beta}_{\mathrm{t}})}}\right].$$

To ease the presentation in the following sections, we denote

$$\boldsymbol{M}_{\mathrm{w}(\mathcal{A})} = \mathbb{E}\left[\left\{1 + \frac{ce^{f(\boldsymbol{x};\boldsymbol{\beta}_{\mathrm{t}})}}{\varphi(\boldsymbol{x})}\right\}e^{f(\boldsymbol{x};\boldsymbol{\beta}_{\mathrm{t}})}\dot{g}_{(\mathcal{A})}^{\otimes 2}(\boldsymbol{x};\boldsymbol{\theta}_{\mathrm{t}})\right],$$

and $a_N = \sqrt{Ne^{\alpha_{\mathrm{t}}}}$ in the appendix. Note that

$$N_1 = \sum_{i=1}^{N} y_i = N\mathbb{E}\left\{\frac{e^{\alpha_{\mathrm{t}}+f(\boldsymbol{x};\boldsymbol{\beta}_{\mathrm{t}})}}{1 + e^{\alpha_{\mathrm{t}}+f(\boldsymbol{x};\boldsymbol{\beta}_{\mathrm{t}})}}\right\}\{1 + o_P(1)\}$$

$$= Ne^{\alpha_{\mathrm{t}}}\mathbb{E}\left\{e^{f(\boldsymbol{x};\boldsymbol{\beta}_{\mathrm{t}})}\right\}\{1 + o_P(1)\} = a_N^2\mathbb{E}\left\{e^{f(\boldsymbol{x};\boldsymbol{\beta}_{\mathrm{t}})}\right\}\{1 + o_P(1)\}.$$

## B.2  Proof of Theorem 1

*Proof of Theorem 1.* We consider the target of IPW adaptive lasso estimator:

$$Q_{\mathrm{w}}(\boldsymbol{\theta}) = -\sum_{i=1}^{N}\frac{\delta_i}{\pi(\boldsymbol{x}_i, y_i)}[y_i g(\boldsymbol{x}_i;\boldsymbol{\theta}) - \log\{1 + e^{g(\boldsymbol{x}_i;\boldsymbol{\theta})}\}] + \lambda_N\sum_{j=1}^{p}\hat{w}_j|\beta_{(j)}|$$

$$= -\ell_{\mathrm{w}}(\boldsymbol{\theta}) + \lambda_N\sum_{j=1}^{p}\hat{w}_j|\beta_{(j)}|,$$

where $\hat{w}_j = 1/|\hat{\beta}_{\mathrm{pl}(j)}|, 1 \leq j \leq p$. Then, we have that $\hat{\boldsymbol{u}}_N = a_N(\hat{\boldsymbol{\theta}}_{\mathrm{w}} - \boldsymbol{\theta}_{\mathrm{t}})$ is the minimizer of

$$\gamma_{\mathrm{w}}^{N}(\boldsymbol{u}) = Q_{\mathrm{w}}(\boldsymbol{\theta}_{\mathrm{t}} + a_N^{-1}\boldsymbol{u}) - Q_{\mathrm{w}}(\boldsymbol{\theta}_{\mathrm{t}}).$$

**Asymptotic normality:**  We prove the asymptotic normality part in this paragraph. By Taylor's expansion,

$$\gamma_{\mathrm{w}}^{N}(\boldsymbol{u}) = -\frac{1}{a_N}\boldsymbol{u}^{\mathrm{T}}\dot{\ell}_{\mathrm{w}}(\boldsymbol{\theta}_{\mathrm{t}}) + \frac{1}{2a_N^2}\sum_{i=1}^{N}\frac{\delta_i}{\pi(\boldsymbol{x}_i, y_i)}\phi(\boldsymbol{x}_i;\boldsymbol{\theta}_{\mathrm{t}})\{\boldsymbol{u}^{\mathrm{T}}\dot{g}(\boldsymbol{x}_i;\boldsymbol{\theta}_{\mathrm{t}})\}^2 - \Delta_{\mathrm{w}} + R_{\mathrm{w}}$$

$$+ \frac{\lambda_N}{a_N}\sum_{j=1}^{p}\hat{w}_j a_N\left(\left|\beta_{\mathrm{t}(j)} + \frac{u_{(j)}}{a_N}\right| - |\beta_{\mathrm{t}(j)}|\right).$$

We first consider the limit behavior of the IPW target function by prove the asymptotic normality. In [22], the authors established that under Assumptions 1 to 3,

$$a_N^{-1}\dot{\ell}_{\mathrm{w}}(\boldsymbol{\theta}_{\mathrm{t}}) \rightsquigarrow \boldsymbol{M}_{\mathrm{w}}^{1/2}\boldsymbol{Z},$$

$$\frac{1}{a_N^2}\sum_{i=1}^{N}\frac{\delta_i}{\pi(\boldsymbol{x}_i, y_i)}\phi(\boldsymbol{x}_i;\boldsymbol{\theta}_{\mathrm{t}})\dot{g}^{\otimes 2}(\boldsymbol{x}_i;\boldsymbol{\theta}_{\mathrm{t}}) \xrightarrow{P} \boldsymbol{M},$$

and

$$\Delta_{\mathrm{w}} = o_P(1), \quad R_{\mathrm{w}} = o_P(1).$$

Thus,

$$-\ell_{\mathrm{w}}(\boldsymbol{\theta}_{\mathrm{t}}) \rightsquigarrow -\boldsymbol{u}^{\mathrm{T}}\boldsymbol{M}_{\mathrm{w}}^{1/2}\boldsymbol{Z} + \frac{1}{2}\boldsymbol{u}^{\mathrm{T}}\boldsymbol{M}\boldsymbol{u}.$$

Next, we consider the limit behavior of the adaptive lasso penalty. Since we assume $\hat{\boldsymbol{\beta}}_{\mathrm{pl}}$ to be a consistent estimator, we know that when $j \in \mathcal{A}$, i.e., $\boldsymbol{\beta}_{\mathrm{t}(j)} \neq 0$,

$$\hat{w}_j = |\hat{\beta}_{\mathrm{pl}(j)}|^{-\gamma} \xrightarrow{P} |\beta_{\mathrm{t}(j)}|^{-\gamma} > 0,$$

and

$$a_N\left(\left|\beta_{\mathrm{t}(j)} + \frac{u_{(j)}}{a_N}\right| - |\beta_{\mathrm{t}(j)}|\right) \rightarrow \mathrm{sgn}(\beta_{\mathrm{t}(j)})u_{(j)}.$$

Therefore, for $j \in \mathcal{A}$, we have that

$$\frac{\lambda_N}{a_N}\hat{w}_j a_N \left(\left|\beta_{\mathrm{t}(j)} + \frac{u_{(j)}}{a_N}\right| - |\beta_{\mathrm{t}(j)}|\right) = o_P(1)$$

since $\lambda_N/a_N = \lambda_N/\sqrt{Ne^{\alpha_{\mathrm{t}}}} \to 0$. On the other hand, when $j \in \mathcal{A}^c$, i.e., $\beta_{\mathrm{t}(j)} = 0$, we have that for $u_{(j)} \neq 0$,

$$\frac{\lambda_N}{a_N}\hat{w}_j a_N \left(\left|\beta_{\mathrm{t}(j)} + \frac{u_{(j)}}{a_N}\right| - |\beta_{\mathrm{t}(j)}|\right) = \frac{\lambda_N}{a_N}\hat{w}_j|u_{(j)}| = \frac{\lambda_N}{a_N|\hat{\beta}_{\mathrm{pl}(j)}|^\gamma}|u_{(j)}| \xrightarrow{P} \infty,$$

since $\lambda_N/(\sqrt{Ne^{\alpha_{\mathrm{t}}}}|\hat{\beta}_{\mathrm{pl}(j)}|^\gamma) \xrightarrow{P} \infty$. Then, we have that $\gamma_{\mathrm{w}}^N(\boldsymbol{u}) \rightsquigarrow \gamma_{\mathrm{w}}(\boldsymbol{u})$, where

$$\gamma_{\mathrm{w}}(\boldsymbol{u}) = \begin{cases} \frac{1}{2}\boldsymbol{u}_{(\mathcal{A})}^{\mathrm{T}}\boldsymbol{M}_{(\mathcal{A})}\boldsymbol{u}_{(\mathcal{A})} - \boldsymbol{u}_{(\mathcal{A})}^{\mathrm{T}}\boldsymbol{M}_{\mathrm{w}}^{1/2}\boldsymbol{Z}_{(\mathcal{A})} & \text{if } u_{(j)} = 0, \forall j \in \mathcal{A}^c \\ \infty & \text{otherwise.} \end{cases}$$

Note that the unique minimizer of $\gamma_{\mathrm{w}}^N(\boldsymbol{u})$ is $(\boldsymbol{M}_{(\mathcal{A})}^{-1}\boldsymbol{M}_{\mathrm{w}}^{1/2}\boldsymbol{Z}_{(\mathcal{A})}^{\mathrm{T}}, \boldsymbol{0})^{\mathrm{T}}$ if we put all the indexes of active variables in front. Thus, following the results of [11] and [10], we have the minimizer of $\gamma_{\mathrm{w}}^N(\boldsymbol{u})$, i.e., $\hat{\boldsymbol{u}}_N$, satisfies that

$$\hat{\boldsymbol{u}}_{N(\mathcal{A})} \rightsquigarrow \boldsymbol{M}_{(\mathcal{A})}^{-1}\boldsymbol{M}_{\mathrm{w}}^{1/2}\boldsymbol{Z}_{(\mathcal{A})} \text{ and } \hat{\boldsymbol{u}}_{N(\mathcal{A}^c)} \rightsquigarrow \boldsymbol{0}.$$

Thus,

$$\hat{\boldsymbol{u}}_{N(\mathcal{A})} = a_N(\hat{\boldsymbol{\theta}}_{\mathrm{w}(\mathcal{A})} - \boldsymbol{\theta}_{\mathrm{t}(\mathcal{A})}) \rightsquigarrow \mathbb{N}(\boldsymbol{0}, \boldsymbol{M}_{(\mathcal{A})}^{-1}\boldsymbol{M}_{\mathrm{w}(\mathcal{A})}\boldsymbol{M}_{(\mathcal{A})}^{-1}).$$

Since

$$\sqrt{N_1} = a_N \mathbb{E}^{1/2}\left\{e^{f(\boldsymbol{x};\boldsymbol{\beta}_{\mathrm{t}})}\right\}\{1 + o_P(1)\},$$

applying Slusky's theorem, we have

$$\sqrt{N_1}\boldsymbol{V}_{\mathrm{w}(\mathcal{A})}^{-1/2}(\hat{\boldsymbol{\theta}}_{\mathrm{w}(\mathcal{A})} - \boldsymbol{\theta}_{\mathrm{t}(\mathcal{A})}) \rightsquigarrow \mathbb{N}(\boldsymbol{0}, \boldsymbol{I}).$$

**Consistency in variable selection** We prove the consistency in variable selection in this paragraph. From the result of asymptotic normality, we know that $\hat{\beta}_{\mathrm{w}(j)} \xrightarrow{P} \beta_{\mathrm{t}(j)}$ for every $j \in \mathcal{A}$ and therefore $\mathbb{P}(j \in \hat{\mathcal{A}}_{\mathrm{w}}) \to 1$. Thus, we only consider $j' \in \mathcal{A}^c$. When $j' \in \hat{\mathcal{A}}_{\mathrm{w}}$, we know that by K-K-T optimality conditions, we have

$$\lambda_N\hat{w}_{j'}\mathrm{sgn}(\hat{\beta}_{(j')}) = \dot{\ell}_{\mathrm{w}}(\hat{\boldsymbol{\theta}}_{\mathrm{w}}),$$

which means

$$\frac{\lambda_N\hat{w}_{j'}\mathrm{sgn}(\hat{\beta}_{(j')})}{a_N} = \frac{\dot{\ell}_{\mathrm{w}}(\hat{\boldsymbol{\theta}}_{\mathrm{w}})}{a_N}$$

$$= \frac{\dot{\ell}_{\mathrm{w}}(\boldsymbol{\theta}_{\mathrm{t}})}{a_N} + \frac{a_N\left\{\dot{\ell}_{\mathrm{w}}(\hat{\boldsymbol{\theta}}_{\mathrm{w}}) - \dot{\ell}_{\mathrm{w}}(\boldsymbol{\theta}_{\mathrm{t}})\right\}}{a_N^2} =: I_1 + I_2.$$

We have known that $I_1 = \dot{\ell}_{\mathrm{w}}(\boldsymbol{\theta}_{\mathrm{t}})/a_N \rightsquigarrow \boldsymbol{Z}_{\mathrm{w}}$. We now prove that proof that $I_2 = O_P(1)$. We apply Taylor expansion to the $k$-th element of $\dot{\ell}_{\mathrm{w}}(\hat{\boldsymbol{\theta}}_{\mathrm{w}})$ and have that

$$\frac{a_N\left\{\dot{\ell}_{(k)}(\hat{\boldsymbol{\theta}}_{\mathrm{w}}) - \dot{\ell}_{(k)}(\boldsymbol{\theta}_{\mathrm{t}})\right\}}{a_N^2} = -\frac{1}{a_N^2}\sum_{i=1}^N \frac{\delta_i}{\pi(\boldsymbol{x}, y_i)}\phi(\boldsymbol{x}_i; \boldsymbol{\theta}_{\mathrm{t}})\dot{g}_{(k)}(\boldsymbol{x}_i; \boldsymbol{\theta}_{\mathrm{t}})\dot{g}^{\mathrm{T}}(\boldsymbol{x}_i; \boldsymbol{\theta}_{\mathrm{t}})\hat{\boldsymbol{u}}_N + \tilde{\Delta}_{\mathrm{w}(k)} + \tilde{R}_{\mathrm{w}(k)},$$

where,

$$\hat{\boldsymbol{u}}_N = a_N(\hat{\boldsymbol{\theta}}_{\mathrm{w}} - \boldsymbol{\theta}_{\mathrm{t}}) = O_P(1),$$

$$\tilde{\Delta}_{\mathrm{w}(k)} = \frac{1}{a_N^2}\sum_{i=1}^N \frac{\delta_i}{\pi(\boldsymbol{x}_i, y_i)}\{y_i - p(\boldsymbol{x}_i; \boldsymbol{\theta}_{\mathrm{t}})\}\sum_{j=1}^d \ddot{g}_{(kj)}(\boldsymbol{x}_i; \boldsymbol{\theta}_{\mathrm{t}})\hat{u}_{N(j)},$$

and

$$\tilde{R}_{\mathrm{w}(k)} = -\frac{1}{2a_N^3} \sum_{i=1}^{N} \frac{\delta_i}{\pi(\boldsymbol{x}_i, y_i)} \phi(\boldsymbol{x}_i; \acute{\boldsymbol{\theta}}_k) \left\{ 1 - 2p(\boldsymbol{x}_i; \acute{\boldsymbol{\theta}}_k) \right\} \dot{g}_{(k)}(\boldsymbol{x}_i; \acute{\boldsymbol{\theta}}_k) \hat{\boldsymbol{u}}_N^{\mathrm{T}} \dot{g}^{\otimes 2}(\boldsymbol{x}_i; \acute{\boldsymbol{\theta}}_k) \hat{\boldsymbol{u}}_N$$

$$- \frac{2}{2a_N^3} \sum_{i=1}^{N} \frac{\delta_i}{\pi(\boldsymbol{x}_i, y_i)} \phi(\boldsymbol{x}_i; \acute{\boldsymbol{\theta}}_k) \left\{ \hat{\boldsymbol{u}}_N^{\mathrm{T}} \frac{\partial \dot{g}_{(k)}(\boldsymbol{x}_i; \acute{\boldsymbol{\theta}}_k)}{\partial \boldsymbol{\theta}} \right\} \left\{ \hat{\boldsymbol{u}}_N^{\mathrm{T}} \dot{g}(\boldsymbol{x}_i; \acute{\boldsymbol{\theta}}_k) \right\}$$

$$- \frac{1}{2a_N^3} \sum_{i=1}^{N} \frac{\delta_i}{\pi(\boldsymbol{x}_i, y_i)} \phi(\boldsymbol{x}_i; \acute{\boldsymbol{\theta}}_k) \dot{g}_{(k)}(\boldsymbol{x}_i; \acute{\boldsymbol{\theta}}_k) \left\{ \hat{\boldsymbol{u}}_N^{\mathrm{T}} \ddot{g}(\boldsymbol{x}_i; \acute{\boldsymbol{\theta}}_k) \hat{\boldsymbol{u}}_N \right\}$$

$$+ \frac{1}{2a_N^3} \sum_{i=1}^{N} \frac{\delta_i}{\pi(\boldsymbol{x}_i, y_i)} \left\{ y_i - p(\boldsymbol{x}_i; \acute{\boldsymbol{\theta}}_k) \right\} \hat{\boldsymbol{u}}_N^{\mathrm{T}} \frac{\partial^2 \dot{g}_{(k)}(\boldsymbol{x}_i; \acute{\boldsymbol{\theta}}_k)}{\partial \boldsymbol{\theta}^2} \hat{\boldsymbol{u}}_N.$$

where $\acute{\boldsymbol{\theta}}_k$ is between $\hat{\boldsymbol{\theta}}_{\mathrm{mle}}$ and $\boldsymbol{\theta}_{\mathrm{t}}$. First, we prove that $\tilde{R}_{(k)}$ is $o_P(1)$. We have that

$$|\tilde{R}_{\mathrm{w}(k)}| \leq \frac{\|\hat{\boldsymbol{u}}_N\|^2}{2a_N^3} \sum_{i=1}^{N} \frac{\delta_i}{\pi(\boldsymbol{x}_i, y_i)} \phi(\boldsymbol{x}_i; \acute{\boldsymbol{\theta}}_k) \left| 1 - 2p(\boldsymbol{x}_i; \acute{\boldsymbol{\theta}}_k) \right| \left| \dot{g}_{(k)}(\boldsymbol{x}_i; \acute{\boldsymbol{\theta}}_k) \right| \left\| \dot{g}(\boldsymbol{x}_i; \acute{\boldsymbol{\theta}}_k) \right\|^2$$

$$+ \frac{2\|\hat{\boldsymbol{u}}_N\|^2}{2a_N^3} \sum_{i=1}^{N} \frac{\delta_i}{\pi(\boldsymbol{x}_i, y_i)} \phi(\boldsymbol{x}_i; \acute{\boldsymbol{\theta}}_k) \left\| \frac{\partial \dot{g}_{(k)}(\boldsymbol{x}_i; \acute{\boldsymbol{\theta}}_k)}{\partial \boldsymbol{\theta}} \right\| \left\| \dot{g}(\boldsymbol{x}_i; \acute{\boldsymbol{\theta}}_k) \right\|$$

$$+ \frac{\|\hat{\boldsymbol{u}}_N\|^2}{2a_N^2} \sum_{i=1}^{N} \frac{\delta_i}{\pi(\boldsymbol{x}_i, y_i)} \phi(\boldsymbol{x}_i; \acute{\boldsymbol{\theta}}_k) \left| \dot{g}_{(k)}(\boldsymbol{x}_i; \acute{\boldsymbol{\theta}}_k) \right| \left\| \ddot{g}(\boldsymbol{x}_i; \acute{\boldsymbol{\theta}}_k) \right\|$$

$$+ \frac{\|\hat{\boldsymbol{u}}_N\|^2}{2a_N^3} \sum_{i=1}^{N} \frac{\delta_i}{\pi(\boldsymbol{x}_i, y_i)} p(\boldsymbol{x}_i; \acute{\boldsymbol{\theta}}_k) \left\| \frac{\partial^2 \dot{g}_{(k)}(\boldsymbol{x}_i; \acute{\boldsymbol{\theta}}_k)}{\partial \boldsymbol{\theta}^2} \right\|$$

$$+ \frac{\|\hat{\boldsymbol{u}}_N\|^2}{2a_N^3} \sum_{i=1}^{N} \frac{\delta_i}{\pi(\boldsymbol{x}_i, y_i)} y_i \left\| \frac{\partial^2 \dot{g}_{(k)}(\boldsymbol{x}_i; \acute{\boldsymbol{\theta}}_k)}{\partial \boldsymbol{\theta}^2} \right\|$$

$$\leq \frac{\|\hat{\boldsymbol{u}}_N\|^2}{2a_N^3} \sum_{i=1}^{N} \frac{\delta_i}{\pi(\boldsymbol{x}_i, y_i)} p(\boldsymbol{x}_i; \acute{\boldsymbol{\theta}}_k) C(\boldsymbol{x}_i; \acute{\boldsymbol{\theta}}_k) + \frac{\|\hat{\boldsymbol{u}}_N\|^2}{2a_N^3} \sum_{i=1}^{N} \frac{\delta_i}{\pi(\boldsymbol{x}_i, y_i)} y_i B(\boldsymbol{x}_i)$$

$$\leq \frac{\|\hat{\boldsymbol{u}}_N\|^2 e^{\acute{\alpha}_k - \alpha_{\mathrm{t}}} e^{\alpha_{\mathrm{t}}}}{2a_N^3} \sum_{i=1}^{N} \frac{\delta_i}{\pi(\boldsymbol{x}_i, y_i)} e^{f(\boldsymbol{x}_i; \acute{\boldsymbol{\beta}}_k)} C(\boldsymbol{x}_i; \acute{\boldsymbol{\theta}}_k) + \frac{\|\hat{\boldsymbol{u}}_N\|^2}{2a_N^3} \sum_{i=1}^{N} \frac{\delta_i}{\pi(\boldsymbol{x}_i, y_i)} y_i B(\boldsymbol{x}_i)$$

$$\leq \frac{\|\hat{\boldsymbol{u}}_N\|^2 e^{\acute{\alpha}_k - \alpha_{\mathrm{t}}}}{2N a_N} \sum_{i=1}^{N} \frac{\delta_i}{\pi(\boldsymbol{x}_i, y_i)} e^{f(\boldsymbol{x}_i; \acute{\boldsymbol{\beta}}_k)} C(\boldsymbol{x}_i; \acute{\boldsymbol{\theta}}_k) + \frac{\|\hat{\boldsymbol{u}}_N\|^2}{2a_N^3} \sum_{i=1}^{N} \frac{\delta_i}{\pi(\boldsymbol{x}_i, y_i)} y_i B(\boldsymbol{x}_i)$$

$$= o_P(1),$$

where

$$C(\boldsymbol{x}_i; \acute{\boldsymbol{\theta}}) = \left| \dot{g}_{(k)}(\boldsymbol{x}_i; \acute{\boldsymbol{\theta}}) \right| \left\{ \left\| \dot{g}(\boldsymbol{x}_i; \acute{\boldsymbol{\theta}}_k) \right\|^2 + \left\| \ddot{g}(\boldsymbol{x}_i; \acute{\boldsymbol{\theta}}) \right\| \right\}$$

$$+ \left\| \frac{\partial \dot{g}_{(k)}(\boldsymbol{x}_i; \acute{\boldsymbol{\theta}}_k)}{\partial \boldsymbol{\theta}} \right\| \left\| \dot{g}(\boldsymbol{x}_i; \acute{\boldsymbol{\theta}}_k) \right\| + \left\| \frac{\partial^2 \dot{g}_{(k)}(\boldsymbol{x}_i; \acute{\boldsymbol{\theta}}_k)}{\partial \boldsymbol{\theta}^2} \right\|.$$

Therefore, we proved that $\tilde{R}_{\mathrm{w}(k)} = o_P(1)$. Next, we prove that $\tilde{\Delta}_{\mathrm{w}(k)} = o_P(1)$. We know that $\mathbb{E}\left[ a_N^{-2} \sum_{i=1}^{N} \delta_i / \pi(\boldsymbol{x}_i, y_i) \left\{ y_i - p(\boldsymbol{x}_i; \boldsymbol{\theta}_{\mathrm{t}}) \right\} \ddot{g}(\boldsymbol{x}_i; \boldsymbol{\theta}_{\mathrm{t}}) \right] = \boldsymbol{0}$. We also have that for the every element of $a_N^{-2} \sum_{i=1}^{N} \delta_i / \pi(\boldsymbol{x}_i, y_i) \left\{ y_i - p(\boldsymbol{x}_i; \boldsymbol{\theta}_{\mathrm{t}}) \right\} \ddot{g}(\boldsymbol{x}_i; \boldsymbol{\theta}_{\mathrm{t}})$, we have

$$\mathbb{V}\left[ a_N^{-2} \sum_{i=1}^{N} \frac{\delta_i}{\pi(\boldsymbol{x}_i, y_i)} \left\{ y_i - p(\boldsymbol{x}_i; \boldsymbol{\theta}_{\mathrm{t}}) \right\} \ddot{g}_{(jl)}(\boldsymbol{x}_i; \boldsymbol{\theta}_{\mathrm{t}}) \right]$$

$$\leq \frac{1}{a_N^4} \sum_{i=1}^{N} \mathbb{E}\left\{p(\boldsymbol{x}_i; \boldsymbol{\theta}_t)\ddot{g}_{(jl)}^2(\boldsymbol{x}_i; \boldsymbol{\theta}_t)\right\} \leq \frac{1}{a_N^2}\mathbb{E}[e^{f(\boldsymbol{x};\boldsymbol{\beta}_t)}\|\ddot{g}(\boldsymbol{x}; \boldsymbol{\theta}_t)\|^2] \to 0.$$

Thus, due to Chebyshev's inequality, we know that $\tilde{\Delta}_w = o_P(1)$. Since we know that $\frac{1}{a_N^2}\sum_{i=1}^{N} \delta_i/\pi(\boldsymbol{x}_i, y_i)\phi(\boldsymbol{x}_i; \boldsymbol{\theta}_t)\dot{g}^{\otimes 2}(\boldsymbol{x}_i; \boldsymbol{\theta}_t) = H_w = O_P(1)$. Hence, we have that

$$\frac{\dot{\ell}_w(\hat{\boldsymbol{\theta}}_w)}{a_N} = O_P(1).$$

Note that we also have

$$\frac{\lambda_N \hat{w}_{j'}}{a_N} = \frac{\lambda_N}{a_N}\frac{1}{|\hat{\beta}_{\text{pl}(j')}|^\gamma} \xrightarrow{P} \infty.$$

Therefore,

$$\mathbb{P}(j' \in \hat{\mathcal{A}}_w) \leq \mathbb{P}\left\{\lambda_N \hat{w}_{j'}\text{sgn}(\hat{\beta}_{(j')}) = \dot{\ell}_w(\hat{\boldsymbol{\theta}}_w)\right\}$$

$$= \mathbb{P}\left\{\frac{\lambda_N \hat{w}_{j'}\text{sgn}(\hat{\beta}_{(j')})}{a_N} = \frac{\dot{\ell}_w(\hat{\boldsymbol{\theta}}_w)}{a_N}\right\} \to 0.$$

Thus, we prove the part of consistency of variable selection. $\square$

### B.3   Proof of Proposition 1

We first give a lemma for general optimal functions.

**Lemma 1.** *Assume that $h(\boldsymbol{x})^2$ and $\varphi(\boldsymbol{x})$ are integrable function with $\mathbb{E}\{\varphi(\boldsymbol{x})\} = 1$. The optimal function $\varphi^{**}(\boldsymbol{x})$ that minimize the value $\mathbb{E}\left\{\frac{h^2(\boldsymbol{x})}{\varphi(\boldsymbol{x})}\right\}$ is given as $\varphi^{**}(\boldsymbol{x}) = \frac{h(\boldsymbol{x})}{\mathbb{E}\{h(\boldsymbol{x})\}}$.*

*Proof.* Appying Cauchy-Schwartz inequality, we have that

$$\mathbb{E}\{h(\boldsymbol{x})\}^2 = \mathbb{E}\left\{\frac{h(\boldsymbol{x})}{\sqrt{\varphi(\boldsymbol{x})}}\sqrt{\varphi(\boldsymbol{x})}\right\}^2 \leq \mathbb{E}\left\{\frac{h^2(\boldsymbol{x})}{\varphi(\boldsymbol{x})}\right\}\mathbb{E}\{\varphi(\boldsymbol{x})\} = \mathbb{E}\left\{\frac{h^2(\boldsymbol{x})}{\varphi(\boldsymbol{x})}\right\}.$$

Therefore, we have that $\mathbb{E}\left\{\frac{h^2(\boldsymbol{x})}{\varphi(\boldsymbol{x})}\right\} \geq \mathbb{E}\{h(\boldsymbol{x})\}^2$ and the equality holds if and only if $\sqrt{\varphi(\boldsymbol{x})} = Kh(\boldsymbol{x})/\sqrt{\varphi(\boldsymbol{x})}$, where $K$ is a constant. Therefore, $\varphi^{**}(\boldsymbol{x}) = Kh(\boldsymbol{x})$, and since $\mathbb{E}\{\varphi^{**}(\boldsymbol{x})\} = 1$, we know that $\varphi^{**}(\boldsymbol{x}) = h(\boldsymbol{x})/\mathbb{E}\{h(\boldsymbol{x})\}$. $\square$

Now, we prove Proposition 1.

*Proof.* We first calculate the optimal function that minimizes $\text{tr}(\boldsymbol{V}_{w(\mathcal{A})})$. We have that

$$\text{tr}(\boldsymbol{V}_{w(\mathcal{A})}) = \text{tr}\left\{\boldsymbol{M}_{(\mathcal{A})}^{-1}\boldsymbol{M}_{w(\mathcal{A})}\boldsymbol{M}_{(\mathcal{A})}^{-1}\right\}$$

$$= \text{tr}\left\{\boldsymbol{M}_{(\mathcal{A})}^{-1}\mathbb{E}\left[\left\{1 + \frac{ce^{f(\boldsymbol{x};\boldsymbol{\beta}_t)}}{\varphi(\boldsymbol{x})}\right\}e^{f(\boldsymbol{x};\boldsymbol{\beta}_t)}\dot{g}_{(\mathcal{A})}^{\otimes 2}(\boldsymbol{x}; \boldsymbol{\theta}_t)\right]\boldsymbol{M}_{(\mathcal{A})}^{-1}\right\}.$$

We focus on the values that related to $\varphi(\boldsymbol{x})$. We know that

$$\mathbb{E}\left\{\varphi^{-1}(\boldsymbol{x})e^{2f(\boldsymbol{x};\boldsymbol{\beta}_t)}\dot{g}_{(\mathcal{A})}^{\otimes 2}(\boldsymbol{x}; \boldsymbol{\theta}_t)\right\} = e^{-2\alpha_t}\{1 + o_P(1)\}\mathbb{E}\left\{\varphi^{-1}(\boldsymbol{x})p^2(\boldsymbol{x}; \boldsymbol{\theta}_t)\dot{g}_{(\mathcal{A})}^{\otimes 2}(\boldsymbol{x}; \boldsymbol{\theta}_t)\right\}.$$

Therefore, we need to minimize

$$\text{tr}\left[\mathbb{E}\left\{\frac{p^2(\boldsymbol{x}; \boldsymbol{\theta}_t)\boldsymbol{M}_{(\mathcal{A})}^{-1}\dot{g}_{(\mathcal{A})}^{\otimes 2}(\boldsymbol{x}; \boldsymbol{\theta}_t)\boldsymbol{M}_{(\mathcal{A})}^{-1}}{\varphi(\boldsymbol{x})}\right\}\right]$$

$$= \mathbb{E}\left[\text{tr}\left\{\frac{p^2(\boldsymbol{x}; \boldsymbol{\theta}_t)\boldsymbol{M}_{(\mathcal{A})}^{-1}\dot{g}_{(\mathcal{A})}^{\otimes 2}(\boldsymbol{x}; \boldsymbol{\theta}_t)\boldsymbol{M}_{(\mathcal{A})}^{-1}}{\varphi(\boldsymbol{x})}\right\}\right] = \mathbb{E}\left[\frac{p^2(\boldsymbol{x}; \boldsymbol{\theta}_t)\|\boldsymbol{M}_{(\mathcal{A})}^{-1}\dot{g}_{(\mathcal{A})}(\boldsymbol{x}; \boldsymbol{\theta}_t)\|^2}{\varphi(\boldsymbol{x})}\right].$$

Appying Lemma 1. We know that the minimizer is given as

$$\varphi_{\mathrm{A-OS}}(\boldsymbol{x}) = \frac{p(\boldsymbol{x};\boldsymbol{\theta}_{\mathrm{t}})\|M_{(\mathcal{A})}^{-1}\dot{g}_{(\mathcal{A})}(\boldsymbol{x};\boldsymbol{\theta}_{\mathrm{t}})\|}{\mathbb{E}\left\{p(\boldsymbol{x};\boldsymbol{\theta}_{\mathrm{t}})\|M_{(\mathcal{A})}^{-1}\dot{g}_{(\mathcal{A})}(\boldsymbol{x};\boldsymbol{\theta}_{\mathrm{t}})\|\right\}}.$$

Next, we calculate the optimal function that minimize $\mathrm{tr}(M_{\mathrm{w}(\mathcal{A})})$. We have that

$$\mathrm{tr}(M_{\mathrm{w}(\mathcal{A})}) = \mathrm{tr}\left\{\mathbb{E}\left[\left\{1 + \frac{ce^{f(\boldsymbol{x};\boldsymbol{\beta}_{\mathrm{t}})}}{\varphi(\boldsymbol{x})}\right\}e^{f(\boldsymbol{x};\boldsymbol{\beta}_{\mathrm{t}})}\dot{g}_{(\mathcal{A})}^{\otimes 2}(\boldsymbol{x};\boldsymbol{\theta}_{\mathrm{t}})\right]\right\}.$$

Therefore, we need to minimize

$$\mathrm{tr}\left[\mathbb{E}\left\{\frac{p^2(\boldsymbol{x};\boldsymbol{\theta}_{\mathrm{t}})\dot{g}_{(\mathcal{A})}^{\otimes 2}(\boldsymbol{x};\boldsymbol{\theta}_{\mathrm{t}})}{\varphi(\boldsymbol{x})}\right\}\right] = \mathbb{E}\left[\frac{p^2(\boldsymbol{x};\boldsymbol{\theta}_{\mathrm{t}})\|\dot{g}_{(\mathcal{A})}(\boldsymbol{x};\boldsymbol{\theta}_{\mathrm{t}})\|^2}{\varphi(\boldsymbol{x})}\right].$$

Appying Lemma 1. We know that the minimizer is given as

$$\varphi_{\mathrm{L-OS}}(\boldsymbol{x}) = \frac{p(\boldsymbol{x};\boldsymbol{\theta}_{\mathrm{t}})\|\dot{g}_{(\mathcal{A})}(\boldsymbol{x};\boldsymbol{\theta}_{\mathrm{t}})\|}{\mathbb{E}\left\{p(\boldsymbol{x};\boldsymbol{\theta}_{\mathrm{t}})\|\dot{g}_{(\mathcal{A})}(\boldsymbol{x};\boldsymbol{\theta}_{\mathrm{t}})\|\right\}}.$$

$\square$

## B.4 Proof of Theorem 2

*Proof.* In the proof of Thereom 1, we know that

$$a_N(\hat{\boldsymbol{\theta}}_{\mathrm{w}(\mathcal{A})}^{\mathrm{adp}} - \boldsymbol{\theta}_{\mathrm{t}(\mathcal{A})}) \rightsquigarrow M_{(\mathcal{A})}^{-1}M_{\mathrm{w}(\mathcal{A})}^{1/2}Z_{(\mathcal{A})}.$$

To simplify the representation, We define a function

$$h(\boldsymbol{\theta}) = e^{-2\alpha_{\mathrm{t}}}\mathrm{MSPE}(\boldsymbol{\theta}) = e^{-2\alpha_{\mathrm{t}}}\mathbb{E}\left[\{p(\boldsymbol{x};\boldsymbol{\theta}) - p(\boldsymbol{x};\boldsymbol{\theta}_{\mathrm{t}})\}^2\right].$$

We have that

$$\frac{\partial\{p(\boldsymbol{x};\boldsymbol{\theta}) - p(\boldsymbol{x};\boldsymbol{\theta}_{\mathrm{t}})\}^2}{\partial\boldsymbol{\theta}} = 2\{p(\boldsymbol{x};\boldsymbol{\theta}) - p(\boldsymbol{x};\boldsymbol{\theta}_{\mathrm{t}})\}\phi(\boldsymbol{x};\boldsymbol{\theta})\dot{g}(\boldsymbol{x};\boldsymbol{\theta}),$$

and

$$\frac{\partial^2\{p(\boldsymbol{x};\boldsymbol{\theta}) - p(\boldsymbol{x};\boldsymbol{\theta}_{\mathrm{t}})\}^2}{\partial\boldsymbol{\theta}\partial\boldsymbol{\theta}^{\mathrm{T}}} = 2\phi^2(\boldsymbol{x};\boldsymbol{\theta})\dot{g}^{\otimes 2}(\boldsymbol{x};\boldsymbol{\theta}) + 2\{p(\boldsymbol{x};\boldsymbol{\theta}) - p(\boldsymbol{x};\boldsymbol{\theta}_{\mathrm{t}})\}\frac{\partial\phi(\boldsymbol{x};\boldsymbol{\theta})}{\partial\boldsymbol{\theta}}\dot{g}(\boldsymbol{x};\boldsymbol{\theta})$$
$$+ 2\{p(\boldsymbol{x};\boldsymbol{\theta}) - p(\boldsymbol{x};\boldsymbol{\theta}_{\mathrm{t}})\}\phi(\boldsymbol{x};\boldsymbol{\theta})\ddot{g}(\boldsymbol{x};\boldsymbol{\theta}).$$

Note that $|p(\boldsymbol{x};\boldsymbol{\theta}) - p(\boldsymbol{x};\boldsymbol{\theta}_{\mathrm{t}})| \le 2$ and thus,

$$\left|\frac{\partial\{p(\boldsymbol{x};\boldsymbol{\theta}) - p(\boldsymbol{x};\boldsymbol{\theta}_{\mathrm{t}})\}^2}{\partial\boldsymbol{\theta}}\right| \le 2\left|p(\boldsymbol{x};\boldsymbol{\theta}) - p(\boldsymbol{x};\boldsymbol{\theta}_{\mathrm{t}})\right|\left|\phi(\boldsymbol{x};\boldsymbol{\theta})\dot{g}(\boldsymbol{x};\boldsymbol{\theta})\right| \le 4B(\boldsymbol{x}),$$

and

$$\left|\frac{\partial^2\{p(\boldsymbol{x};\boldsymbol{\theta}) - p(\boldsymbol{x};\boldsymbol{\theta}_{\mathrm{t}})\}^2}{\partial\boldsymbol{\theta}\partial\boldsymbol{\theta}^{\mathrm{T}}}\right|$$
$$\le 2\left|\phi^2(\boldsymbol{x};\boldsymbol{\theta})\dot{g}^{\otimes 2}(\boldsymbol{x};\boldsymbol{\theta})\right| + 2\left|\{p(\boldsymbol{x};\boldsymbol{\theta}) - p(\boldsymbol{x};\boldsymbol{\theta}_{\mathrm{t}})\}\right|\left|\frac{\partial\phi(\boldsymbol{x};\boldsymbol{\theta})}{\partial\boldsymbol{t}}\dot{g}(\boldsymbol{x};\boldsymbol{\theta})\right|$$
$$+ 2\left|\{p(\boldsymbol{x};\boldsymbol{\theta}) - p(\boldsymbol{x};\boldsymbol{\theta}_{\mathrm{t}})\}\right|\left|\phi(\boldsymbol{x};\boldsymbol{\theta})\ddot{g}(\boldsymbol{x};\boldsymbol{\theta})\right| \le 10B(\boldsymbol{x}).$$

Hence, due to donimating convergence theorem, we know that the expectation and derivitive are exchangable. Thus, we have that

$$\frac{\partial h(\boldsymbol{\theta})}{\partial\boldsymbol{\theta}} = e^{-2\alpha_{\mathrm{t}}}\mathbb{E}\left[2\{p(\boldsymbol{x};\boldsymbol{\theta}) - p(\boldsymbol{x};\boldsymbol{\theta}_{\mathrm{t}})\}\phi(\boldsymbol{x};\boldsymbol{\theta})\dot{g}(\boldsymbol{x};\boldsymbol{\theta})\right],$$

and

$$\frac{\partial h(\boldsymbol{\theta})}{\partial \boldsymbol{\theta} \partial \boldsymbol{\theta}^{\mathrm{T}}} = e^{-2\alpha_{\mathrm{t}}} \mathbb{E} \Big[ 2\phi^2(\boldsymbol{x};\boldsymbol{\theta})\dot{g}^{\otimes 2}(\boldsymbol{x};\boldsymbol{\theta}) + 2\left\{ p(\boldsymbol{x};\boldsymbol{\theta}) - p(\boldsymbol{x};\boldsymbol{\theta}_{\mathrm{t}}) \right\} \frac{\partial \phi(\boldsymbol{x};\boldsymbol{\theta})}{\partial \boldsymbol{\theta}} \dot{g}(\boldsymbol{x};\boldsymbol{\theta})$$
$$+ 2\left\{ p(\boldsymbol{x};\boldsymbol{\theta}) - p(\boldsymbol{x};\boldsymbol{\theta}_{\mathrm{t}}) \right\} \phi(\boldsymbol{x};\boldsymbol{\theta})\ddot{g}(\boldsymbol{x};\boldsymbol{\theta}) \Big].$$

Due to donimating convergence theorem, we also know that the first and second derivitive of $h(\boldsymbol{\theta})$ are continous. We have that

$$\frac{\partial h(\boldsymbol{\theta})}{\partial \boldsymbol{\theta}}\Big|_{\boldsymbol{\theta}=\boldsymbol{\theta}_{\mathrm{t}}} = \boldsymbol{0}, \; \frac{\partial^2 h(\boldsymbol{t})}{\partial \boldsymbol{\theta} \partial \boldsymbol{\theta}^{\mathrm{T}}}\Big|_{\boldsymbol{\theta}=\boldsymbol{\theta}_{\mathrm{t}}} = 2e^{-2\alpha_{\mathrm{t}}} \mathbb{E}\left[ \phi^2(\boldsymbol{x};\boldsymbol{\theta}_{\mathrm{t}})\dot{g}^{\otimes 2}(\boldsymbol{x};\boldsymbol{\theta}_{\mathrm{t}}) \right]$$

Note that $e^{-2\alpha_{\mathrm{t}}} \mathbb{E}\left[ \phi^2(\boldsymbol{x};\boldsymbol{\theta}_{\mathrm{t}})\dot{g}^{\otimes 2}(\boldsymbol{x};\boldsymbol{\theta}_{\mathrm{t}}) \right] \to \boldsymbol{\Omega}$ due to donimating covergence theorem. Now applying Theorem 1.12(ii) in [18], we have that

$$a_N^2 \left\{ h(\hat{\boldsymbol{\theta}}_{\mathrm{w}(\mathcal{A})}^{\mathrm{adp}}) - h(\boldsymbol{\theta}_{\mathrm{t}(\mathcal{A})}) \right\} = a_N^2 e^{-2\alpha_{\mathrm{t}}} \mathbb{E}\left[ \left\{ p(\boldsymbol{x};\hat{\boldsymbol{\theta}}_{\mathrm{w}(\mathcal{A})}) - p(\boldsymbol{x};\boldsymbol{\theta}_{\mathrm{t}(\mathcal{A})}) \right\}^2 \right]$$
$$\rightsquigarrow \frac{1}{2!} 2 \boldsymbol{Z}_{(\mathcal{A})}^{\mathrm{T}} \boldsymbol{M}_{\mathrm{w}(\mathcal{A})}^{1/2} \boldsymbol{M}_{(\mathcal{A})}^{-1} \boldsymbol{\Omega}_{(\mathcal{A})} \boldsymbol{M}_{(\mathcal{A})}^{-1} \boldsymbol{M}_{\mathrm{w}(\mathcal{A})}^{1/2} \boldsymbol{Z}_{(\mathcal{A})}$$
$$= \boldsymbol{Z}_{(\mathcal{A})}^{\mathrm{T}} \boldsymbol{M}_{\mathrm{w}(\mathcal{A})}^{1/2} \boldsymbol{M}_{(\mathcal{A})}^{-1} \boldsymbol{\Omega}_{(\mathcal{A})} \boldsymbol{M}_{(\mathcal{A})}^{-1} \boldsymbol{M}_{\mathrm{w}(\mathcal{A})}^{1/2} \boldsymbol{Z}_{(\mathcal{A})}.$$

Considering $N_1 = a_N^2 \mathbb{E}\left\{ e^{f(\boldsymbol{x};\boldsymbol{\beta}_{\mathrm{t}})} \right\} \{ 1 + o_P(1) \}$, applying Slutsky's theorem, we have that

$$N_1 e^{-2\alpha_{\mathrm{t}}} \mathbb{E}\left[ \left\{ p(\boldsymbol{x};\hat{\boldsymbol{\theta}}_{\mathrm{w}(\mathcal{A})}) - p(\boldsymbol{x};\boldsymbol{\theta}_{\mathrm{t}(\mathcal{A})}) \right\}^2 \right]$$
$$\rightsquigarrow \mathbb{E}^{-1}\left\{ e^{f(\boldsymbol{x};\boldsymbol{\beta}_{\mathrm{t}})} \right\} \boldsymbol{Z}_{(\mathcal{A})}^{\mathrm{T}} \boldsymbol{M}_{\mathrm{w}(\mathcal{A})}^{1/2} \boldsymbol{M}_{(\mathcal{A})}^{-1} \boldsymbol{\Omega}_{(\mathcal{A})} \boldsymbol{M}_{(\mathcal{A})}^{-1} \boldsymbol{M}_{\mathrm{w}(\mathcal{A})}^{1/2} \boldsymbol{Z}_{(\mathcal{A})}.$$

Since $\boldsymbol{Z}_{\mathrm{w}(\mathcal{A})} = \mathbb{N}(\boldsymbol{0}, \boldsymbol{I})$, we have

$$\mathbb{E}\left\{ \boldsymbol{Z}_{(\mathcal{A})}^{\mathrm{T}} \boldsymbol{M}_{\mathrm{w}(\mathcal{A})}^{1/2} \boldsymbol{M}_{(\mathcal{A})}^{-1} \boldsymbol{\Omega}_{(\mathcal{A})} \boldsymbol{M}_{(\mathcal{A})}^{-1} \boldsymbol{M}_{\mathrm{w}(\mathcal{A})}^{1/2} \boldsymbol{Z}_{(\mathcal{A})} \right\} = \mathrm{tr}\left\{ \boldsymbol{M}_{(\mathcal{A})}^{-1} \boldsymbol{\Omega}_{(\mathcal{A})} \boldsymbol{M}_{(\mathcal{A})}^{-1} \boldsymbol{M}_{\mathrm{w}(\mathcal{A})} \right\}$$
$$= \mathrm{tr}\left\{ \boldsymbol{M}_{(\mathcal{A})}^{-1} \boldsymbol{\Omega}_{(\mathcal{A})} \boldsymbol{M}_{(\mathcal{A})}^{-1} \mathbb{E}\left[ \left\{ 1 + \frac{c e^{f(\boldsymbol{x};\boldsymbol{\beta}_{\mathrm{t}})}}{\varphi(\boldsymbol{x})} \right\} e^{f(\boldsymbol{x};\boldsymbol{\beta}_{\mathrm{t}})} \dot{g}_{(\mathcal{A})}^{\otimes 2}(\boldsymbol{x};\boldsymbol{\theta}_{\mathrm{t}}) \right] . \right\}$$

We focus on the values that related to $\varphi(\boldsymbol{x})$. We know that

$$\mathbb{E}\left\{ \varphi^{-1}(\boldsymbol{x}) e^{2f(\boldsymbol{x};\boldsymbol{\beta}_{\mathrm{t}})} \dot{g}_{(\mathcal{A})}^{\otimes 2}(\boldsymbol{x};\boldsymbol{\theta}_{\mathrm{t}}) \right\} = e^{-2\alpha_{\mathrm{t}}} \{ 1 + o_P(1) \} \mathbb{E}\left\{ \varphi^{-1}(\boldsymbol{x}) p^2(\boldsymbol{x};\boldsymbol{\theta}_{\mathrm{t}}) \dot{g}_{(\mathcal{A})}^{\otimes 2}(\boldsymbol{x};\boldsymbol{\theta}_{\mathrm{t}}) \right\}.$$

Therefore, we need to minimize

$$\mathrm{tr}\left[ \mathbb{E}\left\{ \boldsymbol{M}_{(\mathcal{A})}^{-1} \boldsymbol{\Omega}_{(\mathcal{A})} \boldsymbol{M}_{(\mathcal{A})}^{-1} \frac{p^2(\boldsymbol{x};\boldsymbol{\theta}_{\mathrm{t}}) \dot{g}_{(\mathcal{A})}^{\otimes 2}(\boldsymbol{x};\boldsymbol{\theta}_{\mathrm{t}})}{\varphi(\boldsymbol{x})} \right\} \right]$$
$$= \mathbb{E}\left[ \mathrm{tr}\left\{ \boldsymbol{M}_{(\mathcal{A})}^{-1} \boldsymbol{\Omega}_{(\mathcal{A})} \boldsymbol{M}_{(\mathcal{A})}^{-1} \frac{p^2(\boldsymbol{x};\boldsymbol{\theta}_{\mathrm{t}}) \dot{g}_{(\mathcal{A})}^{\otimes 2}(\boldsymbol{x};\boldsymbol{\theta}_{\mathrm{t}})}{\varphi(\boldsymbol{x})} \right\} \right]$$
$$= \mathbb{E}\left[ \mathrm{tr}\left\{ \frac{p^2(\boldsymbol{x};\boldsymbol{\theta}_{\mathrm{t}}) \boldsymbol{\Omega}_{(\mathcal{A})}^{1/2} \boldsymbol{M}_{(\mathcal{A})}^{-1} \dot{g}_{(\mathcal{A})}^{\otimes 2}(\boldsymbol{x};\boldsymbol{\theta}_{\mathrm{t}}) \boldsymbol{M}_{(\mathcal{A})}^{-1} \boldsymbol{\Omega}_{(\mathcal{A})}^{1/2}}{\varphi(\boldsymbol{x})} \right\} \right]$$
$$= \mathbb{E}\left[ \frac{p^2(\boldsymbol{x};\boldsymbol{\theta}_{\mathrm{t}}) \| \boldsymbol{\Omega}_{(\mathcal{A})}^{1/2} \boldsymbol{M}_{(\mathcal{A})}^{-1} \dot{g}_{(\mathcal{A})}(\boldsymbol{x};\boldsymbol{\theta}_{\mathrm{t}}) \|^2}{\varphi(\boldsymbol{x})} \right].$$

Appying Lemma 1. We know that the minimizer is given as

$$\varphi_{\mathrm{P-OS}}(\boldsymbol{x}) = \frac{p(\boldsymbol{x};\boldsymbol{\theta}_{\mathrm{t}}) \| \boldsymbol{\Omega}_{(\mathcal{A})}^{1/2} \boldsymbol{M}_{(\mathcal{A})}^{-1} \dot{g}_{(\mathcal{A})}(\boldsymbol{x};\boldsymbol{\theta}_{\mathrm{t}}) \|}{\mathbb{E}\left\{ p(\boldsymbol{x};\boldsymbol{\theta}_{\mathrm{t}}) \| \boldsymbol{\Omega}_{(\mathcal{A})}^{1/2} \boldsymbol{M}_{(\mathcal{A})}^{-1} \dot{g}_{(\mathcal{A})}(\boldsymbol{x};\boldsymbol{\theta}_{\mathrm{t}}) \| \right\}}.$$

$\qquad\qquad\qquad\qquad\qquad\qquad\qquad\qquad\qquad\qquad\qquad\qquad\qquad\qquad\qquad\qquad\qquad\square$

### B.5 Proof of Proposition 2

*Proof.* First, we know that $g(\boldsymbol{x}; \boldsymbol{\theta}) = g(\boldsymbol{A}\boldsymbol{x}; \boldsymbol{B}^{\mathrm{T}}\boldsymbol{\theta})$. Since the equation holds for all $\boldsymbol{x}$ and $\boldsymbol{\theta}$, if we take derivitive with respect to $\boldsymbol{\theta}$ on both sides, the equation still holds. Thus, we have that

$$\dot{g}(\boldsymbol{x}; \boldsymbol{\theta}) = \boldsymbol{B}\dot{g}(\boldsymbol{A}\boldsymbol{x}; \boldsymbol{B}^{\mathrm{T}}\boldsymbol{\theta}).$$

If we scale the whole covariate variable $\boldsymbol{x}$ to $\tilde{\boldsymbol{x}} = \boldsymbol{A}\boldsymbol{x}$, we need to reparameterize $\boldsymbol{\theta}_{\mathrm{t}}$ to $\tilde{\boldsymbol{\theta}}_{\mathrm{t}} = \boldsymbol{B}^{\mathrm{T}}\boldsymbol{\theta}_{\mathrm{t}}$ to remain the problem invariant. We have that for $\tilde{\boldsymbol{x}}$ and $\tilde{\boldsymbol{\theta}}_{\mathrm{t}}$

$$\dot{g}(\tilde{\boldsymbol{x}}; \tilde{\boldsymbol{\theta}}_{\mathrm{t}}) = \dot{g}(\boldsymbol{A}\boldsymbol{x}; \boldsymbol{B}^{\mathrm{T}}\boldsymbol{\theta}_{\mathrm{t}}) = \boldsymbol{B}^{-1}\dot{g}(\boldsymbol{x}; \boldsymbol{\theta}_{\mathrm{t}}).$$

Now, we know that

$$\tilde{\boldsymbol{M}} = \mathbb{E}\{e^{f(\tilde{\boldsymbol{x}}; \tilde{\boldsymbol{\beta}}_{\mathrm{t}})}\dot{g}^{\otimes 2}(\tilde{\boldsymbol{x}}; \tilde{\boldsymbol{\theta}}_{\mathrm{t}})\} = \boldsymbol{B}^{-1}\mathbb{E}\{e^{f(\boldsymbol{x}; \boldsymbol{\beta}_{\mathrm{t}})}\dot{g}^{\otimes 2}(\boldsymbol{x}; \boldsymbol{\theta}_{\mathrm{t}})\}(\boldsymbol{B}^{\mathrm{T}})^{-1} = \boldsymbol{B}^{-1}\boldsymbol{M}(\boldsymbol{B}^{\mathrm{T}})^{-1},$$

and

$$\tilde{\boldsymbol{\Omega}} = \mathbb{E}\{e^{2f(\tilde{\boldsymbol{x}}; \tilde{\boldsymbol{\beta}}_{\mathrm{t}})}\dot{g}^{\otimes 2}(\tilde{\boldsymbol{x}}; \tilde{\boldsymbol{\theta}}_{\mathrm{t}})\} = \boldsymbol{B}^{-1}\mathbb{E}\{e^{2f(\boldsymbol{x}; \boldsymbol{\theta}_{\mathrm{t}})}\dot{g}^{\otimes 2}(\boldsymbol{x}; \boldsymbol{\theta}_{\mathrm{t}})\}(\boldsymbol{B}^{\mathrm{T}})^{-1} = \boldsymbol{B}^{-1}\boldsymbol{\Omega}(\boldsymbol{B}^{\mathrm{T}})^{-1}.$$

Thus, we have that

$$\begin{aligned}
\|\tilde{\boldsymbol{\Omega}}^{1/2}\tilde{\boldsymbol{M}}^{-1}\dot{g}(\tilde{\boldsymbol{x}}; \tilde{\boldsymbol{\theta}}_{\mathrm{t}})\|^2 &= \dot{g}^{\mathrm{T}}(\tilde{\boldsymbol{x}}; \tilde{\boldsymbol{\theta}}_{\mathrm{t}})\tilde{\boldsymbol{M}}^{-1}\tilde{\boldsymbol{\Omega}}\tilde{\boldsymbol{M}}^{-1}\dot{g}(\tilde{\boldsymbol{x}}; \tilde{\boldsymbol{\theta}}_{\mathrm{t}}) \\
&= \dot{g}^{\mathrm{T}}(\boldsymbol{x}; \boldsymbol{\theta}_{\mathrm{t}})(\boldsymbol{B}^{-1})^{\mathrm{T}}(\boldsymbol{B}^{\mathrm{T}})\boldsymbol{M}^{-1}\boldsymbol{B}\boldsymbol{B}^{-1}\boldsymbol{\Omega}(\boldsymbol{B}^{\mathrm{T}})^{-1}\boldsymbol{B}^{\mathrm{T}}\boldsymbol{M}^{-1}\boldsymbol{B}\boldsymbol{B}^{-1}\dot{g}(\boldsymbol{x}; \boldsymbol{\theta}_{\mathrm{t}}) \\
&= \dot{g}^{\mathrm{T}}(\boldsymbol{x}; \boldsymbol{\theta}_{\mathrm{t}})\boldsymbol{M}^{-1}\boldsymbol{\Omega}\boldsymbol{M}^{-1}\dot{g}(\boldsymbol{x}; \boldsymbol{\theta}_{\mathrm{t}}) = \|\boldsymbol{\Omega}^{1/2}\boldsymbol{M}^{-1}\dot{g}(\boldsymbol{x}; \boldsymbol{\theta}_{\mathrm{t}})\|^2.
\end{aligned}$$

Therefore, the leveraging term is invariant. For the probability term, we know that is only related to value $g(\boldsymbol{x}; \boldsymbol{\theta}_{\mathrm{t}}) = g(\tilde{\boldsymbol{x}}; \tilde{\boldsymbol{\theta}}_{\mathrm{t}})$, we know that it does not change after scaling inactive variables. This complete the proof. $\square$

### B.6 Proof of Theorem 3

*Proof of Theorem 3.* Consider maximum sampled conditional likeihood function with adaptive lasso penalty:

$$\begin{aligned}
Q_{\mathrm{mscl}}^{\hat{\boldsymbol{\theta}}_{\mathrm{pl}}}(\boldsymbol{\theta}) &= -\sum_{i=1}^{N}\delta_i^{\hat{\boldsymbol{\theta}}_{\mathrm{pl}}}[y_i g(\boldsymbol{x}_i; \boldsymbol{\theta}) - \log\{1 + e^{g(\boldsymbol{x}_i; \boldsymbol{\theta}) + l_i}\}] + \lambda_N \sum_{j=1}^{p}\hat{w}_j|\beta_{(j)}| \\
&= -\ell_{\mathrm{mscl}}^{\hat{\boldsymbol{\theta}}_{\mathrm{pl}}}(\boldsymbol{\theta}) + \lambda_N \sum_{j=1}^{p}\hat{w}_j|\beta_{(j)}|,
\end{aligned}$$

where $\hat{w}_j = 1/|\hat{\beta}_{\mathrm{pl}(j)}|^{\gamma}, 1 \le j \le p$. Then, we have that $\hat{\boldsymbol{u}}_N = a_N(\hat{\boldsymbol{\theta}}_{\mathrm{mscl}}^{\hat{\boldsymbol{\theta}}_{\mathrm{pl}}} - \boldsymbol{\theta}_{\mathrm{t}})$ is the minimizer of

$$\gamma_{\mathrm{mscl}}^{\hat{\boldsymbol{\theta}}_{\mathrm{pl}}}(\boldsymbol{u}) = Q_{\mathrm{mscl}}^{\hat{\boldsymbol{\theta}}_{\mathrm{pl}}}(\boldsymbol{\theta}_{\mathrm{t}} + a_N^{-1}\boldsymbol{u}) - Q_{\mathrm{mscl}}^{\hat{\boldsymbol{\theta}}_{\mathrm{pl}}}(\boldsymbol{\theta}_{\mathrm{t}}).$$

**Asymptotic normality:** We prove the asymptotic normality part in this paragraph. By Taylor's expansion,

$$\begin{aligned}
\gamma_{\mathrm{mscl}}^{\hat{\boldsymbol{\theta}}_{\mathrm{pl}}}(\boldsymbol{u}) &= -\frac{1}{a_N}\boldsymbol{u}^{\mathrm{T}}\dot{\ell}_{\mathrm{mscl}}^{\hat{\boldsymbol{\theta}}_{\mathrm{pl}}}(\boldsymbol{\theta}_{\mathrm{t}}) + \frac{1}{2a_N^2}\sum_{i=1}^{N}\delta_i^{\hat{\boldsymbol{\theta}}_{\mathrm{pl}}}\phi_{\pi}^{\hat{\boldsymbol{\theta}}_{\mathrm{pl}}}(\boldsymbol{x}_i; \boldsymbol{\theta}_{\mathrm{t}})\{\boldsymbol{u}^{\mathrm{T}}\dot{g}(\boldsymbol{x}_i; \boldsymbol{\theta}_{\mathrm{t}})\}^2 - \Delta_{\mathrm{mscl}}^{\hat{\boldsymbol{\theta}}_{\mathrm{pl}}} + R_{\mathrm{mscl}}^{\hat{\boldsymbol{\theta}}_{\mathrm{pl}}} \\
&\quad + \frac{\lambda_N}{a_N}\sum_{j=1}^{p}\hat{w}_j a_N\left(\left|\beta_{\mathrm{t}(j)} + \frac{u_{(j)}}{a_N}\right| - |\beta_{\mathrm{t}(j)}|\right).
\end{aligned}$$

First, we consider the limit behavior of the MSCL function. In [22], the authors proved that under Assumptions 1 and 3,

$$a_N^{-1}\dot{\ell}_{\mathrm{mscl}}^{\hat{\boldsymbol{\theta}}_{\mathrm{pl}}}(\boldsymbol{\theta}_{\mathrm{t}}) \rightsquigarrow (\boldsymbol{\Lambda}_{\mathrm{mscl}}^{\mathrm{pl}})^{1/2}\boldsymbol{Z}.$$

$$\frac{1}{a_N^2}\sum_{i=1}^{N}\delta_i^{\hat{\boldsymbol{\theta}}_{\mathrm{pl}}}\phi_{\pi}^{\hat{\boldsymbol{\theta}}_{\mathrm{pl}}}(\boldsymbol{x}_i; \boldsymbol{\theta}_{\mathrm{t}})\dot{g}^{\otimes 2}(\boldsymbol{x}_i; \boldsymbol{\theta}_{\mathrm{t}}) \xrightarrow{P} \boldsymbol{\Lambda}_{\mathrm{mscl}}^{\mathrm{pl}},$$

and

$$\Delta_{\mathrm{mscl}}^{\hat{\boldsymbol{\theta}}_{\mathrm{pl}}} = o_P(1), \quad R_{\mathrm{mscl}}^{\hat{\boldsymbol{\theta}}_{\mathrm{pl}}} = o_P(1).$$

Thus,

$$-\ell_{\mathrm{w}}^{\hat{\boldsymbol{\theta}}_{\mathrm{pl}}}(\boldsymbol{\theta}_{\mathrm{t}}) \rightsquigarrow -\boldsymbol{u}^{\mathrm{T}}(\boldsymbol{\Lambda}_{\mathrm{mscl}}^{\mathrm{pl}})^{1/2}\boldsymbol{Z} + \frac{1}{2}\boldsymbol{u}^{\mathrm{T}}\boldsymbol{\Lambda}_{\mathrm{mscl}}^{\mathrm{pl}}\boldsymbol{u} + o_P(1).$$

Next, we consider the limit behavior of the adaptive lasso penalty. Since we assume $\hat{\boldsymbol{\beta}}_{\mathrm{pl}}$ tp be a consistent estimator, we know that when $j \in \mathcal{A}$, i.e., $\beta_{\mathrm{t}(j)} \neq 0$,

$$\hat{w}_j = |\hat{\beta}_{\mathrm{pl}(j)}|^{-\gamma} \xrightarrow{P} |\beta_{\mathrm{t}(j)}|^{-\gamma} > 0,$$

and

$$a_N \left( \left| \beta_{\mathrm{t}(j)} + \frac{u_{(j)}}{a_N} \right| - |\beta_{\mathrm{t}(j)}| \right) \to \mathrm{sgn}(\beta_{\mathrm{t}(j)})u_{(j)}.$$

Therefore, for $j \in \mathcal{A}$, we have that

$$\frac{\lambda_N}{a_N}\hat{w}_j a_N \left( \left| \beta_{\mathrm{t}(j)} + \frac{u_{(j)}}{a_N} \right| - |\beta_{\mathrm{t}(j)}| \right) \xrightarrow{P} 0,$$

since $\lambda_N/a_N = \lambda_N/\sqrt{Ne^{\alpha_{\mathrm{t}}}} \to 0$. On the other hand, when $j \in \mathcal{A}^c$, i.e., $\beta_{\mathrm{t}(j)} = 0$, we have that for $u_{(j)} \neq 0$,

$$\frac{\lambda_N}{a_N}\hat{w}_j a_N \left( \left| \beta_{\mathrm{t}(j)} + \frac{u_{(j)}}{a_N} \right| - |\beta_{\mathrm{t}(j)}| \right) = \frac{\lambda_N}{a_N}\hat{w}_j|u_{(j)}| = \frac{\lambda_N}{a_N|\hat{\beta}_{\mathrm{pl}(j)}|^{\gamma}}|u_{(j)}| \xrightarrow{P} \infty,$$

since $\lambda_N/(\sqrt{Ne^{\alpha_{\mathrm{t}}}}|\hat{\beta}_{\mathrm{pl}(j)}|^{\gamma}) \xrightarrow{P} \infty$. Then, we have that $\gamma_{\mathrm{mscl}}^{\hat{\boldsymbol{\theta}}_{\mathrm{pl}}}(\boldsymbol{u}) \rightsquigarrow \gamma_{\mathrm{mscl}}(\boldsymbol{u})$, where

$$\gamma_{\mathrm{mscl}}(\boldsymbol{u}) = \begin{cases} \frac{1}{2}\boldsymbol{u}_{(\mathcal{A})}^{\mathrm{T}}\boldsymbol{\Lambda}_{\mathrm{mscl}}^{\mathrm{pl}}\boldsymbol{u}_{(\mathcal{A})} - \boldsymbol{u}_{(\mathcal{A})}^{\mathrm{T}}(\boldsymbol{\Lambda}_{\mathrm{mscl}}^{\mathrm{pl}})^{1/2}\boldsymbol{Z}_{(\mathcal{A})} & \text{if } u_{(j)} = 0, \forall j \notin \mathcal{A} \\ \infty & \text{otherwise.} \end{cases}$$

Note that the unique minimizer of $\gamma_{\mathrm{mscl}}(\boldsymbol{u})$ is $((\boldsymbol{\Lambda}_{\mathrm{mscl}}^{\mathrm{pl}})^{-1}\boldsymbol{Z}_{(\mathcal{A})}^{\mathrm{T}}, \boldsymbol{0})^{\mathrm{T}}$ if we put all the indexes of active variables in front. Thus, following the results of [11] and [10], we have the minimizer of $\gamma_{\mathrm{mscl}}^{\hat{\boldsymbol{\theta}}_{\mathrm{pl}}}(\boldsymbol{u})$, $\hat{\boldsymbol{u}}_N$, satisfies that

$$\hat{\boldsymbol{u}}_{N(\mathcal{A})} \rightsquigarrow (\boldsymbol{\Lambda}_{\mathrm{mscl}}^{\mathrm{pl}})^{1/2}\boldsymbol{Z}_{(\mathcal{A})} \text{ and } \hat{\boldsymbol{u}}_{N(\mathcal{A}^c)} \rightsquigarrow \boldsymbol{0}.$$

Thus,

$$\hat{\boldsymbol{u}}_{N(\mathcal{A})} = a_N(\hat{\boldsymbol{\theta}}_{\mathrm{mscl}(\mathcal{A})} - \boldsymbol{\theta}_{\mathrm{t}(\mathcal{A})}) \rightsquigarrow \mathbb{N}(\boldsymbol{0}, (\boldsymbol{\Lambda}_{\mathrm{mscl}}^{\mathrm{pl}})^{-1}).$$

We know that

$$\sqrt{N_1}\boldsymbol{V}_{\mathrm{mscl}}^{-1/2}(\boldsymbol{\Lambda}_{\mathrm{mscl}}^{\mathrm{pl}})^{-1} = a_N\mathbb{E}^{1/2}\left\{e^{f(\boldsymbol{x};\boldsymbol{\beta}_{\mathrm{t}})}\right\}\mathbb{E}^{-1/2}\left\{e^{f(\boldsymbol{x};\boldsymbol{\beta}_{\mathrm{t}})}\right\}\{1 + o_P(1)\} = a_N\{1 + o_P(1)\}.$$

Hence, applying Slusky's theorem, we have

$$\sqrt{N_1}\boldsymbol{V}_{\mathrm{mscl}(\mathcal{A})}^{-1/2}(\hat{\boldsymbol{\theta}}_{\mathrm{mscl}(\mathcal{A})} - \boldsymbol{\theta}_{\mathrm{t}(\mathcal{A})}) \rightsquigarrow \mathbb{N}(\boldsymbol{0}, \boldsymbol{I}).$$

Therefore, we prove the part of aymptotic normality.

**Consistency in variable selection** We prove the consistency in variable selection in this paragraph. From the result of asymptotic normality, we know that $\hat{\beta}_{\mathrm{mscl}(j)} \xrightarrow{P} \beta_{\mathrm{t}(j)}$ for every $j \in \mathcal{A}$ and therefore $\mathbb{P}(j \in \hat{\mathcal{A}}_{\mathrm{mscl}}) \to 1$. Thus, we only consider $j' \in \mathcal{A}^c$. When $j' \in \hat{\mathcal{A}}_{\mathrm{mscl}}$, we know that by K-K-T optimality conditions, we have

$$\lambda_N\hat{w}_{j'}\mathrm{sgn}(\hat{\beta}_{\mathrm{mscl}(j')}) = \dot{\ell}_{\mathrm{mscl}}^{\hat{\boldsymbol{\theta}}_{\mathrm{pl}}}(\hat{\boldsymbol{\theta}}_{\mathrm{mscl}}),$$

which means

$$\frac{\lambda_N\hat{w}_{j'}\mathrm{sgn}(\hat{\beta}_{\mathrm{mscl}(j')})}{a_N} = \frac{\dot{\ell}_{\mathrm{mscl}}^{\hat{\boldsymbol{\theta}}_{\mathrm{pl}}}(\hat{\boldsymbol{\theta}}_{\mathrm{mscl}})}{a_N}$$

$$= \frac{\dot{\ell}_{\mathrm{mscl}}^{\hat{\boldsymbol{\theta}}_{\mathrm{pl}}}(\boldsymbol{\theta}_{\mathrm{t}})}{a_N} + \frac{a_N\left\{\dot{\ell}_{\mathrm{mscl}}^{\hat{\boldsymbol{\theta}}_{\mathrm{pl}}}(\hat{\boldsymbol{\theta}}_{\mathrm{mscl}}^{\hat{\boldsymbol{\theta}}_{\mathrm{pl}}}) - \dot{\ell}_{\mathrm{mscl}}^{\hat{\boldsymbol{\theta}}_{\mathrm{pl}}}(\boldsymbol{\theta}_{\mathrm{t}})\right\}}{a_N^2} = I_1 + I_2.$$

We have known that $I_1 = \dot{\ell}_{\mathrm{mscl}}^{\hat{\boldsymbol{\theta}}_{\mathrm{pl}}}(\boldsymbol{\theta}_{\mathrm{t}})/a_N \rightsquigarrow \boldsymbol{Z}_{\mathrm{mscl}}$. We now prove that proof that $I_2 = O_P(1)$. We apply Taylor expansion to the $k$-th element of $\dot{\ell}_{\mathrm{mscl}}^{\hat{\boldsymbol{\theta}}_{\mathrm{pl}}}(\hat{\boldsymbol{\theta}}_{\mathrm{mscl}}^{\hat{\boldsymbol{\theta}}_{\mathrm{pl}}})$ and have that

$$\frac{a_N\left\{\dot{\ell}_{(k)}^{\hat{\boldsymbol{\theta}}_{\mathrm{pl}}}(\hat{\boldsymbol{\theta}}_{\mathrm{mscl}}^{\hat{\boldsymbol{\theta}}_{\mathrm{pl}}}) - \dot{\ell}_{(k)}^{\hat{\boldsymbol{\theta}}_{\mathrm{pl}}}(\boldsymbol{\theta}_{\mathrm{t}})\right\}}{a_N^2} = -\frac{1}{a_N^2}\sum_{i=1}^N \delta_i^{\hat{\boldsymbol{\theta}}_{\mathrm{pl}}}\phi_\pi^{\hat{\boldsymbol{\theta}}_{\mathrm{pl}}}(\boldsymbol{x}_i;\boldsymbol{\theta}_{\mathrm{t}})\dot{g}_{(k)}(\boldsymbol{x}_i;\boldsymbol{\theta}_{\mathrm{t}})\dot{g}^{\mathrm{T}}(\boldsymbol{x}_i;\boldsymbol{\theta}_{\mathrm{t}})\hat{\boldsymbol{u}}_N + \tilde{\Delta}_{(k)}^{\hat{\boldsymbol{\theta}}_{\mathrm{pl}}} + \tilde{R}_{(k)}^{\hat{\boldsymbol{\theta}}_{\mathrm{pl}}},$$

where,

$$\hat{\boldsymbol{u}}_N = a_N(\hat{\boldsymbol{\theta}}_{\mathrm{mscl}}^{\hat{\boldsymbol{\theta}}_{\mathrm{pl}}} - \boldsymbol{\theta}_{\mathrm{t}}) = O_P(1),$$

$$\tilde{\Delta}_{\mathrm{mscl}(k)}^{\hat{\boldsymbol{\theta}}_{\mathrm{pl}}} = \frac{1}{a_N^2}\sum_{i=1}^N \delta_i^{\hat{\boldsymbol{\theta}}_{\mathrm{pl}}}\left\{y_i - p_\pi^{\hat{\boldsymbol{\theta}}_{\mathrm{pl}}}(\boldsymbol{x}_i;\boldsymbol{\theta}_{\mathrm{t}})\right\}\sum_{j=1}^d \ddot{g}_{(kj)}(\boldsymbol{x}_i;\boldsymbol{\theta}_{\mathrm{t}})\tilde{u}_{(j)},$$

and

$$\tilde{R}_{\mathrm{mscl}(k)}^{\hat{\boldsymbol{\theta}}_{\mathrm{pl}}} = -\frac{1}{2a_N^3}\sum_{i=1}^N \delta_i^{\hat{\boldsymbol{\theta}}_{\mathrm{pl}}}\phi_\pi^{\hat{\boldsymbol{\theta}}_{\mathrm{pl}}}(\boldsymbol{x}_i;\acute{\boldsymbol{\theta}}_k)\left\{1 - 2p_\pi^{\hat{\boldsymbol{\theta}}_{\mathrm{pl}}}(\boldsymbol{x}_i;\acute{\boldsymbol{\theta}}_k)\right\}\dot{g}_{(k)}(\boldsymbol{x}_i;\acute{\boldsymbol{\theta}}_k)\hat{\boldsymbol{u}}_N^{\mathrm{T}}\dot{g}^{\otimes 2}(\boldsymbol{x}_i;\acute{\boldsymbol{\theta}}_k)\hat{\boldsymbol{u}}_N$$

$$- \frac{2}{2a_N^3}\sum_{i=1}^N \delta_i^{\hat{\boldsymbol{\theta}}_{\mathrm{pl}}}\phi_\pi^{\hat{\boldsymbol{\theta}}_{\mathrm{pl}}}(\boldsymbol{x}_i;\acute{\boldsymbol{\theta}}_k)\left\{\hat{\boldsymbol{u}}_N^{\mathrm{T}}\frac{\partial\dot{g}_{(k)}(\boldsymbol{x}_i;\acute{\boldsymbol{\theta}}_k)}{\partial\boldsymbol{\theta}}\right\}\left\{\hat{\boldsymbol{u}}_N^{\mathrm{T}}\dot{g}(\boldsymbol{x}_i;\acute{\boldsymbol{\theta}}_k)\right\}$$

$$- \frac{1}{2a_N^2}\sum_{i=1}^N \delta_i^{\hat{\boldsymbol{\theta}}_{\mathrm{pl}}}\phi_\pi^{\hat{\boldsymbol{\theta}}_{\mathrm{pl}}}(\boldsymbol{x}_i;\acute{\boldsymbol{\theta}}_k)\dot{g}_{(k)}(\boldsymbol{x}_i;\acute{\boldsymbol{\theta}}_k)\left\{\hat{\boldsymbol{u}}_N^{\mathrm{T}}\ddot{g}(\boldsymbol{x}_i;\acute{\boldsymbol{\theta}}_k)\hat{\boldsymbol{u}}_N\right\}$$

$$+ \frac{1}{2a_N^3}\sum_{i=1}^N \delta_i^{\hat{\boldsymbol{\theta}}_{\mathrm{pl}}}\left\{y_i - p_\pi^{\hat{\boldsymbol{\theta}}_{\mathrm{pl}}}(\boldsymbol{x}_i;\acute{\boldsymbol{\theta}}_k)\right\}\hat{\boldsymbol{u}}_N^{\mathrm{T}}\frac{\partial^2\dot{g}_{(k)}(\boldsymbol{x}_i;\acute{\boldsymbol{\theta}}_k)}{\partial\boldsymbol{\theta}^2}\hat{\boldsymbol{u}}_N.$$

where $\acute{\boldsymbol{\theta}}_k$ is between $\hat{\boldsymbol{\theta}}_{\mathrm{mscl}}$ and $\boldsymbol{\theta}_{\mathrm{t}}$. First, we prove that $\tilde{R}_{\mathrm{mscl}(k)}$ is $o_P(1)$. We have that

$$|\tilde{R}_{\mathrm{mscl}(k)}^{\hat{\boldsymbol{\theta}}_{\mathrm{pl}}}| \leq \frac{\|\hat{\boldsymbol{u}}_N\|^2}{2a_N^3}\sum_{i=1}^N \delta_i^{\hat{\boldsymbol{\theta}}_{\mathrm{pl}}}\phi_\pi^{\hat{\boldsymbol{\theta}}_{\mathrm{pl}}}(\boldsymbol{x}_i;\acute{\boldsymbol{\theta}}_k)\left|1 - 2p_\pi^{\hat{\boldsymbol{\theta}}_{\mathrm{pl}}}(\boldsymbol{x}_i;\acute{\boldsymbol{\theta}}_k)\right|\left|\dot{g}_{(k)}(\boldsymbol{x}_i;\acute{\boldsymbol{\theta}}_k)\right|\left\|\dot{g}(\boldsymbol{x}_i;\acute{\boldsymbol{\theta}}_k)\right\|^2$$

$$+ \frac{2\|\hat{\boldsymbol{u}}_N\|^2}{2a_N^3}\sum_{i=1}^N \delta_i^{\hat{\boldsymbol{\theta}}_{\mathrm{pl}}}\phi_\pi^{\hat{\boldsymbol{\theta}}_{\mathrm{pl}}}(\boldsymbol{x}_i;\acute{\boldsymbol{\theta}}_k)\left\|\frac{\partial\dot{g}_{(k)}(\boldsymbol{x}_i;\acute{\boldsymbol{\theta}}_k)}{\partial\boldsymbol{\theta}}\right\|\left\|\dot{g}(\boldsymbol{x}_i;\acute{\boldsymbol{\theta}}_k)\right\|$$

$$+ \frac{\|\hat{\boldsymbol{u}}_N\|^2}{2a_N^3}\sum_{i=1}^N \delta_i^{\hat{\boldsymbol{\theta}}_{\mathrm{pl}}}\phi_\pi^{\hat{\boldsymbol{\theta}}_{\mathrm{pl}}}(\boldsymbol{x}_i;\acute{\boldsymbol{\theta}}_k)\left|\dot{g}_{(k)}(\boldsymbol{x}_i;\acute{\boldsymbol{\theta}}_k)\right|\left\|\ddot{g}(\boldsymbol{x}_i;\acute{\boldsymbol{\theta}}_k)\right\|$$

$$+ \frac{\|\hat{\boldsymbol{u}}_N\|^2}{2a_N^3}\sum_{i=1}^N \delta_i^{\hat{\boldsymbol{\theta}}_{\mathrm{pl}}}p_\pi^{\hat{\boldsymbol{\theta}}_{\mathrm{pl}}}(\boldsymbol{x}_i;\acute{\boldsymbol{\theta}}_k)\left\|\frac{\partial^2\dot{g}_{(k)}(\boldsymbol{x}_i;\acute{\boldsymbol{\theta}}_k)}{\partial\boldsymbol{\theta}^2}\right\| + \frac{\|\hat{\boldsymbol{u}}_N\|^2}{2a_N^3}\sum_{i=1}^N \delta_i^{\hat{\boldsymbol{\theta}}_{\mathrm{pl}}}y_i\left\|\frac{\partial^2\dot{g}_{(k)}(\boldsymbol{x}_i;\acute{\boldsymbol{\theta}}_k)}{\partial\boldsymbol{\theta}^2}\right\|$$

$$\leq \frac{\|\hat{\boldsymbol{u}}_N\|^2}{2a_N^3}\sum_{i=1}^N \delta_i^{\hat{\boldsymbol{\theta}}_{\mathrm{pl}}}p_\pi^{\hat{\boldsymbol{\theta}}_{\mathrm{pl}}}(\boldsymbol{x}_i;\acute{\boldsymbol{\theta}}_k)C(\boldsymbol{x}_i;\acute{\boldsymbol{\theta}}_k) + \frac{\|\hat{\boldsymbol{u}}_N\|^2}{2a_N^3}\sum_{i=1}^N \delta_i^{\hat{\boldsymbol{\theta}}_{\mathrm{pl}}}y_i B(\boldsymbol{x}_i)$$

$$\leq \frac{\|\hat{\boldsymbol{u}}_N\|^2 e^{\acute{\alpha}_k - \alpha_{\mathrm{t}}}e^{\alpha_{\mathrm{t}}}}{2a_N^3}\sum_{i=1}^N \delta_i^{\hat{\boldsymbol{\theta}}_{\mathrm{pl}}}e^{f(\boldsymbol{x}_i;\acute{\boldsymbol{\beta}}_k) - \log\{\rho\varphi(\boldsymbol{x}_i)\}}C(\boldsymbol{x}_i;\acute{\boldsymbol{\theta}}_k) + \frac{\|\hat{\boldsymbol{u}}_N\|^2}{2a_N^3}\sum_{i=1}^N \delta_i^{\hat{\boldsymbol{\theta}}_{\mathrm{pl}}}y_i B(\boldsymbol{x}_i)$$

$$\leq \frac{\|\hat{\boldsymbol{u}}_N\|^2 e^{\acute{\alpha}_k - \alpha_{\mathrm{t}}}}{2Na_N}\sum_{i=1}^N \delta_i^{\hat{\boldsymbol{\theta}}_{\mathrm{pl}}}e^{f(\boldsymbol{x}_i;\acute{\boldsymbol{\beta}}_k) - \log\{\rho\varphi(\boldsymbol{x}_i)\}}C(\boldsymbol{x}_i;\acute{\boldsymbol{\theta}}_k) + \frac{\|\hat{\boldsymbol{u}}_N\|^2}{2a_N^3}\sum_{i=1}^N \delta_i^{\hat{\boldsymbol{\theta}}_{\mathrm{pl}}}y_i B(\boldsymbol{x}_i)$$

$$= \frac{\|\hat{\boldsymbol{u}}_N\|^2 e^{\acute{\alpha}_k - \alpha_{\mathrm{t}}}}{2Na_N\rho}\sum_{i=1}^N \delta_i^{\hat{\boldsymbol{\theta}}_{\mathrm{pl}}}\varphi^{-1}(\boldsymbol{x}_i)e^{f(\boldsymbol{x}_i;\acute{\boldsymbol{\beta}}_k)}C(\boldsymbol{x}_i;\acute{\boldsymbol{\theta}}_k) + \frac{\|\hat{\boldsymbol{u}}_N\|^2}{2a_N^3}\sum_{i=1}^N \delta_i^{\hat{\boldsymbol{\theta}}_{\mathrm{pl}}}y_i B(\boldsymbol{x}_i)$$

$$\leq \frac{\|\hat{\boldsymbol{u}}_N\|^2 e^{\acute{\alpha}_k - \alpha_\mathrm{t}}}{2Na_N\rho} \sum_{i=1}^{N} \delta_i^{\hat{\boldsymbol{\theta}}_\mathrm{pl}} \varphi^{-1}(\boldsymbol{x}_i) B(\boldsymbol{x}_i) + \frac{\|\hat{\boldsymbol{u}}_N\|^2}{2a_N^3} \sum_{i=1}^{N} y_i B(\boldsymbol{x}_i)$$

$$= o_P(1),$$

where

$$C(\boldsymbol{x}_i; \acute{\boldsymbol{\theta}}) = \left| \dot{g}_{(k)}(\boldsymbol{x}_i; \acute{\boldsymbol{\theta}}) \right| \left\{ \left\| \dot{g}(\boldsymbol{x}_i; \acute{\boldsymbol{\theta}}_k) \right\|^2 + \left\| \ddot{g}(\boldsymbol{x}_i; \acute{\boldsymbol{\theta}}) \right\| \right\}$$

$$+ \left\| \frac{\partial \dot{g}_{(k)}(\boldsymbol{x}_i; \acute{\boldsymbol{\theta}}_k)}{\partial \boldsymbol{\theta}} \right\| \left\| \dot{g}(\boldsymbol{x}_i; \acute{\boldsymbol{\theta}}_k) \right\| + \left\| \frac{\partial^2 \dot{g}_{(k)}(\boldsymbol{x}_i; \acute{\boldsymbol{\theta}}_k)}{\partial \boldsymbol{\theta}^2} \right\|.$$

Therefore, we proved that $\tilde{R}_{\mathrm{mscl}(k)} = o_P(1)$. Next, we prove that $\tilde{\Delta}_{\mathrm{mscl}(k)} = o_P(1)$. We know that $\mathbb{E}\left[ a_N^{-2} \sum_{i=1}^{N} \delta_i^{\hat{\boldsymbol{\theta}}_\mathrm{pl}} \left\{ y_i - p_\pi^{\hat{\boldsymbol{\theta}}_\mathrm{pl}}(\boldsymbol{x}_i; \boldsymbol{\theta}_\mathrm{t}) \right\} \ddot{g}(\boldsymbol{x}_i; \boldsymbol{\theta}_\mathrm{t}) \mid \hat{\boldsymbol{\theta}}_\mathrm{pl} \right] = \mathbf{0}$. We also have that for the every element of $a_N^{-2} \sum_{i=1}^{N} \{ y_i - p(\boldsymbol{x}_i; \boldsymbol{\theta}_\mathrm{t}) \} \ddot{g}(\boldsymbol{x}_i; \boldsymbol{\theta}_\mathrm{t})$, we have

$$\mathbb{V}\left[ a_N^{-2} \sum_{i=1}^{N} \delta_i^{\hat{\boldsymbol{\theta}}_\mathrm{pl}} \left\{ y_i - p_\pi^{\hat{\boldsymbol{\theta}}_\mathrm{pl}}(\boldsymbol{x}_i; \boldsymbol{\theta}_\mathrm{t}) \right\} \ddot{g}_{(jl)}(\boldsymbol{x}_i; \boldsymbol{\theta}_\mathrm{t}) \mid \hat{\boldsymbol{\theta}}_\mathrm{pl} \right]$$

$$\leq \frac{1}{a_N^4} \sum_{i=1}^{N} \mathbb{E}\left\{ \delta_i^{\hat{\boldsymbol{\theta}}_\mathrm{pl}} p_\pi^{\hat{\boldsymbol{\theta}}_\mathrm{pl}}(\boldsymbol{x}_i; \boldsymbol{\theta}_\mathrm{t}) \ddot{g}_{(jl)}^2(\boldsymbol{x}_i; \boldsymbol{\theta}_\mathrm{t}) \mid \hat{\boldsymbol{\theta}}_\mathrm{pl} \right\} \leq \frac{1}{a_N^2} \mathbb{E}[e^{f(\boldsymbol{x};\boldsymbol{\beta}_\mathrm{t})} \|\ddot{g}(\boldsymbol{x}; \boldsymbol{\theta}_\mathrm{t})\|^2] \to 0.$$

Thus, due to Chebyshev's inequality, we know that $\tilde{\Delta}_{\mathrm{mscl}}^{\hat{\boldsymbol{\theta}}_\mathrm{pl}} = o_P(1)$. Since we know that $\frac{1}{a_N^2} \sum_{i=1}^{N} \delta_i^{\hat{\boldsymbol{\theta}}_\mathrm{pl}} \phi_\pi^{\hat{\boldsymbol{\theta}}_\mathrm{pl}}(\boldsymbol{x}_i; \boldsymbol{\theta}_\mathrm{t}) \dot{g}^{\otimes 2}(\boldsymbol{x}_i; \boldsymbol{\theta}_\mathrm{t}) = O_P(1)$. Hence, we have that

$$\frac{\dot{\ell}_{\mathrm{mscl}}(\hat{\boldsymbol{\theta}}_{\mathrm{mscl}})}{a_N} = O_P(1).$$

Note that we also have

$$\frac{\lambda_N \hat{w}_{j'}}{a_N} = \frac{\lambda_N}{a_N} \frac{1}{|\hat{\beta}_{\mathrm{pl}(j')}|^\gamma} \xrightarrow{P} \infty.$$

Therefore,

$$\mathbb{P}(j' \in \mathcal{A}_N) \leq \mathbb{P}\left\{ \lambda_N \hat{w}_{j'} \mathrm{sgn}(\hat{\beta}_{\mathrm{mscl}(j')}) = \ell_{\mathrm{mscl}}^{\hat{\boldsymbol{\theta}}_p}(\hat{\boldsymbol{\theta}}_{\mathrm{mscl}}) \right\}$$

$$= \mathbb{P}\left\{ \frac{\lambda_N \hat{w}_{j'} \mathrm{sgn}(\hat{\beta}_{\mathrm{mscl}(j')})}{a_N} = \frac{\ell_{\mathrm{mscl}}^{\hat{\boldsymbol{\theta}}_p}(\hat{\boldsymbol{\theta}}_{\mathrm{mscl}})}{a_N} \right\} \to 0.$$

Thus, we prove the part of consistency of variable selection. $\qquad\square$

### B.7 Proof of Theorem 4

*Proof.* Letting $h = 1 + c\{\varphi(\boldsymbol{x})\}^{-1} e^{f(\boldsymbol{x};\boldsymbol{\beta}_\mathrm{t})}$, $\boldsymbol{v} = \sqrt{e^{f(\boldsymbol{x};\boldsymbol{\beta}_\mathrm{t})}} \dot{g}_{(\mathcal{A})}(\boldsymbol{x}; \tilde{\boldsymbol{\theta}})$, $\boldsymbol{f} = h^{\frac{1}{2}} \boldsymbol{v}$, and $\boldsymbol{g} = h^{-\frac{1}{2}} \boldsymbol{v}$, we have that

$$\mathbb{E}(\boldsymbol{g}\boldsymbol{f}^\mathrm{T}) = \mathbb{E}(\boldsymbol{f}\boldsymbol{g}^\mathrm{T}) = \mathbb{E}(\boldsymbol{v}\boldsymbol{v}^\mathrm{T}) = \mathbb{E}\left\{ e^{f(\boldsymbol{x};\boldsymbol{\beta}_\mathrm{t})} \dot{g}_{(\mathcal{A})}^{\otimes 2}(\boldsymbol{x}; \boldsymbol{\theta}_\mathrm{t}) \right\} = \boldsymbol{M}_{(\mathcal{A})},$$

$$\mathbb{E}(\boldsymbol{f}\boldsymbol{f}^\mathrm{T}) = \mathbb{E}(h\boldsymbol{v}\boldsymbol{v}^\mathrm{T}) = \mathbb{E}\left[ \left\{ 1 + \frac{c e^{f(\boldsymbol{z};\boldsymbol{\beta}_\mathrm{t})}}{\varphi(\boldsymbol{x})} \right\} e^{f(\boldsymbol{x};\boldsymbol{\beta}_\mathrm{t})} \dot{g}_{(\mathcal{A})}^{\otimes 2}(\boldsymbol{x}; \boldsymbol{\theta}_\mathrm{t}) \right] = \boldsymbol{M}_{\mathrm{w}(\mathcal{A})},$$

and

$$\mathbb{E}(\boldsymbol{g}\boldsymbol{g}^\mathrm{T}) = \mathbb{E}(h^{-1}\boldsymbol{v}\boldsymbol{v}^\mathrm{T}) = \mathbb{E}\left[ \frac{e^{f(\boldsymbol{x};\boldsymbol{\beta}_\mathrm{t})} \dot{g}_{(\mathcal{A})}^{\otimes 2}(\boldsymbol{x}; \boldsymbol{\theta}_\mathrm{t})}{1 + c\varphi^{-1}(\boldsymbol{x}) e^{f(\boldsymbol{x};\boldsymbol{\beta}_\mathrm{t})}} \right] = \boldsymbol{\Lambda}_{\mathrm{mscl}(\mathcal{A})}.$$

Now, applying the matrix form of Cauchy-Schwartz's inequality (see [19]), we have that

$$\boldsymbol{\Lambda}_{\mathrm{mscl}(\mathcal{A})} = \mathbb{E}(\boldsymbol{g}\boldsymbol{g}^\mathrm{T}) \geq \mathbb{E}(\boldsymbol{g}\boldsymbol{f}^\mathrm{T})\{\mathbb{E}(\boldsymbol{f}\boldsymbol{f}^\mathrm{T})\}^{-1}\mathbb{E}(\boldsymbol{f}\boldsymbol{g}^\mathrm{T})$$

$$= \boldsymbol{M}_{(\mathcal{A})}\{\boldsymbol{M}_{\mathrm{w}(\mathcal{A})}\}^{-1}\boldsymbol{M}_{(\mathcal{A})} = \mathbb{E}\left\{ e^{f(\boldsymbol{x};\boldsymbol{\beta}_\mathrm{t})} \right\} \{\boldsymbol{V}_{\mathrm{w}(\mathcal{A})}\}^{-1}.$$

Therefore, simple algebra shows that

$$\boldsymbol{V}_{\mathrm{mscl}(\mathcal{A})} = \mathbb{E}\left\{ e^{f(\boldsymbol{x};\boldsymbol{\beta}_\mathrm{t})} \right\} \{\boldsymbol{\Lambda}_{\mathrm{mscl}(\mathcal{A})}\}^{-1} \leq \boldsymbol{V}_{\mathrm{w}(\mathcal{A})},$$

which complete the proof $\qquad\square$

## C  Details about the practical algorithm and its complexity

### C.1  Two-step algorithm

We take a pilot sample by uniform sampling with the sampling rate $\rho_1 = N_{\mathrm{pl}}/2N_1$ for the ones and $\rho_0 = N_{\mathrm{pl}}/2N_0$ for the zeros. Denote a pilot sample of actual sample size $N_{\mathrm{pl}}^*$ as $\{(\boldsymbol{x}_i^{\mathrm{pl}}, y_i^{\mathrm{pl}})\}_{i=1}^{N_{\mathrm{pl}}^*}$, the pilot estimate of $\boldsymbol{\theta}$ as $\hat{\boldsymbol{\theta}}_{\mathrm{pl}}$, and the pilot estimate of the active set as $\hat{\mathcal{A}}_{\mathrm{pl}} = \{j : \hat{\beta}_{\mathrm{pl}(j)} \neq 0\}$. We propose the following moment estimators of $\boldsymbol{M}_{(\mathcal{A})}$ and $\boldsymbol{\Omega}_{(\mathcal{A})}$:

$$\hat{\boldsymbol{M}}_{(\hat{\mathcal{A}}_{\mathrm{pl}})}^{\mathrm{pl}} = \frac{1}{N_{\mathrm{pl}}} \sum_{i=1}^{N_{\mathrm{pl}}^*} \frac{e^{f(\boldsymbol{x}_i^{\mathrm{pl}T}\hat{\boldsymbol{\beta}}_{\mathrm{pl}})} \dot{g}_{(\hat{\mathcal{A}}_{\mathrm{pl}})}^{\otimes 2}(\boldsymbol{x}_i^{\mathrm{pl}}; \hat{\boldsymbol{\theta}}_{\mathrm{pl}})}{\rho_0 + y_i^{\mathrm{pl}}(\rho_1 - \rho_0)}, \tag{11}$$

$$\hat{\boldsymbol{\Omega}}_{(\hat{\mathcal{A}}_{\mathrm{pl}})}^{\mathrm{pl}} = \frac{1}{N_{\mathrm{pl}}} \sum_{i=1}^{N_{\mathrm{pl}}^*} \frac{e^{2f(\boldsymbol{x}_i^{\mathrm{pl}T}\hat{\boldsymbol{\beta}}_{\mathrm{pl}})} \dot{g}_{(\hat{\mathcal{A}}_{\mathrm{pl}})}^{\otimes 2}(\boldsymbol{x}_i^{\mathrm{pl}}; \hat{\boldsymbol{\theta}}_{\mathrm{pl}})}{\rho_0 + y_i^{\mathrm{pl}}(\rho_1 - \rho_0)}, \tag{12}$$

respectively. We also use the following moment estimator to estimate the denominator of (4):

$$\frac{1}{N_{\mathrm{pl}}} \sum_{i=1}^{N_{\mathrm{pl}}^*} \frac{\omega_i^{\mathrm{A-OS}}}{\rho_0 + y_i^{\mathrm{pl}}(\rho_1 - \rho_0)}, \tag{13}$$

where $\omega_i^{\mathrm{A-OS}} = p(\boldsymbol{x}_i^{\mathrm{pl}}; \hat{\boldsymbol{\theta}}_{\mathrm{pl}}) \| (\hat{\boldsymbol{M}}_{(\hat{\mathcal{A}}_{\mathrm{pl}})}^{\mathrm{pl}})^{-1} \dot{g}_{(\hat{\mathcal{A}}_{\mathrm{pl}})}(\boldsymbol{x}_i^{\mathrm{pl}}; \hat{\boldsymbol{\theta}}_{\mathrm{pl}}) \|$. If using (5) or (7), we use

$$\omega_i^{\mathrm{L-OS}} = p(\boldsymbol{x}_i^{\mathrm{pl}}; \hat{\boldsymbol{\theta}}_{\mathrm{pl}}) \| \dot{g}_{(\hat{\mathcal{A}}_{\mathrm{pl}})}(\boldsymbol{x}_i^{\mathrm{pl}}; \hat{\boldsymbol{\theta}}_{\mathrm{pl}}) \|, \text{ or}$$

$$\omega_i^{\mathrm{P-OS}} = p(\boldsymbol{x}_i^{\mathrm{pl}}; \hat{\boldsymbol{\theta}}_{\mathrm{pl}}) \| (\hat{\boldsymbol{\Omega}}_{(\hat{\mathcal{A}}_{\mathrm{pl}})}^{\mathrm{pl}})^{1/2} (\hat{\boldsymbol{M}}_{(\hat{\mathcal{A}}_{\mathrm{pl}})}^{\mathrm{pl}})^{-1} \dot{g}_{(\hat{\mathcal{A}}_{\mathrm{pl}})}(\boldsymbol{x}_i^{\mathrm{pl}}; \hat{\boldsymbol{\theta}}_{\mathrm{pl}}) \|,$$

respectively, instead of $\omega_i^{\mathrm{A-OS}}$ in (13). Now, we present the proposed two-step procedure in Algorithm 3 with more details than Algorithm 2 in Section 4.

---

**Algorithm 3** Subsampling adaptive lasso algorithm

---

1:      • Take a pilot sample $\{(\boldsymbol{x}_i^{\mathrm{pl}}, y_i^{\mathrm{pl}})\}_{i=1}^{N_{\mathrm{pl}}^*}$ of expected sample size $N_{\mathrm{pl}}$ using $\{\pi(y_i) = \rho_0 + y_i(\rho_1 - \rho_0)\}_{i=1}^N$ and obtain a pilot estimator

$$\hat{\boldsymbol{\theta}}_{\mathrm{pl}} := \arg\max_{\boldsymbol{\theta}} \left\{ \sum_{i=1}^{N_{\mathrm{pl}}^*} [y_i^{\mathrm{pl}} g(\boldsymbol{x}_i^{\mathrm{pl}}; \boldsymbol{\theta}) - \log\{1 + e^{g(\boldsymbol{x}_i^{\mathrm{pl}}; \boldsymbol{\theta}) + l}\}] - \lambda_{\mathrm{pl}} \sum_{j=1}^p |\beta_{(j)}| \right\}, \tag{14}$$

     where $N_{\mathrm{pl}}^*$ is the actual pilot sample size and $l = \log(N_0/N_1)$. We call this first stage screening.

     • Calculate approximate optimal sampling probabilities $\{\hat{\pi}(\boldsymbol{x}_i, y_i) = y_i + (1 - y_i)\rho\hat{\varphi}(x_i)\}_{i=1}^N$ by replacing $\hat{\varphi}(x_i)$ with $\varphi_{\mathrm{A-OS}}^{\mathrm{adp}}(\boldsymbol{x}_i; \hat{\boldsymbol{\theta}}_{\mathrm{pl}})$, $\varphi_{\mathrm{L-OS}}^{\mathrm{adp}}(\boldsymbol{x}_i; \hat{\boldsymbol{\theta}}_{\mathrm{pl}})$, or $\varphi_{\mathrm{P-OS}}^{\mathrm{adp}}(\boldsymbol{x}_i; \hat{\boldsymbol{\theta}}_{\mathrm{pl}})$, based on (4), (5), or (7), respectively. The denominator of (4) is estimated using (13), and we replace $\omega_i^{\mathrm{A-OS}}$ with $\omega_i^{\mathrm{L-OS}}$ or $\omega_i^{\mathrm{P-OS}}$ for the denominator of (5) or (7), respectively. If using $\pi_{\mathrm{A-OS}}^{\mathrm{adp}}(\boldsymbol{x})$ or $\pi_{\mathrm{P-OS}}^{\mathrm{adp}}(\boldsymbol{x})$, estimate $\boldsymbol{M}_{(\mathcal{A})}$ and $\boldsymbol{\Omega}_{(\mathcal{A})}$ using the moment estimators in (11) and (12), respectively.

2: Use Algorithm 1 with the estimated optimal sampling probabilities to obtain a subsample $\{(\boldsymbol{x}_i^{\mathrm{sub}}, y_i^{\mathrm{sub}})\}_{i=1}^{N_{\mathrm{sub}}^*}$ and compute the adaptive lasso estimator:

$$\hat{\boldsymbol{\theta}}_{\mathrm{mscl}}^{\mathrm{adp}} := \arg\max_{\boldsymbol{\theta}} \left\{ \sum_{i=1}^{N_{\mathrm{sub}}^*} [y_i^{\mathrm{sub}} g(\boldsymbol{x}_i^{\mathrm{sub}}; \boldsymbol{\theta}) - \log\{1 + e^{g(\boldsymbol{x}_i^{\mathrm{sub}}; \boldsymbol{\theta}) + l_i}\}] - \lambda_N \sum_{j \in \hat{\mathcal{A}}_{\mathrm{pl}}} \frac{|\beta_{(j)}|}{|\hat{\beta}_{\mathrm{pl}(j)}|^\gamma} \right\},$$

     where $N_{\mathrm{sub}}^*$ is the actual subsample size, based on the smaller model obtain from the first stage screening. We call this step the second stage screening.

---

**Remark 3.** *Our algorithm naturally integrates the MSCL function with the adaptive lasso penalty. It can also be implemented when $p > N$ as long as the dimension of selected variables is smaller than $N$ in the first-stage screening. If the model is sparse and the data are massive, this is usually possible in practice. Screening algorithms such as sure independence screening [6] can also be used for the first stage screening to guarantee that the dimension of second-stage screening is smaller than the subsample size. Furthermore, the first stage screening can help to speed up the computation, as shown by the analysis of computational complexity in the next section.*

We consider a coordinate desent method to calculate the estimators defined in Algorithm 3. (see [8], [9] and [27]). In each cycle, we need to find an optimal direction $\boldsymbol{d}$ at a starting point $\tilde{\boldsymbol{\theta}}$. We consider the quardratic approximation of $Q_{\mathrm{mscl}}^{\hat{\boldsymbol{\theta}}_{\mathrm{pl}}}(\tilde{\boldsymbol{\theta}} + \boldsymbol{d}) - Q_{\mathrm{mscl}}^{\hat{\boldsymbol{\theta}}_{\mathrm{pl}}}(\tilde{\boldsymbol{\theta}})$, which is

$$
\begin{aligned}
& Q_{\mathrm{mscl}}^{\hat{\boldsymbol{\theta}}_{\mathrm{pl}}}(\tilde{\boldsymbol{\theta}} + \boldsymbol{d}) - Q_{\mathrm{mscl}}^{\hat{\boldsymbol{\theta}}_{\mathrm{pl}}}(\tilde{\boldsymbol{\theta}}) \\
& = \sum_{i=1}^{N} \delta_i[-y_i g(\boldsymbol{x}_i; \tilde{\boldsymbol{\theta}} + \boldsymbol{d}) + \log\{1 + e^{g(\boldsymbol{x}_i; \tilde{\boldsymbol{\theta}} + \boldsymbol{d}) + l_i}\}] + \lambda_N \sum_{j=1}^{p} \hat{w}_j |\beta_{(j)} + d_{(j)}| \\
& \quad - \sum_{i=1}^{N} \delta_i[-y_i g(\boldsymbol{x}_i; \tilde{\boldsymbol{\theta}}) - \log\{1 + e^{g(\boldsymbol{x}_i; \tilde{\boldsymbol{\theta}}) + l_i}\}] + \lambda_N \sum_{j=1}^{p} \hat{w}_j |\beta_{(j)}| \\
& \approx \dot{\ell}_{\mathrm{mscl}}^{\hat{\boldsymbol{\theta}}_{\mathrm{pl}}}(\tilde{\boldsymbol{\theta}})^T \boldsymbol{d} + \frac{1}{2} \boldsymbol{d}^T \ddot{\ell}_{\mathrm{mscl}}^{\hat{\boldsymbol{\theta}}_{\mathrm{pl}}}(\tilde{\boldsymbol{\theta}}) \boldsymbol{d} + \lambda_N \sum_{j=1}^{p} \left\{ \hat{w}_j |\tilde{\beta}_{(j)} + d_{(j)}| - \hat{w}_j |\tilde{\beta}_{(j)}| \right\}
\end{aligned}
$$

where $\dot{\ell}_{\mathrm{mscl}}^{\hat{\boldsymbol{\theta}}_{\mathrm{pl}}}(\tilde{\boldsymbol{\theta}}) = -\sum_{i=1}^{N} \delta_i^{\hat{\boldsymbol{\theta}}_{\mathrm{pl}}} \left\{ y_i - p_\pi^{\hat{\boldsymbol{\theta}}_{\mathrm{pl}}}(\boldsymbol{x}_i, \tilde{\boldsymbol{\theta}}) \right\} \dot{g}(\boldsymbol{x}_i; \tilde{\boldsymbol{\theta}})$ and $\ddot{\ell}_{\mathrm{mscl}}^{\hat{\boldsymbol{\theta}}_{\mathrm{pl}}}(\tilde{\boldsymbol{\theta}}) = \sum_{i=1}^{N} \delta_i^{\hat{\boldsymbol{\theta}}_{\mathrm{pl}}} \phi_\pi^{\hat{\boldsymbol{\theta}}_{\mathrm{pl}}}(\boldsymbol{x}_i, \tilde{\boldsymbol{\theta}}) \dot{g}^{\otimes 2}(\boldsymbol{x}_i; \tilde{\boldsymbol{\theta}})$. Thus, using coordinate desent to obtain the optimal direction, the quadratic approximation for the $j$-th element is given as

$$
Q_{\mathrm{mscl}}^{\hat{\boldsymbol{\theta}}_{\mathrm{pl}}}(\boldsymbol{d} + z\boldsymbol{e}_j) - Q_{\mathrm{mscl}}^{\hat{\boldsymbol{\theta}}_{\mathrm{pl}}}(\boldsymbol{d}) = \dot{\ell}_{\mathrm{mscl}(j)}(\tilde{\boldsymbol{\theta}})z + \left\{ \ddot{\ell}_{\mathrm{mscl}}(\tilde{\boldsymbol{\theta}})\boldsymbol{d} \right\}_{(j)} z + \frac{1}{2} \ddot{\ell}_{\mathrm{mscl}(jj)}(\tilde{\boldsymbol{\theta}})z^2
$$
$$
+ \lambda_N \hat{w}_j |\tilde{\beta}_{(j)} + d_{(j)} + z| - \lambda_N \hat{w}_j |\tilde{\beta}_{(j)} + d_{(j)}|.
$$

Then, we have the value of $z$ that minimize $Q_{\mathrm{mscl}}^{\hat{\boldsymbol{\theta}}_{\mathrm{pl}}}(\boldsymbol{d} + z\boldsymbol{e}_j) - Q_{\mathrm{mscl}}^{\hat{\boldsymbol{\theta}}_{\mathrm{pl}}}(\boldsymbol{d})$ is

$$
z^{**} = \begin{cases} \dfrac{\dot{\ell}_{\mathrm{mscl}(j)}^{\hat{\boldsymbol{\theta}}_{\mathrm{pl}}}(\tilde{\boldsymbol{\theta}}) + \left\{ \ddot{\ell}_{\mathrm{mscl}}^{\hat{\boldsymbol{\theta}}_{\mathrm{pl}}}(\tilde{\boldsymbol{\theta}})\boldsymbol{d} \right\}_{(j)} + \lambda \hat{w}_j}{-\ddot{\ell}_{\mathrm{mscl}(jj)}^{\hat{\boldsymbol{\theta}}_{\mathrm{pl}}}(\tilde{\boldsymbol{\theta}})} & \text{if } \tilde{\beta}_{(j)} + d_{(j)} + z \geq 0 \\[4mm] \dfrac{\dot{\ell}_{\mathrm{mscl}(j)}^{\hat{\boldsymbol{\theta}}_{\mathrm{pl}}}(\tilde{\boldsymbol{\theta}}) + \left\{ \ddot{\ell}_{\mathrm{mscl}}^{\hat{\boldsymbol{\theta}}_{\mathrm{pl}}}(\tilde{\boldsymbol{\theta}})\boldsymbol{d} \right\}_{(j)} - \lambda \hat{w}_j}{-\ddot{\ell}_{\mathrm{mscl}(jj)}^{\hat{\boldsymbol{\theta}}_{\mathrm{pl}}}(\tilde{\boldsymbol{\theta}})} & \text{if } \tilde{\beta}_{(j)} + d_{(j)} + z \leq 0 \\[4mm] -\tilde{\beta}_{(j)} - d_{(j)} & \text{otherwise,} \end{cases}
$$

which is the same as

$$
z^{**} = \max\left\{ z_1, -\tilde{\beta}_{(j)} - d_{(j)} \right\} - \max\left\{ -z_2, \tilde{\beta}_{(j)} + d_{(j)} \right\} + \tilde{\beta}_{(j)} + d_{(j)},
$$

where

$$
z_1 = \frac{\dot{\ell}_{\mathrm{mscl}(j)}^{\hat{\boldsymbol{\theta}}_{\mathrm{pl}}}(\tilde{\boldsymbol{\theta}}) + \left\{ \ddot{\ell}_{\mathrm{mscl}}^{\hat{\boldsymbol{\theta}}_{\mathrm{pl}}}(\tilde{\boldsymbol{\theta}})\boldsymbol{d} \right\}_{(j)} + \lambda \hat{w}_j}{-\ddot{\ell}_{\mathrm{mscl}(jj)}^{\hat{\boldsymbol{\theta}}_{\mathrm{pl}}}(\tilde{\boldsymbol{\theta}})},
$$

and

$$
z_2 = \frac{\dot{\ell}_{\mathrm{mscl}(j)}^{\hat{\boldsymbol{\theta}}_{\mathrm{pl}}}(\tilde{\boldsymbol{\theta}}) + \left\{ \ddot{\ell}_{\mathrm{mscl}}^{\hat{\boldsymbol{\theta}}_{\mathrm{pl}}}(\tilde{\boldsymbol{\theta}})\boldsymbol{d} \right\}_{(j)} - \lambda \hat{w}_j}{-\ddot{\ell}_{\mathrm{mscl}(jj)}^{\hat{\boldsymbol{\theta}}_{\mathrm{pl}}}(\tilde{\boldsymbol{\theta}})}.
$$

For the special form of $g(\boldsymbol{x}; \boldsymbol{\theta}) = \alpha + f(\boldsymbol{x}^{\mathrm{T}}\boldsymbol{\beta})$, we know that

$$
\ddot{\ell}_{\mathrm{mscl}}^{\hat{\boldsymbol{\theta}}_{\mathrm{pl}}}(\tilde{\boldsymbol{\theta}}) = \sum_{i=1}^{N} \delta_i^{\hat{\boldsymbol{\theta}}_{\mathrm{pl}}} \phi_\pi^{\hat{\boldsymbol{\theta}}_{\mathrm{pl}}}(\boldsymbol{x}_i, \tilde{\boldsymbol{\theta}}) \dot{g}^{\otimes 2}(\boldsymbol{x}_i; \tilde{\boldsymbol{\theta}}) = \boldsymbol{G}^{\mathrm{T}}\boldsymbol{\Phi}\boldsymbol{G},
$$

where

$$
\boldsymbol{G} = \begin{pmatrix} 1 & \dot{f}(\boldsymbol{x}_1^{\mathrm{T}}\tilde{\boldsymbol{\beta}})\boldsymbol{x}_1^{\mathrm{T}} \\ 1 & \dot{f}(\boldsymbol{x}_2^{\mathrm{T}}\tilde{\boldsymbol{\beta}})\boldsymbol{x}_2^{\mathrm{T}} \\ \vdots & \vdots \\ 1 & \dot{f}(\boldsymbol{x}_N^{\mathrm{T}}\tilde{\boldsymbol{\beta}})\boldsymbol{x}_N^{\mathrm{T}} \end{pmatrix}
$$

and $\boldsymbol{\Phi} = diag\{\delta_i^{\hat{\boldsymbol{\theta}}_{\mathrm{pl}}}\phi_\pi^{\hat{\boldsymbol{\theta}}_{\mathrm{pl}}}(\boldsymbol{x}_i,\tilde{\boldsymbol{\theta}})\}$. Thus, we have

$$
\left\{\ddot{\ell}_{\mathrm{mscl}}^{\hat{\boldsymbol{\theta}}_{\mathrm{pl}}}(\tilde{\boldsymbol{\theta}})\boldsymbol{d}\right\}_{(j)} = \left(\boldsymbol{G}^{\mathrm{T}}\boldsymbol{\Phi}\boldsymbol{G}\boldsymbol{d}\right)^{\mathrm{T}}\boldsymbol{e}_j = (\boldsymbol{G}\boldsymbol{d})^{\mathrm{T}}\boldsymbol{\Phi}(\boldsymbol{G}\boldsymbol{e}_j) = (\boldsymbol{G}\boldsymbol{d})^{\mathrm{T}}\boldsymbol{\Phi}(\boldsymbol{G}\boldsymbol{e}_j) = (\boldsymbol{G}\boldsymbol{d})^{\mathrm{T}}\boldsymbol{\Phi}\boldsymbol{G}_{(j)}.
$$

Therefore, we can store $\boldsymbol{G}\boldsymbol{d}$ and keep updating $\boldsymbol{G}\boldsymbol{d}$ with

$$
\boldsymbol{G}(\boldsymbol{d} + z\boldsymbol{e}_j) = \boldsymbol{G}\boldsymbol{d} + z\boldsymbol{G}\boldsymbol{e}_j = \boldsymbol{G}\boldsymbol{d} + \boldsymbol{G}_{(j)}z.
$$

Thus, we do not need to obtain the full matrix $\ddot{\ell}_{\mathrm{mscl}}(\tilde{\boldsymbol{\theta}}) = \boldsymbol{G}^{\mathrm{T}}\boldsymbol{\Phi}\boldsymbol{G}$. We only need to calculate the diagnoal elements: $\ddot{\ell}_{\mathrm{mscl}(jj)}(\tilde{\boldsymbol{\theta}}) = \boldsymbol{G}_{(j)}^{\mathrm{T}}\boldsymbol{\Phi}\boldsymbol{G}_{(j)}, j = 1,...,p+1$ and $\left\{\dot{\ell}_{\mathrm{mscl}}(\tilde{\boldsymbol{\theta}})\boldsymbol{d}\right\}_{(j)} = (\boldsymbol{G}\boldsymbol{d})^{\mathrm{T}}\boldsymbol{\Phi}\boldsymbol{G}_{(j)}, j = 1,...,p+1$. From the analysis above, we can notice that the computaional complexity of one cycle calculating optimal direction $\boldsymbol{d}$ is $O(\zeta_{\mathrm{in}}Np)$, where $\zeta_{\mathrm{in}}$ denotes the number of inner iteration.

## C.2   Computational complexity

We analyze the computational complexity of the two-step algorithm. To facilitate the presentation, we consider a special case for our model when $g(\boldsymbol{x};\boldsymbol{\theta}) = \alpha + f(\boldsymbol{x}^{\mathrm{T}}\boldsymbol{\beta})$, and assume that the number of variables selected at the first-stage screening is $q$. Coordinate descent is a widely used optimization algorithm for solving lasso and adaptive lasso [see 9]. We consider the improved coordinate descent algorithm proposed in [27], which requires inner iterations to determine an optimal direction and outer iterations to update the estimator. Considering the form of $g(\boldsymbol{x};\boldsymbol{\theta}) = \alpha + f(\boldsymbol{x}^{\mathrm{T}}\boldsymbol{\beta})$, the computational complexity for coordinate descent with data of size $N$ and dimension $p$ is $O(\zeta_{\mathrm{in}}Np)$ per inner-cycle where $\zeta_{\mathrm{in}}$ represents the number of inner iterations (detailed derivations of this complexity is presented Section C). Thus, the computational complexity of full data lasso is $O(\zeta_{\mathrm{out}\times\mathrm{in}}Np)$, where $\zeta_{\mathrm{out}\times\mathrm{in}} = \zeta_{\mathrm{out}}\zeta_{\mathrm{in}}$ and $\zeta_{\mathrm{out}}$ is the number of outer iterations. The computational complexity of the full data adaptive lasso is $O(\zeta_{\mathrm{pl}}^{\mathrm{mle}}Np^2 + \zeta_{\mathrm{out}\times\mathrm{in}}Np)$ with the MLE as the pilot estimator, and $O(\zeta_{\mathrm{pl,out}\times\mathrm{in}}^{\mathrm{las}}Np + \zeta_{\mathrm{out}\times\mathrm{in}}Nq)$ with lasso as the pilot estimator, where $\zeta_{\mathrm{pl}}^{\mathrm{MLE}}$ and $\zeta_{\mathrm{pl,out}\times\mathrm{in}}^{\mathrm{las}}$ are the iteration numbers in the two pilot estimators, respectively. The coordinate descent algorithm often requires a large $\zeta_{\mathrm{out}\times\mathrm{in}}$ or $\zeta_{\mathrm{pl,out}\times\mathrm{in}}^{\mathrm{las}}$ while Newton's algorithm requires a small $\zeta_{\mathrm{pl}}^{\mathrm{mle}}$, so it is often the case that $\zeta_{\mathrm{pl}}^{\mathrm{mle}}p < \zeta_{\mathrm{out}\times\mathrm{in}}$. Therefore, the time complexity of the adaptive lasso is $O(\zeta_{\mathrm{out}\times\mathrm{in}}Np)$, which is the same as the full data lasso estimator.

Now, we analyze the time complexity of Algorithm 3. We start with the computational complexity of the optimal probabilities, for which the main computational cost is to approximate $\|\boldsymbol{M}_{(\mathcal{A})}^{-1}\dot{g}_{(\mathcal{A})}(\boldsymbol{x}_i;\boldsymbol{\theta})\|$ or $\|\boldsymbol{\Omega}_{(\mathcal{A})}^{1/2}\boldsymbol{M}_{(\mathcal{A})}^{-1}\dot{g}_{(\mathcal{A})}(\boldsymbol{x}_i;\boldsymbol{\theta})\|$, respectively, for $i = 1,...,N$. Since $\dot{g}_{(\hat{\mathcal{A}}_{\mathrm{pl}})}(\boldsymbol{x}_i;\boldsymbol{\theta}) = (1, \dot{f}(\boldsymbol{x}_i^{\mathrm{T}}\boldsymbol{\beta})\boldsymbol{x}_{i(\hat{\mathcal{A}}_{\mathrm{pl}})}^{\mathrm{T}})^{\mathrm{T}}$, the computational complexity of calculating $\dot{g}_{(\hat{\mathcal{A}}_{\mathrm{pl}})}(\boldsymbol{x}_i;\boldsymbol{\theta})$'s is $O(Nq)$, and the computational complexity of $\hat{\boldsymbol{M}}_{(\hat{\mathcal{A}}_{\mathrm{pl}})}^{\mathrm{pl}}$ or $\hat{\boldsymbol{\Omega}}_{(\hat{\mathcal{A}}_{\mathrm{pl}})}^{\mathrm{pl}}$ is $O\{N_{\mathrm{pl}}(q+1)^2\} = O(N_{\mathrm{pl}}q^2)$. Taking the inverse $(\hat{\boldsymbol{M}}_{(\hat{\mathcal{A}}_{\mathrm{pl}})}^{\mathrm{pl}})^{-1}$ and finding the square root $(\hat{\boldsymbol{\Omega}}_{(\hat{\mathcal{A}}_{\mathrm{pl}})}^{\mathrm{pl}})^{1/2}$ both take $O(q^3)$ time. Thus, the computational complexity of calculating $\|(\hat{\boldsymbol{M}}_{(\hat{\mathcal{A}}_{\mathrm{pl}})}^{\mathrm{pl}})^{-1}\dot{g}_{(\hat{\mathcal{A}}_{\mathrm{pl}})}(\boldsymbol{x}_i;\boldsymbol{\theta})\|$'s or $\|(\hat{\boldsymbol{\Omega}}_{(\hat{\mathcal{A}}_{\mathrm{pl}})}^{\mathrm{pl}})^{1/2}(\hat{\boldsymbol{M}}_{(\hat{\mathcal{A}}_{\mathrm{pl}})}^{\mathrm{pl}})^{-1}\dot{g}_{(\hat{\mathcal{A}}_{\mathrm{pl}})}(\boldsymbol{x}_i;\boldsymbol{\theta})\|$'s is $O(Nq + N_{\mathrm{pl}}q^2 + q^3 + Nq^2) = O(Nq^2)$. Therefore, the complexity of approximating the optimal probabilities in (4) or (7) is $O(Nq^2)$. The computational complexity of approximating the optimal probabilities in (5) is only $O(Nq)$, because there is no need to compute $\hat{\boldsymbol{M}}_{(\hat{\mathcal{A}}_{\mathrm{pl}})}^{\mathrm{pl}}$ or $\hat{\boldsymbol{\Omega}}_{(\hat{\mathcal{A}}_{\mathrm{pl}})}^{\mathrm{pl}}$. Next, we analyze the complexity of parameter estimation. The average subsample size with the sampling rate $\rho$ is on average

$$
\mathbb{E}(N_1) + \rho\{N - \mathbb{E}(N_1)\} = N\mathbb{E}\{f(\boldsymbol{x};\boldsymbol{\theta}_{\mathrm{t}})\}\{(1-\rho)e^{\alpha_{\mathrm{t}}} + \rho + o(1)\} = O\{N(e^{\alpha_{\mathrm{t}}} + \rho)\}.
$$

Using the coordinate descent algorithm, the computational complexity of the two-step algorithm is $O\{\zeta^{\text{las}}_{\text{pl,out}\times\text{in}}N_{\text{pl}}p + Nq^2 + \zeta_{\text{out}\times\text{in}}N(e^{\alpha_{\text{t}}} + \rho)q\}$ using optimal probabilities in (4) or (7), and it is $O\{\zeta^{\text{las}}_{\text{pl,out}\times\text{in}}N_{\text{pl}}p + Nq + \zeta_{\text{out}\times\text{in}}N(e^{\alpha_{\text{t}}} + \rho)q\}$ using the optimal probabilities in (5). For optimal probabilities in (4) or (7), when $\zeta_{\text{out}\times\text{in}} > q/(e^{\alpha} + \rho)$, the dominating term of the complexity is $\zeta_{\text{out}\times\text{in}}N(e^{\alpha_{\text{t}}} + \rho)q$. Remember that $\zeta_{\text{out}\times\text{in}} = \zeta_{\text{out}}\zeta_{\text{in}}$ and $\zeta_{\text{out}}$ is usually large for the coordinate descent algorithm. Therefore, $\zeta_{\text{out}\times\text{in}} > q/(e^{\alpha} + \rho)$ is often satisfied in practice. The dominating term for the time complexity of optimal probabilities in (5) is also $\zeta_{\text{out}\times\text{in}}N(e^{\alpha_{\text{t}}} + \rho)q$. Compared with full data estimators, both the sample size and the dimension are reduced. If we set the subsample size to be the same order of $N_1$, which is often the case in practice for balancing the ones and zeros, the time complexity of Algorithm 2 is of order $O(\zeta_{\text{out}\times\text{in}}Ne^{\alpha_{\text{t}}}q)$, which is significantly faster than that of the full data estimator.

## D  Detalis of simulation settings

In this section, we present more details of the simulation settings in the main paper. In Section D.1, we provide detailed simulation settings of the example in Section 1. In Section D.2, we present detailed settings in Section 5.

### D.1  Simulation in Section 1

We first present the detailed settings of the simulations in Section 1, where we illustrate the scale-dependent issues of optimal subsampling probabilities. Our simulation based on logistic regression models with the true parameter $\boldsymbol{\beta}_{\text{t}}$ to be 6-dimentional vectors and covariates $\boldsymbol{x} \sim lognormal(\mathbf{0}, \boldsymbol{\Sigma})$ with the $(i, j)$-th element of $\boldsymbol{\Sigma}$ is given as $\boldsymbol{\Sigma}_{ij} = 0.5^{|i-j|}, 1 \leq i, j \leq 6$. We consider two cases of parameters:

(a) **Non-sparse parameter**: $\boldsymbol{\beta}_{\text{t}} = (-1, -1, -0.01, -0.01, -0.01, -0.01)$ and $\alpha_{\text{t}} = -4$.

(b) **Sparse parameter**: $\boldsymbol{\beta}_{\text{t}} = (-1, 0, 0, 0, 0, 0)$ and $\alpha_{\text{t}} = -5$.

We generate full data of size $N = 500000$ according to the above logistic models. To investigate the effects of scale transformation, we multiply the $\boldsymbol{x}_{(6)}$ with $s$ ($s = 0.01, 0.1, 1, 10, 100$) and divide $\boldsymbol{\beta}_{\text{t}(6)}$ with the same $s$ to remain $\boldsymbol{x}^{\text{T}}\boldsymbol{\beta}_{\text{t}}$ to be the same and thus the logistics regression model does not change. We obtain subsamples with optimal subsampling probabilities described in [22] with transformed $\boldsymbol{x}$ under each $s$ and calculate the resultant subsampling estimators. We set the nominal pilot sample size to $N_{\text{pl}} = 800$ and nominal subsample size $N_{\text{sub}} = 1000$ (see details in [22]). We repeat the experiment for 500 times under each scale and compute the mean prediction error.

### D.2  Simulations in Section 5

For the estimation procedures in the second step of our two-step algorithm, we choose $\gamma = 1$, which means that the weights in the adaptive lasso penalty are $\hat{w}_j = 1/|\hat{\beta}_{\text{pl}(j)}|, 1 \leq j \leq q$, with $q$ being the number of selected variables in the first stage screening. Furthermore, we consider uniform sampling, the full data lasso and the full data adaptive lasso as baselines for comparison. For the uniform sampling method, we use a similar two-step algorithm as presented in Algorithm 2 but set the sampling function in the second step as $\varphi(\boldsymbol{x}) = 1$, which means the sampling probabilities are a constant $\rho$. We use lasso to implement the first stage screening and adaptive lasso with $\gamma = 1$ to implement the second stage screening for a fair comparison. For full data lasso, we directly apply the lasso algorithm to the full data. For the full data adaptive lasso, we use the full data MLE estimator as the pilot estimator to construct the weights and then apply the adaptive lasso algorithm to the full data set.

## E  Additional simuations

In this Section, we give some addtional simulation results. More simulation results of variable selection is provided in Section E.1. We also provide addtional simulation results to compare our approach with standardization to resolve scale dependent issues in Section E.2.

## E.1 Addtional variable selection results

Table 4: Mean number of selected variables in Case A and Case B

| $\rho$ | first-stage | Uni | A-OS | L-OS | P-OS |
|---|---|---|---|---|---|
| | Case A (five active variables) | | | | |
| 0.0025 | 14.87(0.30) | 4.99(0.02) | 5.02(0.02) | 5.03(0.02) | 5.01(0.02) |
| 0.005 | 14.68(0.28) | 5.05(0.02) | 5.07(0.02) | 5.06(0.02) | 5.08(0.02) |
| 0.0075 | 14.20(0.27) | 5.09(0.03) | 5.08(0.02) | 5.08(0.02) | 5.09(0.03) |
| 0.01 | 14.46(0.30) | 5.13(0.02) | 5.13(0.02) | 5.13(0.02) | 5.12(0.03) |
| | Case B (four active variables) | | | | |
| $\rho$ | first-stage | Uni | A-OS | L-OS | P-OS |
| 0.0025 | 16.81(0.33) | 3.91(0.02) | 4.01(0.03) | 4.00(0.02) | 4.01(0.03) |
| 0.005 | 17.88(0.34) | 4.04(0.03) | 4.10(0.03) | 4.08(0.04) | 4.07(0.03) |
| 0.0075 | 17.19(0.33) | 4.06(0.02) | 4.13(0.03) | 4.10(0.03) | 4.12(0.03) |
| 0.01 | 17.51(0.34) | 4.07(0.03) | 4.08(0.03) | 4.08(0.03) | 4.08(0.03) |

Table 5: Rates of excluding active variables (false negative rate) in Case A and Case B

| $\rho$ | Uni | A-OS | L-OS | P-OS |
|---|---|---|---|---|
| | Case A | | | |
| 0.0025 | 0.094(0.013) | 0.076(0.012) | 0.068(0.011) | 0.070(0.012) |
| 0.005 | 0.052(0.010) | 0.048(0.010) | 0.050(0.010) | 0.044(0.010) |
| 0.0075 | 0.054(0.010) | 0.052(0.010) | 0.052(0.010) | 0.052(0.010) |
| 0.01 | 0.040(0.009) | 0.036(0.008) | 0.036(0.008) | 0.036(0.008) |
| | Case B | | | |
| $\rho$ | Uni | A-OS | L-OS | P-OS |
| 0.0025 | 0.108(0.014) | 0.074(0.012) | 0.074(0.012) | 0.070(0.012) |
| 0.005 | 0.046(0.009) | 0.034(0.008) | 0.034(0.008) | 0.036(0.008) |
| 0.0075 | 0.062(0.011) | 0.046(0.009) | 0.046(0.009) | 0.044(0.009) |
| 0.01 | 0.056(0.010) | 0.050(0.010) | 0.052(0.010) | 0.050(0.010) |

Table 6: Rates of selecting the true model

| $\rho$ | Uni | A-OS | L-OS | P-OS |
|---|---|---|---|---|
| | Case A | | | |
| 0.0025 | 0.856(0.016) | 0.870(0.015) | 0.868(0.015) | 0.878(0.015) |
| 0.005 | 0.884(0.014) | 0.872(0.015) | 0.880(0.015) | 0.872(0.015) |
| 0.0075 | 0.858(0.016) | 0.868(0.015) | 0.868(0.015) | 0.868(0.015) |
| 0.01 | 0.854(0.016) | 0.866(0.015) | 0.862(0.015) | 0.868(0.015) |
| | Case B | | | |
| $\rho$ | Uni | A-OS | L-OS | P-OS |
| 0.0025 | 0.862(0.015) | 0.872(0.015) | 0.864(0.015) | 0.876(0.015) |
| 0.005 | 0.898(0.014) | 0.888(0.014) | 0.898(0.014) | 0.902(0.013) |
| 0.0075 | 0.848(0.016) | 0.850(0.016) | 0.854(0.016) | 0.848(0.016) |
| 0.01 | 0.874(0.015) | 0.874(0.015) | 0.876(0.015) | 0.872(0.015) |
| | Case C | | | |
| $\rho$ | Uni | A-OS | L-OS | P-OS |
| 0.0025 | 0.824(0.017) | 0.874(0.015) | 0.880(0.015) | 0.884(0.014) |
| 0.005 | 0.880(0.015) | 0.884(0.014) | 0.884(0.014) | 0.882(0.014) |
| 0.0075 | 0.908(0.013) | 0.916(0.012) | 0.910(0.013) | 0.914(0.013) |
| 0.01 | 0.908(0.013) | 0.910(0.013) | 0.908(0.013) | 0.910(0.013) |

It is seen in Table 6 that, no subsampling method dominates others. Table 2 and Table 5 shows that uniform sampling has higher rates of excluding active variables than optimal subsampling procedures.

Although uniform sampling may have a higher rate of selecting the true model in some cases, given that it is more likely to exclude important variables, optimal sampling may be preferable in practice.

## E.2 Comparison with standardization

Another approach to avoid scale-dependency is to standardize the data. We compare the proposed scale-independent optimal probabilities with the approach of data standardization here. For the data standardization approach, we standardize the data, calcualte the optimal probabilities, and then implement subsampled adpative lasso algorithm. We used the same pilot estimation methods for fair comparisons.

We first compare the eMSE and eMSPE in Figure 5 and Figure 6, respectively. We use sP-OS to denote the approach with data standardization and use P-OS to denote the approach without data standardization.

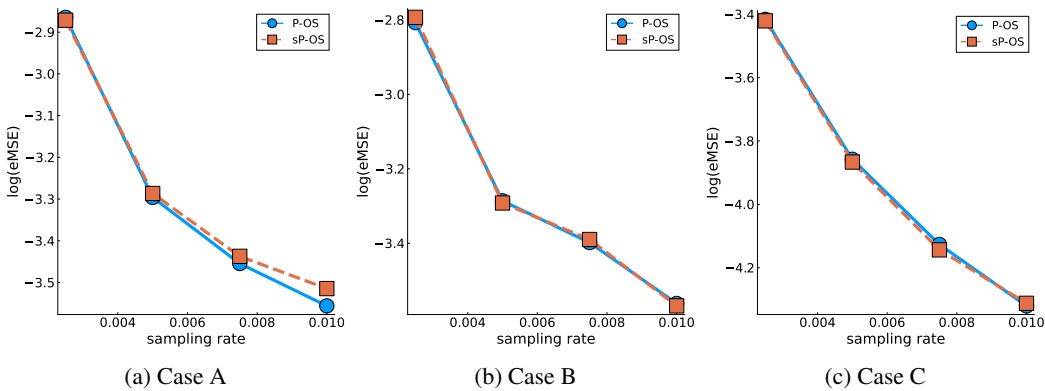

(a) Case A           (b) Case B           (c) Case C

Figure 5: Empirical median squred error of estimated probability for different parameters with different sampling rates. The same pilot sample size is $N_{\mathrm{pl}} = 500$.

In Figure 5, we notice that the performances of P-OS and sP-OS are similar, this is also true for eMSPE. However, standardization may decrease the rate of selecting the true model. We present results of variable selection in Table 7 and Table 8.

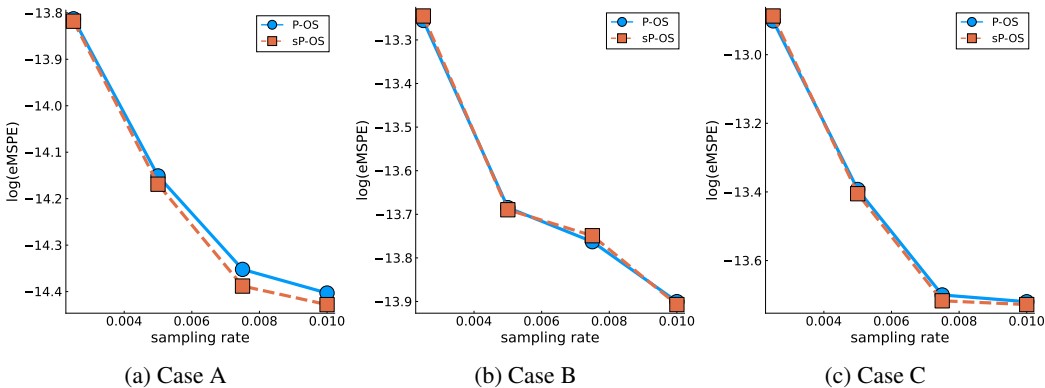

(a) Case A           (b) Case B           (c) Case C

Figure 6: Empirical median squred error of estimated probability for different parameters with different sampling rates. The same pilot sample size is $N_{\mathrm{pl}} = 500$.

We notice in Table 7 and Table 8 that the rates of selecting true models by $\hat{\beta}^{\mathrm{adp}}_{\mathrm{P-OS}}$ is higher than $\hat{\beta}^{\mathrm{adp}}_{\mathrm{sP-OS}}$ without much increase on the rates of excluding active variables. Therefore, although standardization is an approach to solve the scale-dependency issues, it may decrease the rates of selecting true models in practice.

Table 7: Rates of selecting true models

|  | A | | | B | | | C | | |
| $\rho$ | sUni | P-OS | sP-OS | sUni | P-OS | sP-OS | sUni | P-OS | sP-OS |
| --- | --- | --- | --- | --- | --- | --- | --- | --- | --- |
| 0.0025 | 0.848 | 0.878 | 0.862 | 0.850 | 0.876 | 0.862 | 0.828 | 0.884 | 0.880 |
| 0.005 | 0.878 | 0.872 | 0.850 | 0.890 | 0.902 | 0.890 | 0.882 | 0.882 | 0.880 |
| 0.0075 | 0.850 | 0.868 | 0.846 | 0.850 | 0.848 | 0.842 | 0.906 | 0.914 | 0.910 |
| 0.01 | 0.840 | 0.868 | 0.844 | 0.868 | 0.872 | 0.862 | 0.900 | 0.910 | 0.904 |

Table 8: Rates of excluding active variables (false negtive rate)

|  | A | | | B | | | C | | |
| $\rho$ | sUni | P-OS | sP-OS | sUni | P-OS | sP-OS | sUni | P-OS | sP-OS |
| --- | --- | --- | --- | --- | --- | --- | --- | --- | --- |
| 0.0025 | 0.088 | 0.070 | 0.068 | 0.106 | 0.070 | 0.070 | 0.164 | 0.084 | 0.084 |
| 0.005 | 0.056 | 0.044 | 0.044 | 0.046 | 0.036 | 0.036 | 0.100 | 0.066 | 0.066 |
| 0.0075 | 0.052 | 0.052 | 0.052 | 0.062 | 0.044 | 0.044 | 0.062 | 0.046 | 0.046 |
| 0.01 | 0.040 | 0.036 | 0.036 | 0.056 | 0.050 | 0.050 | 0.068 | 0.054 | 0.054 |

