# OpenReview forum: "Scale-invariant Optimal Sampling for Rare-events Data and Sparse Models"
_NeurIPS.cc/2024/Conference — NeurIPS 2024 poster_

### Official Review · Reviewer_8W9n · 2024-07-12

**Soundness:** 3
**Presentation:** 3
**Contribution:** 3
**Rating:** 6
**Confidence:** 2

**Summary:**

This paper studies the problem of learning a sparse model under rare events and scale-invariance. They provide optimal subsampling function to handle scale-invariance for a Lasso-regularized model. They finally provide experimental results for their subsampling scheme.

**Strengths:**

* Well-written
* Looking at MSPE rather than asymptotic variance seems like a good way to get around the scaling issue.

**Weaknesses:**

* This problem setting feels very specific: rare events data, sparse models, and scale-invariance. It would be good to motivate why we should care about all of these simultaneously.
* None of the theoretical results given in the main body seem to have any discussion about proof strategy or novel techniques. It would be interesting to know what technical challenges were posed in developing these results, as well as how they were resolved.
* In section 4, the paper compares the proposed penalized MSCL estimator to the MSCL estimator. As far as I can tell, this MSCL estimator has not been defined in the paper and so these comparisons and discussions are hard to make sense of.

**Questions:**

* In Proposition 1, why is minimizing $tr(M_{w(A)})$ of interest? It seems like all one would care about is minimizing the asymptotic variance (which as far as I understand is $V_{w(A)}$).
* In the preamble of section 3.1 (lines 156-158), the MSPE is motivated by the desire for a scale-invariant objective. Is the problem with the previous approach(es) that scaling $x_j$ would inversely scale the corresponding entries of $V_{w(A)}$?
* In the preamble of section 4 (lines 183 and 184) that discusses the inefficiency of the IPW estimator, the paper says IPW gives less weight to more informative points. Is this a formal (or at least, formalizable) claim, or is it just a heuristic? Either way, can this be elaborated upon?
* The optimality of this work seems to be the optimal subsampling function towards the objective of MSPE. Is it possible to argue that this estimator is information-theoretically the best that one can do?

**Limitations:**

The authors addressed their limitations.

---

> ### Author Rebuttal · Authors · 2024-08-03
>
> Thank you for your insightful review of our paper.
>
> We would like to point out that rare events data and sparse models are common in
> practice, and scale-dependence is a frequent and crucial issue in subsampling
> that has not been addressed in existing literature. Following your great
> suggestion, we will provide more details on the motivation behind our paper's
> settings in the revised version.
>
> We did not compare the penalized MSCL estimator to the MSCL estimator. The
> penalized MSCL estimator is defined in equation (9), while the MSCL estimator is
> simply this without the last penalty term. We did not separately define the MSCL
> as it is not implemented in our paper. We will clarify this in the final
> version. Furthermore, your comment has made us realize that using the subscript
> "mscl" instead of "lik" would more clearly indicate the penalized MSCL
> estimator. We will make this change in the revision.
>
> Please see the following for our response to your specific comments/questions.
>
> **Q1. The reason why minimizing $tr(M_{w(A)})$ is of interest in Proposition 1.**
>
> **R1.** Minimizing $tr(M_{w(A)})$ is of interest because it is the asymptotic
> variance of estimating $M_{(A)}\theta_{t(A)}$, a linearly transformed
> parameter. One advantage of this criterion is it often leads to optimal
> subsampling probabilities that are easier to calculate. We will provide further
> clarification on this point in the final revision.
>
> **Q2. The problem of previous approaches.**
>
> **R2.** Thank you for your insightful question. Yes, the problem with previous
> approaches is essentially caused by the fact that scaling $x_j$ would scale some
> entries of $V_{w(A)}$. However, the resulting effects on the optimal
> probabilities are more complex than might be expected. For example, Figure 1
> demonstrates that scaling a single covariate has markedly different effects on
> the A-OS and L-OS probabilities.
>
> **Q3. Discussion of the inefficiency of the IPW estimator.**
>
> **R3.** The statement that the IPW estimator is not the most efficient due to
> its assignment of lower weights to more informative data points is a heuristic
> explanation for its inefficiency. More specifically, optimal sampling
> probabilities are constructed so that more informative data points have higher
> probabilities $\pi(\mathbf{x}_i^{\mathrm{sub}}, y_i^{\mathrm{sub}})$'s of being
> included in the subsample. However, the IPW estimator assigns smaller weights of
> $1/\pi(\mathbf{x}_i^{\mathrm{sub}}, y_i^{\mathrm{sub}})$'s. This inverse
> relationship suggests that the IPW estimator does not fully utilize the
> information contained in the optimal subsample, which intuitively explains its
> reduced efficiency.
>
> Furthermore, we can also formally prove that the IPW estimator is less efficient
> than the penalized MSCL estimator. Here is a sketch of the proof. Let
>
> $$ h = 1 + c \{\varphi(\mathbf{x})\}^{-1} e^{f(\mathbf{x}; \beta_t)}, $$
>
> $$ \mathbf{v} = \sqrt{e^{f(\mathbf{x}; \beta_t)}} \dot{g}_{(A)}(\mathbf{x}; \theta_t), $$
>
> $$ \mathbf{f} = h^{1/2} \mathbf{v}, $$
>
> $$ \mathbf{g} = h^{-1/2} \mathbf{v}. $$
>
> We have that
>
> $$ \mathbb{E}(\mathbf{g} \mathbf{f}^{T}) = \mathbb{E}(\mathbf{f} \mathbf{g}^{T}) = \mathbb{E}(\mathbf{v} \mathbf{v}^{T}) = M_{(A)}, $$
>
> $$ \mathbb{E}(\mathbf{f} \mathbf{f}^{T}) = M_{w(A)}, $$
>
> $$ \mathbb{E}(\mathbf{g} \mathbf{g}^{T}) = \Lambda_{mscl(A)}. $$
>
> Then, the matrix form of Cauchy-Schwarz's inequality [ref1],
>
> $$ \mathbb{E}(\mathbf{g} \mathbf{g}^{T}) \geq \mathbb{E}(\mathbf{g} \mathbf{f}^{T}) \{\mathbb{E}(\mathbf{f} \mathbf{f}^{T})\}^{-1} \mathbb{E}(\mathbf{f} \mathbf{g}^{T}), $$
>
> leads to
>
> $$ \Lambda_{mscl(A)} \ge V_{w(A)}^{-1}, $$
>
> which implies that
>
> $$ V_{mscl(A)} \le V_{w(A)}. $$
>
> In the revision, we will elaborate on this point, provide a detailed proof, and
> further clarify why the IPW estimator is not the most efficient.
>
> [ref1] Tripathi, Gautam. "A matrix extension of the Cauchy-Schwarz inequality."
> Economics Letters 63.1 (1999): 1-3.
>
> **Q4. Theoretical optimality of the estimator**
>
> **R4.** The proposed penalized MSCL estimator is indeed the best in the sense
> that it is the most efficient estimator among a large class of asymptotically
> unbiased estimators. Specifically, the asymptotic variance $V_{mscl(A)}$ attains
> the Cramer-Rao lower bound for this class of estimators. This result formally
> establishes the estimator's optimality within the class. We will present this
> finding in detail, along with a detailed proof, in the future version of this
> paper.

---

> > ### Comment · Reviewer_8W9n · 2024-08-12
> >
> > Thank you for your response, it clarified a lot of my questions. I will be increasing my score.

---

> > > ### Author Response · Authors · 2024-08-13
> > >
> > > Thank you very much for raising the score on our paper!

---

### Official Review · Reviewer_2g5r · 2024-07-14

**Soundness:** 3
**Presentation:** 3
**Contribution:** 3
**Rating:** 7
**Confidence:** 3

**Summary:**

The scale-invariant optimal subsampling function proposed in the paper addresses the challenge of inactive features in rare-events data by overcoming the issue of scale-dependence in exitsing optimal subsampling methods. In the context of variable selection for rare-events data, where distinguishing active and inactive features is very important, the scale-invariant optimal subsampling function minimizes prediction error regardless of scaling transformations applied to inactive features.

The key is to provide reliable and unbiased subsampling probabilities even when inactive variables are transformed without altering the underlying model. This is important because inappropriate scaling transformations of inactive features can significantly impact traditional optimal subsampling methods, leading to unreliable or misleading results. This resluts in improving the accuracy of parameter estimation and variable selection.

**Strengths:**

Good novelty:The paper introduces a novel scale-invariant optimal subsampling function to address the challenge of inactive features in rare-events data, offering a new approach to improving parameter estimation and variable selection in sparse moedls. This novel contribution could advance the field of rare-events data analysis.

Strong theory:The paper establishes a theoretical foundation for the proposed scale-invariant optimal subsampling function, including discussions on assumptions, proofs, and the mathematical framework supporting the method. This strong theoretical basis enhances the credibility and rigor of the proposed approach. WHile i have not gone through all the proofs, i think  the results seem sound.

Sufficient empirical evidence: Further, the paper demonstrates the practical relevance of the scale-invariant optimal subsampling function through numerical experiments using both simulated and real-world datasets.The results show non-trivial improvements in prediction error minimization and estimation efficiency.

Clarity: The paper is largely well-written,, provides clear and detailed explanations of the methodology, assumptions, and implications of the scale-invariant optimal subsampling function, making it accessible even though there is a lot of notation and theory.

**Weaknesses:**

I think this is a good paper that should be accepted. Minor gripe:

It is not clear to me how exactly the methods in section 1 are affected due to x (and thereby making them NOT scale invariant). I understand this would require going through the earlier papers e..g [21], which i did not do. It would make the paper complete if the authors could adda discussion to this effect in the appendix in the final version.

**Questions:**

Why is the tr(variance) sufficient ot be minimized in the presented results ? In general most of the discussion revolves around assymptotic results, is it interesting to ask questions about non-assymptotic behaviors, such as how would performance of variable selection impact non-assymptotic prediction error? Further, many practical applications could be overparametrized/high-dimensional, how would the results carry over? If not, what would be complications that arise while generalizing them to such settings?

**Limitations:**

Yes

---

> ### Author Rebuttal · Authors · 2024-08-03
>
> Thank you for your insightful and positive review of our paper. We appreciate your constructive suggestion on providing a detailed discussion on why the A-OS
> and L-OS in Section 1 are affected by the scaling of the covariate
> $\mathbf{x}$. Yes, we also believe the point would be clarified by the formulas
> for the A-OS and L-OS probabilities derived from [21] for producing Figure 1. In
> the final version, we will add these details along with a comprehensive
> discussion of the example in Section 1 to the appendix.
>
> Please see our responses to your specific questions below.
>
> **Q1. Sufficiency of minimizing $tr(\mathbf{V}_{w(\mathcal{A})})$.**
>
> **R1.** The subsample estimator is asymptotically unbiased (Theorem 1), implying that lower variance indicates a better subsample estimator. Optimal subsampling aims to improve estimation efficiency by finding subsampling probabilities that minimize variance. However, due to the lack of complete ordering in variance matrices, the trace is often used as a criterion for minimization. For an asymptotically unbiased estimator, the trace ${tr}(\mathbf{V}_{w(\mathcal{A})})$ is
> interpreted as the asymptotic mean squared error (MSE), making its usage appreciated
> in practice.
>
> **Q2. Discussions on non-asymptotic behaviors.**
>
> **R2.** Non-asymptotic behaviors, such as prediction error, are indeed of
> interest in variable selection. Our numerical experiments actually assess these non-asymptotic performance. However, non-asymptotic theoretical results are typically expressed as error bounds, which may be less applicable for defining optimal subsampling probabilities. The asymptotic distribution, on the other hand, provides a direct
> approximation to the distributional behavior of the subsample estimator, which makes it more suitable for defining optimal subsampling probabilities. We do not intend to downplay the importance of non-asymptotic behaviors. In fact, we may investigate the theoretical non-asymptotic behaviors of our estimator in future research.
>
> **Q3. Applications in overparameterized/high-dimensional problems.**
>
> **R3.** For overparameterized/high-dimensional problems, if the model is sparse
> enough that the true model is low-dimensional, we conjecture that results such
> as the scale-invariant probabilities and the theoretical properties of the MSCL
> estimator still hold under some additional assumptions. However, if the model is
> not sparse, our results do not carry over. One reason is that asymptotic
> normality may no longer hold. While it may be possible to derive some
> non-asymptotic error bounds, it is unclear how to use them to define better
> subsampling probabilities for the problem in consideration.

---

> > ### Comment · Reviewer_2g5r · 2024-08-09
> >
> > Thank you for your response and the clarifications. I would recommend updating the paper with these to make it stronger.

---

> > > ### Author Response · Authors · 2024-08-12
> > >
> > > We sincerely appreciate your valuable comments and suggestions. We will incorporate these points and update the paper accordingly.

---

### Official Review · Reviewer_RXcc · 2024-07-15

**Soundness:** 4
**Presentation:** 4
**Contribution:** 3
**Rating:** 8
**Confidence:** 4

**Summary:**

This paper introduces an optimal subsampling algorithm designed for imbalanced data with inactive features. The primary algorithm can be summarized in three steps:

1. Train a pilot model using a lasso-penalized objective.
2. Subsample the data based on the pilot model.
3. Train the subsampled data using an adaptive lasso-penalized algorithm.

Additionally, the paper proposes two options for the model estimator: the IPW estimator and the more accurate MSCL estimator. Empirical results demonstrate the algorithm's effectiveness, showing that it outperforms the baseline estimator which does not apply variable selection.

**Strengths:**

1. The paper addresses a key drawback of a previous "optimal estimator" for rare-events data by incorporating a lasso-based estimator.
2. The proposed algorithm is theoretically sound with solid asymptotic guarantees.
3. Empirical results on both synthetic and real data highlight the algorithm's effectiveness.

**Weaknesses:**

No technical weaknesses were observed in this paper.

**Questions:**

In the first stage screening of Algorithm 2, we use the Lasso estimator to manage computational costs and noise in the sampled data. If more computational resources are available, is Lasso always the best practical choice for pilot estimation?

**Limitations:**

No significant limitations were observed.

---

> ### Author Rebuttal · Authors · 2024-08-02
>
> Thank you very much for your positive feedback on our work along with the
> insightful question. Please see below for our response.
>
> **Q1. If more computational resources are available, is Lasso always the best
> practical choice for pilot estimation?**
>
> **R1.** In general, better pilot estimation leads to a better final
> estimator. While we recommend the Lasso method, it may not be the best choice in
> all cases. More computationally intensive methods can achieve higher accuracy
> than the Lasso. However, if more computational resources are available, it is
> more beneficial to allocate additional resources to subsequent steps rather than
> focusing solely on pilot estimation. Although we cannot assert that the Lasso is
> always the best choice, it is a robust and practical option for the problem at
> hand.
>
> We recommend using the Lasso method for pilot estimation due to its simultaneous
> parameter estimation and active set selection. In addition, it tends to be
> conservative, meaning it has a low risk of excluding important variables. Other
> methods that combine parameter estimation and variable selection, such as sure
> independence screening, may also be considered. We do not recommend using the
> maximum likelihood estimator (MLE) under the full model because including many
> inactive variables reduces the efficiency of the approximated optimal
> subsampling function.
>
> For further illustration, we conducted a preliminary simulation to compare the performance of the Lasso
> pilot (Lasso), Sure Independence Screening (SIS) pilots, and the
> Lasso-followed-by-MLE pilot (LassoMLE). The SIS requires to pre-assign the
> number of variables to select, for which we considered 3, 6, 13, and 20, and they are labeled as SIS3, SIS6, SIS13, and SIS20, respectively. The LassoMLE calculates the MLE
> based on the variables selected by the Lasso.  Below are the eMSEs of the final
> P-OS estimator from 200 iterations of the simulation for Section 5.1 Case C of the main paper:
>
> | $\rho$ | Lasso |  SIS3 |  SIS6 | SIS13 | SIS20 | LassoMLE |
> |-------:|------:|------:|------:|------:|------:|---------:|
> | 0.0025 | 0.031 | 0.735 | 0.026 | 0.039 | 0.049 |    0.053 |
> | 0.0050 | 0.026 |  0.73 | 0.017 | 0.029 | 0.033 |    0.039 |
> | 0.0075 | 0.015 | 0.729 | 0.015 | 0.021 | 0.026 |    0.030 |
>
> The table above reveals that the SIS pilot may yield better results than the Lasso pilot when the pre-assigned number of variables is 6 (SIS6). However, determining the optimal pre-assigned number of variables in practice requires further investigation. On
> average, the Lasso pilot selects 13 variables and shows better performance than the SIS13
> pilot. Unexpectedly, we found that the LassoMLE pilot performed worse than
> the Lasso pilot. Upon further examination, we discovered that the LassoMLE produces larger absolute estimates for some inactive variables. Consequently, these inactive variables receive a smaller penalty in the final adaptive Lasso
> estimator, as the penalty is proportional to $1/|\hat{\beta}|$.
>
> We greatly appreciate your insightful question, which has led to these additional interesting findings. The choice of the optimal pilot estimation method is indeed critical in practice, especially for sparse models. We plan to further investigate the effect of different pilot estimation procedures on the performance of the proposed algorithm.

---

### Author Rebuttal · Authors · 2024-08-03

We sincerely thank all the reviewers for their time and effort in assessing our
work. Their insightful comments and questions are valuable in helping us improve
the paper. Please find our individual responses to the reviewers' comments and
questions below.

---

> ### Comment · Area_Chair_tKHC · 2024-08-12
> **Further comments from Reviewers RXcc and 8W9n?**
>
> Dear Reviewers RXcc and 8W9n:
>
> Can you please respond to the rebuttal as soon as possible? Your comments will be greatly appreciated. Many thanks,
>
> AC

---

### Decision · Program_Chairs · 2024-09-25

**Decision:**

Accept (poster)

**Comment:**

This paper provides scale-invariant optimal subsampling probabilities that minimize the predictor error of the adaptive lasso for rare binary responses. The proposed algorithm involves two steps: (i) to calculate approximate optimal sampling probabilities using a pilot sample and (ii) to compute an optimally sub-sampled adaptive lasso estimator. For the latter, the paper proposes two options: the inverse probability weighted (IPW) estimator and the more accurate maximum sampled conditional likelihood (MSCL) estimator. It would make the paper more influential if the author(s) can improve the following aspects among other things: (i) the motivation for the estimation problem; (ii) discussion on what sense scale invariance matters for optimal subsampling probabilities; (iii) discussion of the inefficiency of the IPW estimator as well as the optimality of the MSCL estimator; (iv) limitations in terms of finite-sample bounds and the effect of different pilot estimation procedures.